# JIR-Arena: The First Comprehensive Benchmark Dataset for Just-in-time Information Recommendation

## Abstract

Just-in-time Information Recommendation (JIR) is a service that delivers the most relevant information precisely when users need it the most. It plays a critical role in filling users' information gaps during pivotal moments like those in learning, work, and social interactions, thereby enhancing decision-making quality and life efficiency with minimal user effort. Recent device-efficient deployment of performant foundation models and the proliferation of intelligent wearable devices have made the realization of always-on JIR assistants feasible. However, despite the potential of JIR systems to transform our daily life, there has been little prior systematic effort to formally define JIR tasks, establish evaluation frameworks, or propose a large-scale multimodal benchmark with high-quality multi-party-sourced ground-truth labels. To bridge this gap, we present a comprehensive mathematical definition of JIR tasks and their associated evaluation metrics. Furthermore, we introduce JIR-Arena, the first multimodal JIR benchmark dataset comprising 34 scenes (831 minutes) with oracle information needs covering 11 types with diverse and information-request-intensive scenarios, designed to evaluate JIR systems across multiple dimensions, including whether they can *i)* accurately infer user information needs, *ii)* provide timely and helpfully relevant recommendations, and *iii)* effectively avoid the inclusion of irrelevant content that might distract users.

Constructing a JIR benchmark is challenging due to the subjectivity of user information needs and the difficulty of achieving reproducible evaluations. To overcome these, our benchmark approximates user need distribution by combining human and large AI model inputs, and enhances objectivity through a multi-turn validation framework. Additionally, we ensure assessment reproducibility by evaluating information recommendation outcomes against static knowledge bases. We also develop a baseline JIR system architecture, and instantiate it with several large foundation models. Our evaluation of the baselines on JIR-Arena reveals that while large foundation model-based JIR systems can simulate user needs with reasonable precision (72.4% average), they struggle with recall (34.7% average) and effective content retrieval. The analysis identifies dual bottlenecks in both user information need prediction and retrieval systems, with semantic mismatch of predicted information need (62.9% of failures) being the primary failure mode. Finally, to facilitate future development of JIR systems and exploration of more JIR application scenarios, we release our code and data in the supplementary materials.

## 1 Introduction

The landscape of information service is transforming from a *reactive* to a *proactive* paradigm. For decades, our information access tools—from search engines to chat assistants—have waited for explicit commands, requiring us to interrupt our workflow, formulate a query, and manually *pull* the information we need. This paradigm, while powerful, is limited in situations where a user is constrained or may not even recognize their own information gap. Just-in-Time Information Recommendation (JIR) represents the next frontier in this evolution. It aims to create an assistant that anticipates our latent needs from context and proactively *pushes* relevant knowledge to us with minimal effort, augmenting our intelligence and productivity. For instance, as shown in Figure 1, during lectures JIR can instantly provide succinct explanations of unfamiliar concepts to help make learning more efficient. Promisingly, the recent convergence of portable/wearable

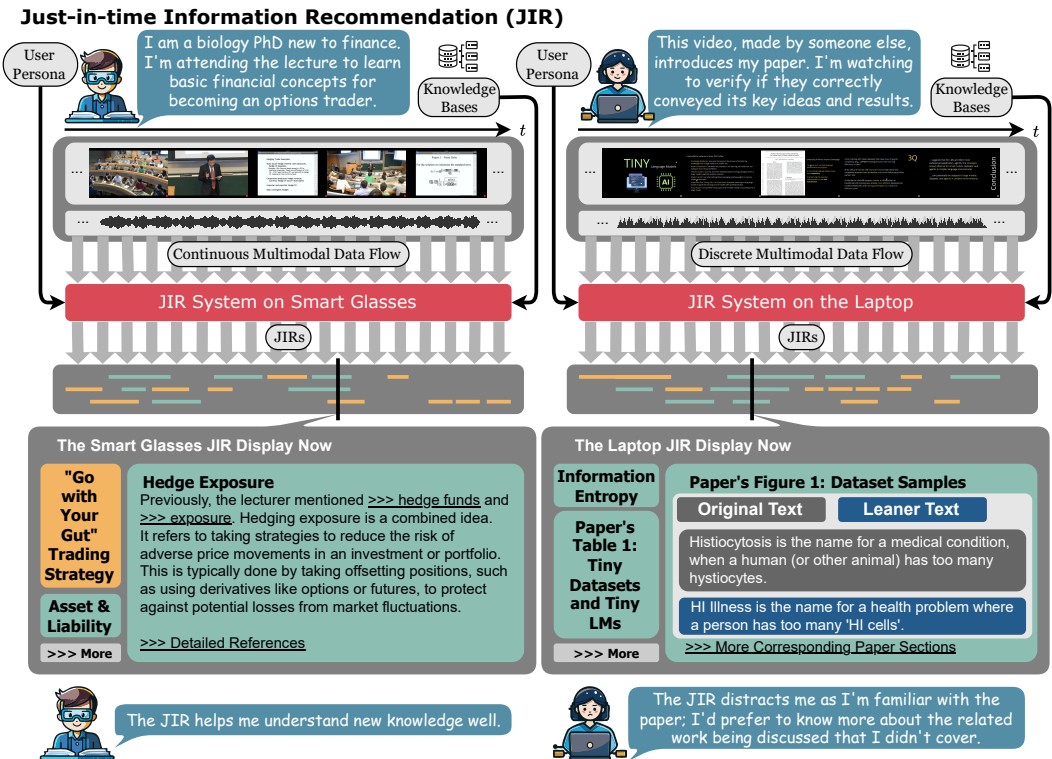

Figure 1: The JIR system takes in user persona and multimodal data stream, and proactively outputs JIRs based on knowledge bases to satisfy the user's instant information needs.

devices (Meizu, 2025; Meta, 2025) and affordable AI deployment (Microsoft et al., 2025; Yang et al., 2025) has made possible performant AI-powered JIR systems to be integrated into everyday devices—from smartphones to VR glasses—capable of addressing immediate information needs with few distractions.

Despite its transformative potential, the advancement of JIR has been impeded by a critical evaluation gap. Unlike many related fields of traditional recommendation (e.g., query suggestion, content recommendation, etc.), JIR is unique in the sense that information is provided to a user with respect to an ongoing stream of information, as presented in Figure 1. This means that evaluation metrics must take into account the *timeliness* aspect besides the *relevance*. Moreover, benchmarks must be carefully constructed to ensure a concrete coupling between this stream of information and the recommended content. Rather than build these benchmarks and evaluation metrics, much of the historical work has evaluated JIR systems from the perspective of user studies (e.g., Rhodes (2000); OpenSearch (2025); Liu et al. (2023)), thus making it difficult to generalize model performance or advance JIR algorithms.

To address the evaluation gap and accelerate research in JIR, we first formalize the JIR task as a partially observable markov decision process, readily transferrable into a JIR agent development pipeline (Section 2), and propose the evaluation framework for JIR systems (Section 4). Moreover, we introduce JIR-ARENA, the first multimodal JIR benchmark with real, diverse, extensive and challenging information-seeking scenarios to assess the *precision*, *recall*, *timeliness* and *relevance* of information recommendations from JIR systems (Section 3).

The main motivation behind JIR-ARENA is to create the first comprehensive benchmark that reflects diverse, realistic human information needs in learning-based settings, enabling quantitative evaluation of JIR algorithms. To fully assess a JIR system's ability to infer user needs and deliver relevant information, JIR-ARENA's construction includes two stages: simulating user queries and completing JIR instances via information retrieval. This process is challenging due to: *i)* the subjectivity of information need annotations—whether generated by individual annotators or large foundation models—fails to capture the distribution of user information needs, making it unsuitable as a benchmark standard; and *ii)* ensuring reproducibility of evaluation results is difficult due to uncontrollable

variables in the JIR instance generation pipeline, such as the ever-growing web-scale document repositories or the evolving capabilities of content-generation models deployed. To address these, we propose: *i)* employing *multi-entity, multi-turn* simulations to approximate the user information need distribution. Specifically, for a given video, multiple large foundation models and human annotators collaboratively propose information needs to ensure a more exhaustive set of needs, with an additional voting step that determines the likelihood of each need being expressed; *ii)* defining the quality of a single JIR instance based on its performance in retrieving information from *static knowledge bases* to satisfy the information need, thereby eliminating the impact of the varied retrieval repositories or content-generation models of each JIR system to be evaluated; and *iii)* incorporating *multi-entity verification* at critical steps to ensure the objectivity and generality of the benchmark dataset. **Overall**, JIR-ARENA encompasses 34 multimedia scenes totaling 831 minutes, focusing on information-intensive scenarios such as lectures and conference talks. Lecture topics span diverse domains, including computer science, neuroscience, finance, mathematics, and chemistry, while conference talks cover subfields of computer science such as AI, programming, cybersecurity, and educational technologies.

Leveraging JIR-ARENA, we conduct the first baseline study of JIR systems and establish baseline results for the JIR task to facilitate further research in this new area. Specifically, we develop a prototypical JIR system with streaming inputs as the JIR-ARENA's baselines (Section 5), with the following findings (Section 6): *i)* JIR systems based on large foundation models can reasonably simulate user information needs (achieving good precision) but often fail to comprehensively cover all user needs (resulting in low recall), especially even for information needs that most users urgently expect the system to cover (needs with higher information need likelihood), recall remains far from satisfactory; and *ii)* effective content retrieval is crucial to satisfying user information needs, yet the commonly used retrieval models underperform, indicating plenty of room for future research.

In summary, our contributions are as follows:

- We formalize the JIR task as a POMDP and establish an evaluation methodology for JIR with multiple new measures, laying out a formal foundation for studying JIR.
- We create JIR-ARENA, the first comprehensive multimodal benchmark dataset for evaluating JIR systems' utility, featuring diverse, extensive, and representative testing scenarios.
- As the first baseline study of JIR systems and a way to support future research, we implement an extensible baseline JIR system, obtain initial empirical performance results of baseline algorithms, and provide insights for future research.
- We facilitate future research in this new topic by fully releasing our code and data.

## 2 JIR TASK FORMULATION

Machines discretize the physical world for processing, allowing us to formalize the JIR task as a Partially Observable Markov Decision Process (POMDP) with discrete time framework: $\langle O, S, A, T, R, \Omega, \lambda \rangle$, where the time unit is the machine's discretization scale.

For JIR, an observation $o \in O$ represents the current multimodal context $c \in C$, the observable aspects of the user's evolving persona $u \in U$, and the knowledge bases $K$ to help issue accurate and useful recommendations. Specifically, $c$ includes any visual, auditory, and sensory data consistent with the user's perspective, and $u$ covers the user's *background* $b \in B$ (e.g., experiences, knowledge, traits, etc.) and their *goals* $g \in G$. In other words, the tuple $o = \langle c, \langle b, g \rangle, K \rangle$ forms the typical input of a JIR system. Correspondingly, a state $s \in S$ represents the complete environment, including any world changes, the underlying user persona evolution, and accessible/inaccessible knowledge bases.

An action $a \in A$ involves initiating/terminating one/more JIR instance(s), or remaining passive when no information needs to be provided. To be specific, each JIR instance $\iota$'s minimal required field is $\iota = \langle t_s, q, Ref, t_e \rangle$, where $t_s/t_e$ is the start/end time, $q$ is the user information need in the form of query, and $Ref$ denotes the reference list for making a recommendation to satisfy the need. Formally, the JIR system's output actions include: *i)* $a = \alpha(\iota)$, which creates a $\iota$ with the value $t_s$, $q$, and $Ref$ set, *ii)* $a = \omega(\iota)$, which assigns the $t_e$ to the $\iota$, or iii) $a = \Delta$, which tells the system to stay idle.

The transition function $T : S \times A \rightarrow S$ models state changes based on the current state and JIR system actions. $R : S \times A \rightarrow \mathbb{R}$ is the reward function that determines the utility to the user when the JIR system takes an action in the current state, which is measured by the timeliness and situational relevance of the JIR instances established/canceled by the action. $\Omega$ reflects the initial observation (e.g., information-seeking scenarios, the user profile pool, and knowledge bases) distribution. The discounting factor $\lambda$ is typically 1 or close to it, as most JIRs have immediate, independent impacts.

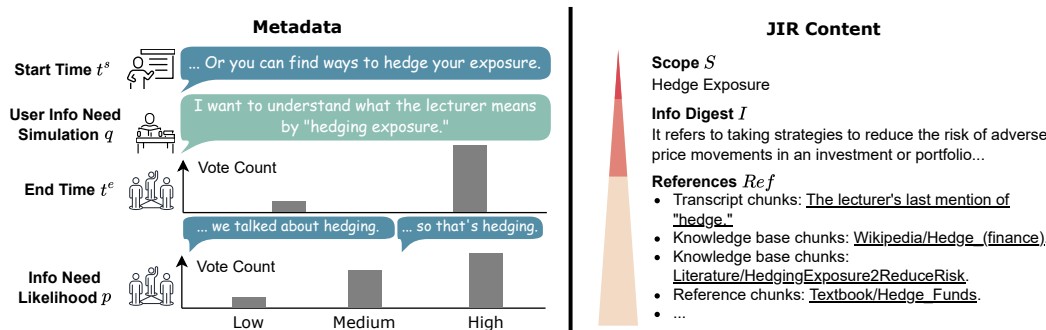

Figure 2: A usable form of a JIR instance, with $\langle t_s, q, Ref, t_e \rangle$ to be the minimally required field for measuring the JIR's quality, $p$ for characterizing the need distribution, and $\langle S, I \rangle$ for system display.

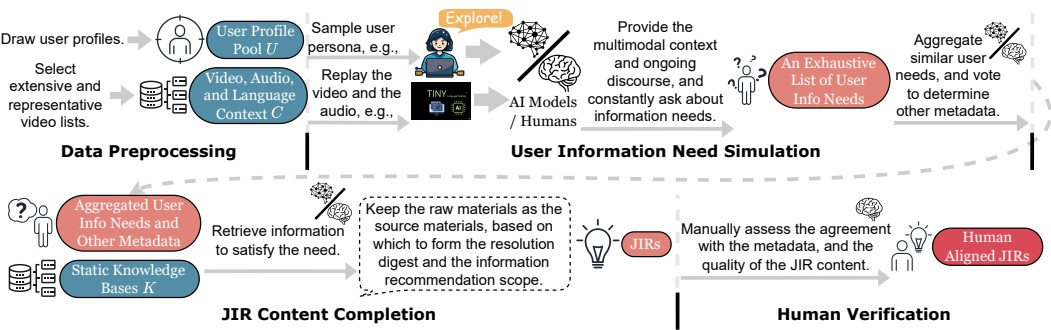

Figure 3: Our pipeline of constructing the JIR-ARENA benchmark dataset.

To solve JIR tasks, a JIR system must design a decision policy $\pi(a_t | \{o_0, o_1, ..., o_t\})$ that maximizes the expected discounted return $\mathbb{E}_{o \sim \Omega}[\sum_{t=0}^{T} \lambda^t R_{t+1} | o_0 = o]$, essentially aiming to provide the right information at the right time.

This paper focuses on *i)* defining the JIR POMDP components, *ii)* standardizing evaluation, and *iii)* establishing JIR-ARENA and a baseline JIR system $\pi_{baseline}$.

## 3 JIR-ARENA BENCHMARK DATASET CURATION

In this section, we describe the benchmark dataset construction methodology. Due to the unique characteristics of the JIR task, building this dataset is more complex than for standard retrieval or recommendation tasks. The dataset includes multiple independent scenes, each with a set of JIR instances. As shown in Figure 2, each instance has two parts: metadata and JIR content. Metadata mainly includes the simulated user information need $q$, start time $t_s$, end time $t_e$, and the likelihood distribution $p$. JIR content covers the scope $S$ and information digest $I$ for system display. References $Ref$ provide source materials to ensure accuracy. The minimal required attributes for evaluation in JIR-ARENA are $t_s$, $q$, $Ref$, and $t_e$. Figure 3 outlines the dataset construction process, with details in the following sections.

### 3.1 STAGE 1: DATA PREPROCESSING

In this stage, we enrich the user profile pool $U$, compile testing scenes $C$, and decide the static knowledge bases $K$. For the user profile pool $U$, human annotators are required to be domain experts likely to have firsthand experience with the tested scenarios in JIR-ARENA. For large foundation models, a role-playing prompt simulates the models as active participants in the given scenarios, instructing the models to request necessary information when needed. Regarding testing scenarios $C$, we focus on two categories characterized by dense information needs: academic lectures and conference talks. Real-world instances of these scenarios are sourced from YOUTUBE, including recorded lectures, talks, and virtual paper explanation videos. The video list is curated by identifying high-subscriber YOUTUBE playlists spanning diverse domains, such as computer science, neuroscience, finance, mathematics, and chemistry for lectures, as well as subtopics within computer science conferences, such as AI and information security. From each playlist, we select the most-viewed videos to ensure scene representativeness. For knowledge bases, we use the Wikipedia

dump (Petroni et al., 2021) covering relevant annotations of academic concepts, scholars, and related books, etc., as the basic source. Additionally, for lectures, we add the textbook where applicable. For conference talks, we incorporate arXiv paper titles, abstracts (Cornell University, 2022), and, if any, the full text of the papers being discussed during the talks. Metadata for these YOUTUBE videos, textbook, and papers is summarized in Appendix B.

### 3.2 STAGE 2: USER INFORMATION NEED SIMULATION

In this stage, we employ both large foundation models and human annotators to simulate user information needs and estimate the need's distribution. It is important to note that information needs from any single individual are highly subjective and tightly coupled with a user's persona, including their background and goals. For instance, students from computer science and neuroscience disciplines may require different supplementary information during the same biologically-plausible AI talk. We model user information needs as a time-dependent random variable $Q$, where at each time point $t$, $Q$ has a probability of taking a specific value $q$. The measurement objective of JIR-ARENA, is thus defined as the ability of any JIR system to accurately characterize the probability distribution of $Q$, and to address the queries. To this end, we *i)* curate *exhaustive* user information needs by combining multi-entity simulations while ensuring their reasonableness, deduplicating them based on temporal overlap and semantic similarity, and voting on their likelihood $p = P_{t_s}(Q = q)$ to reflect the probability that $q$ is expressed by all users. The domain completeness of $Q$ ensured by the exhaustive need simulation, and the likelihood estimation derived from multi-entity voting, justify JIR-ARENA as a reasonable and valid benchmark for evaluating JIR system performance.

To be specific, we utilize two types of modules to simulate user information needs: large language models (LLMs) and human annotators. The LLMs include GPT-4O (OpenAI et al., 2024) and DEEPSEEK-V3 (671B) (DeepSeek-AI et al., 2024). For LLM-based simulations, the process involves using WHISPER-LARGE-V3 (Radford et al., 2022) to transcribe audio content and the visual language model (VLM) NVILA-8B (Liu et al., 2024) to narrate video content, resulting in two input modalities: transcription-only (audio-based) and transcription+narrative (audio+video-based). A detailed rationale and validation of using VLM-generated narrations are provided in Appendix C. Each modality is segmented into units of five sentences, with the preceding context summarized as background and the current unit designated as ongoing discourse. Then LLMs are prompted to identify user information needs for the ongoing discourse based on the background. For every scene, this process yields two simulated user information need lists corresponding to the two input modalities for each LLM. Human annotators, on the other hand, are instructed to pause videos whenever they have questions, record the questions alongside timestamps, and provide their own simulation lists for each scene. All user information needs identified within a scene are aggregated, deduplicated based on temporal overlap and semantic similarity (with a similarity threshold of 0.75 as determined by SENTENCE-TRANSFORMERS/ALL-MINILM-L6-V2 (Reimers and Gurevych, 2019), prioritizing human annotations over LLM-generated ones), and finalized. Subsequently, GPT-4O and DEEPSEEK-V3 (671B) are tasked with voting on the likelihood of each element in the exhaustive information need list being raised by human audiences, yielding the likelihood $p$. The LLMs also vote to determine the end time $t_e$ for each JIR instance, given the subjectivity of end-time judgments. All LLM prompts are detailed in Appendix E.

Notably for human annotators, we conduct an quantitative experiment with four human annotators on nine scenes spanning diverse topics (221 minutes in sum). Analysis reveals *i)* significant variance in user information needs across annotators and *ii)* the ability of LLM simulations to cover these high-variance needs. Consequently, for the remaining videos, exhaustive user information needs are generated solely using LLMs. Detailed analysis are given in Appendix F.

### 3.3 STAGE 3: JIR CONTENT COMPLETION

In the JIR instance completion stage, we implement a three-layer information retrieval pipeline comprising a classical information retrieval model, a quality check using LLMs, and a human quality check to ensure that the reference list for each JIR instance is useful for addressing the corresponding information query. The relevance score of a document is determined based on the number of retrieval layers it passes (e.g., documents not retrieved by the information retrieval model are assigned a relevance score of 0, while those passing the human quality check receive the highest score 3).

The classical information retrieval layer narrows the search space in the document repository. We employ the hybrid model from the PYSERINI library (Lin et al., 2021a) for document indexing and retrieval, utilizing BM25 ranking (Robertson et al., 1995) for sparse indexing and CASTORINI/TCT_COLBERT-V2-HNP-MSMARCO (Lin et al., 2021b) as the dense indexing model.

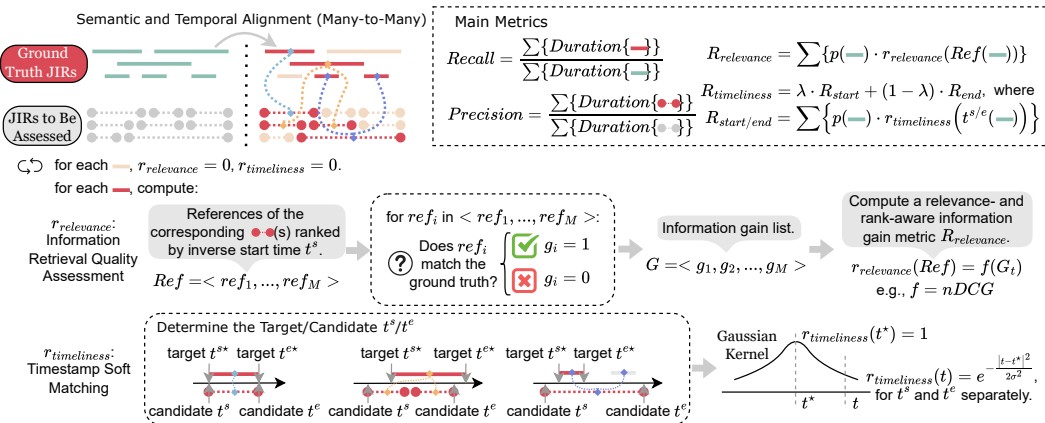

Figure 4: Evaluation metrics of the JIR systems (top-right): Two global metrics, $Recall$ and $Precision$, based on alignment of ground truth and the candidate answer (top-left), and $R_{relevance}$ and $R_{timeliness}$, aggregated from the relevance and timeliness scores of each JIR instance (bottom).

Document granularity is designed as: Wikipedia and arXiv datasets are chunked and indexed at the entry level, while textbooks and papers discussed are segmented into their smallest meaningful units (e.g., subsections, figures, or tables) for indexing. At this stage, the retrieval model returns the top 10 most relevant documents from each knowledge base (e.g., Wikipedia, textbooks) for a given query.

The LLM quality check layer leverages multiple LLMs to filter out irrelevant documents through contextual reasoning. Specifically, for each query and the associated documents retrieved, GPT-4O and DEEPSEEK-V3 (671B) are prompted to assess whether a document could provide useful information for answering the query. If deemed relevant, the LLMs extract the helpful portions of the document. The prompts used in this process are detailed in Appendix G.

Finally, the human quality check layer determines the final references. For each query, if all LLMs agree that a document contains information relevant to answering the query, the human annotators are tasked with making a final decision based on the query's context. The results of these three retrieval layers are then integrated to finalize the references and to assign the hierarchical relevance scores.

### 3.4 STAGE 4: HUMAN VERIFICATION

To verify the quality of the curated JIR instances, we perform a manual verification. The four annotators are tasked with judging the outputs of Stage 2. Note that human verification for Stage 3 has been done with its human quality check layer. For Stage 2, two needs are randomly sampled from each of the videos and assigned to each respective annotator. This results in 272 needs judged. The annotators are instructed to rate the generated *Reason*, *Need*, and *Question* on the following scale: **(0)** Disagree: The generated content is implausible; **(1)** Slightly Disagree: The generated content is somewhat implausible; **(2)** Neutral: The generated content is possibly plausible; **(3)** Slightly Agree: The generated content is somewhat plausible; **(4)** Agree: the generated content is plausible.

The annotators judge the *Need* and *Question* conditioned on the generated *Reason*. The average *Reason* score is 3.26, the average *Need* score is 3.774, and the average *Question* score is 3.801. This indicates that the generated reasons are generally plausible, and that the foundation models are able to generate plausible needs and questions given a reason. The statistics of JIR-ARENA is in Table 3.

## 4 EVALUATION METRICS OF JIR SYSTEMS

Since JIR is not well studied, evaluating a JIR system remains an open problem. We propose that JIR evaluation should focus on three aspects: *i)* accurately inferring user information needs for reliable retrieval and recommendation tasks; *ii)* avoiding redundant recommendations to reduce cognitive load; and *iii)* ensuring that each JIR instance is contextually relevant, satisfies the information need, and is delivered in a timely manner. Guided by these principles, we propose a set of *scene-wise* metrics to evaluate the JIR systems (Figure 4). $Recall$ and $Precision$ are designed to address the first two aspects, measuring the coverage of ground truth JIR instances (JIRs) and the effectiveness of JIRs generated by the system, respectively. For the third aspect, we introduce $R_{relevance}$ and $R_{timeliness}$, which measure the relevance and timeliness of JIR instances. The calculation methods for these metrics are detailed below:

*Recall* **and** *Precision*: To compute those, we first perform semantic and temporal alignment between the sets of ground truth JIRs and candidate JIRs. For each ground truth JIR, we identify candidate JIRs with overlapping time intervals and evaluate their semantic similarity. A match is recorded if the semantic content is sufficiently similar (cosine similarity by SENTENCE-TRANSFORMERS/ALL-MINILM-L6-V2 is higher than 0.55), with the matching process flexibly defined as many-to-many to account for the decomposability of user information needs. We provide the matching algorithm details in Appendix H. *Recall* is then computed as the proportion of ground truth JIRs that are matched, reflecting the system's ability to infer user needs. *Precision* is calculated as the proportion of candidate JIRs that are matched, assessing the system's efficiency in generating effective JIRs.

$R_{relevance}$: This metric evaluates whether the reference lists returned by the JIR system satisfy the user's information need. It aggregates individual relevance scores ($r_{relevance}$) for ground truth JIRs, which are computed using traditional information retrieval metrics. Specifically, for each matched ground truth JIR, the JIR system will provide a ranked list of candidate reference documents. If there are more than one matched candidate JIRs, the reference list is ordered by their start times in descending order, as users pay more attention to the newly appeared JIR. Using relevance scores pre-determined in JIR-ARENA, we calculate an information gain list $G$ for the candidate reference list and apply nDCG[1] (Järvelin and Kekäläinen, 2002) to derive $r_{relevance}$. For unmatched ground truth JIRs, $r_{relevance} = 0$. The global $R_{relevance}$ is a weighted sum of $r_{relevance}$, where weights are determined by normalizing the likelihood $p$ of each JIR within a given scene.

$R_{timeliness}$: This metric assesses the temporal accuracy of the JIR system's recommendations. It balances the alignment of start times ($R_{start}$) and end times ($R_{end}$) through a weighting factor $\lambda$, with each component aggregated across the scene. For individual temporal alignment $R_{timeliness}$, we apply a Gaussian-like kernel to compute a fuzzy match between the ground truth and candidate time points. Specifically, for each matched ground truth JIR, we determine the earliest start time and latest end time among the corresponding candidate JIRs as the timestamp to be evaluated. The timeliness score is then computed as $r_{timeliness} = e^{-\frac{|t-t^\star|^2}{2\sigma^2}}$, where $t^\star$ is the ground truth time, t is the candidate time, and $\sigma$ (set to 3 seconds) represents the tolerance for timing errors. This kernel ensures a smooth decay in score as temporal discrepancies increase. For unmatched ground truth JIRs, $r_{timeliness} = 0$. The global $R_{timeliness}$ is computed as a weighted sum of $r_{timeliness}$, using the same weighting scheme as $R_{relevance}$.

## 5 AN EXTENSIBLE BASELINE JIR SYSTEM

We implement an extensible baseline JIR system that separates the problem into two functional stages: information-need generation and evidence retrieval, allowing controlled comparison across models and enabling oracle substitution for diagnosing component-level limitations.

The information-need generator processes transcript chunks and produces candidate queries; each generated query is paired with a start/end timestamp aligned to transcript segments and submitted to the retrieval module. The generator can be instantiated with any text or multimodal model; in our experiments, we include a broad family of open-source and proprietary models to ensure coverage of model scales and modalities (Anthropic, 2024; OpenAI et al., 2024; Team et al., 2023; xAI, 2025; François et al., 2024; Yang et al., 2025; Microsoft et al., 2025; Grattafiori et al., 2024; Cohere et al., 2025; DeepSeek-AI et al., 2024).

The retrieval module queries a set of static external knowledge sources, including a Wikipedia dump, arXiv papers, and other reference materials (detailed in Appendix B). We implement three retrieval backends representative of commonly used architectures: a sparse BM25 retriever (OpenSearch (OpenSearch, 2025)), a dense retriever (ALL-MINILM-L6-V2 (Reimers and Gurevych, 2019)), and a reranked pipeline where dense retrieval outputs are refined by a cross-encoder reranker (MS-MARCO-MINILM-L-6-V2). These interchangeable backends allow controlled experiments on how retrieval architecture affects downstream evaluation without modifying the generator.

To support component-wise analysis, the baseline provides an oracle mode in which the generator is replaced by ground-truth information needs from the benchmark while keeping the retrieval and evaluation pipeline unchanged. This design isolates the retrieval stage and allows to measure the upper limits achievable when the generation stage is idealized. Evaluation follows the many-to-many

---

[1]nDCG (normalized Discounted Cumulative Gain) measures how well the predicted ranking of search results aligns with the ideal ranking, giving higher importance to relevant items appearing earlier in the list.

Table 1: Performance of baseline JIR systems on JIR-ARENA. We have the **Generative Model** to predict information needs, and the **Retrieval Model** to retrieve relevant documents. $R_{relevance}$ shown under different retrieval model settings: **O** = OpenSearch (BM25), **D** = Dense Retriever, **R** = Dense Retriever + Reranker.

| Generative Model | Recall | Precision | $R_{relevance}$ (O/D/R) | $R_{start}$ | $R_{end}$ | $R_{timeliness}$ |
|---|---|---|---|---|---|---|
| **Upper Bound (Full Oracle)** | | | | | | |
| ORACLE | 1.000 | 1.000 | 1.000/1.000/1.000 | 1.000 | 1.000 | 1.000 |
| **Oracle Generator** | | | | | | |
| ORACLE | 1.000 | 1.000 | 0.215/0.195/0.274 | 1.000 | 1.000 | 1.000 |
| **Closed-Source Generators** | | | | | | |
| CLAUDE-3.7 | 0.327 | 0.697 | 0.048/0.056/0.068 | 0.173 | 0.116 | 0.167 |
| GPT-4O | 0.406 | 0.694 | 0.056/**0.069**/0.081 | 0.254 | 0.125 | 0.241 |
| GEMINI-2.0-FLASH | **0.429** | 0.694 | 0.057/0.067/**0.084** | **0.293** | **0.137** | **0.277** |
| GROK-3 | 0.403 | **0.723** | **0.060**/0.067/0.079 | 0.246 | 0.136 | 0.235 |
| MISTRAL-MEDIUM | 0.365 | 0.694 | 0.045/0.058/0.070 | 0.190 | 0.124 | 0.183 |
| **Closed-Source Generators (Multimodal)** | | | | | | |
| CLAUDE-3.7 | 0.302 | 0.723 | 0.050/0.053/0.065 | 0.140 | 0.103 | 0.137 |
| GPT-4O | 0.378 | **0.781** | 0.058/0.068/0.078 | 0.195 | 0.106 | 0.186 |
| GEMINI-2.0-FLASH | **0.412** | 0.716 | **0.060/0.069/0.083** | **0.264** | **0.121** | **0.250** |
| GROK-3 | 0.344 | 0.760 | 0.058/0.065/0.075 | 0.183 | 0.114 | 0.176 |
| MISTRAL-MEDIUM | 0.325 | 0.736 | 0.049/0.059/0.067 | 0.161 | 0.081 | 0.153 |
| **Open-Source Generators** | | | | | | |
| QWEN3 (4B) | 0.270 | 0.716 | 0.042/0.048/0.055 | 0.134 | 0.084 | 0.129 |
| PHI-4 (14B) | 0.329 | 0.688 | 0.044/0.055/0.063 | 0.179 | 0.097 | 0.171 |
| LLAMA 3.3 (70B) | 0.377 | 0.684 | 0.052/0.064/0.074 | 0.221 | 0.106 | 0.209 |
| COHERE (111B) | 0.390 | **0.719** | 0.051/**0.064**/0.077 | 0.229 | **0.138** | 0.220 |
| DEEPSEEK-V3 (671B) | **0.396** | 0.716 | **0.053**/0.064/**0.078** | **0.244** | 0.121 | **0.232** |
| **Open-Source Generators (Multimodal)** | | | | | | |
| QWEN3 (4B) | 0.230 | 0.746 | 0.044/0.047/0.053 | 0.095 | 0.057 | 0.092 |
| PHI-4 (14B) | 0.293 | 0.742 | 0.047/0.052/0.060 | 0.136 | 0.076 | 0.130 |
| LLAMA 3.3 (70B) | 0.288 | 0.757 | 0.047/0.057/0.064 | 0.135 | 0.075 | 0.129 |
| COHERE (111B) | 0.318 | **0.778** | 0.051/0.056/0.066 | 0.150 | 0.085 | 0.143 |
| DEEPSEEK-V3 (671B) | **0.355** | 0.723 | **0.053/0.062/0.074** | **0.187** | **0.097** | **0.178** |

matching formulation defined in Section 4, including semantic similarity thresholds and temporal-overlap constraints between predicted and ground-truth needs. All experimental components are consistent across models to ensure comparability. Details and prompts are in Appendix I.

## 6 RESULTS

Across the three core components of JIR—inferring user information needs, retrieving documents that can address those needs, and delivering timely recommendations—our results reveal clear gaps between current system behavior and what real users require. Generative models miss many high-likelihood needs, retrieval models struggle without sufficient contextual grounding, and timing remains imperfect. Below, we break down these findings in detail.

### 6.1 BENCHMARK-LEVEL FINDINGS

**Both the need generator and the retrieval module are bottlenecks.** The end-to-end gap is caused by both stages: generators frequently fail to propose recoverable information needs, and retrieval cannot fully recover even when given oracle queries. Oracle substitution—replacing model-generated queries with ground-truth needs while keeping retrieval fixed—raises recall from 0.347 to 1.00 and substantially improves relevance. However, even under oracle generation, the retrieval module achieves only modest relevance ($R_{relevance} \approx 0.21$–0.27 across backends), indicating that retrieval quality also significantly constrains the achievable upper bound. These observations confirm a *dual bottleneck* at the need generator and the evidence retriever. Full analysis appears in Appendix J.

**The generator's recall shortfall is dominated by semantic mismatch.** Error decomposition shows that $\approx 62.91\%$ of failures arise because no generated query is semantically close enough to the ground-truth need within its temporal window. These deficits persist even for high-likelihood needs (i.e., needs multiple annotators/LLMs consider highly probable for real users), indicating that current LLMs struggle to faithfully simulate users' *cognitive state* and *anticipatory reasoning*. Temporal misalignment accounts for only a small minority of errors. This suggests that future improvements to JIR systems may benefit more from improving the *semantic grounding and alignment* of generated information needs rather than increasing generation volume.

**Retrieval failures stem largely from underspecified or context-poor queries.** When generators operate on small transcript chunks or receive only shallow event summaries, they often produce vague or underspecified queries—e.g., *"What is the method here?"*, *"Explain this concept more,"* or *"Why does this matter?"*. Such queries lack the domain tokens, entities, or lexical cues required for effective retrieval. Retrieval backends then surface documents that are topically adjacent but not specifically relevant, lowering both $R_{relevance}$ and $R_{timeliness}$.

Ablations varying chunk size, temporal position, and need density (see Appendix J) reveal monotonic improvements when generators receive richer query-relevant context, reinforcing that retrieval quality is tightly coupled to the richness of generated queries.

**Need types impose distinct, structural challenges.** Recoverability varies widely across the benchmark's 11 need types. Needs requiring **visual grounding** (e.g., references to diagrams, gestures, or slides) and those requiring **implicit discourse inference** exhibit the lowest recall and relevance. For visually grounded needs, recall drops to nearly half that of concept-level needs, revealing a structural mismatch: text-only retrieval cannot resolve visually anchored queries even when generators capture them semantically. Similarly, "missing context," "ambiguous language," or "instructional/action" needs degrade sharply due to generator ambiguity and retrieval sparsity. Detailed per-type metrics, type-specific challenge scores, and decompositions are provided in Appendix K.

## 6.2 MODEL-SPECIFIC PATTERNS

**Power law: larger models generate broader, more recoverable needs.** Model scale correlates positively with recall and the number of matched candidate needs: larger models propose more semantically aligned information needs and cover more ground-truth instances. Smaller models appear more "precise" only because they generate substantially fewer queries, reducing false positives but also limiting coverage. This yields superficially higher precision but much lower recoverability.

A power-law fit between model size and matched candidate count (Appendix L and M) suggests consistent scaling behavior across families.

**Multimodal signals help selectively but introduce large temporal ambiguity.** Multimodal models outperform their text-only counterparts on visually grounded needs, demonstrating that visual features supply meaningful cues unavailable from transcripts alone. However, these models also suffer substantial temporal misalignment: $\approx 86.8\%$ of the multimodal performance gap is attributable to mismatched timestamps. Models often anchor on visually salient frames—slides, gestures, transitions—that are *not* aligned with the annotated need start/end times, producing off-by-span predictions that cannot satisfy temporal matching constraints.

As a result, multimodal models exhibit higher precision but lower recall overall. Examples and cluster-level analyses are presented in Appendix N.

## 6.3 EVALUATION AND ROBUSTNESS

We conduct extensive robustness checks across matching thresholds, temporal kernels, retrieval backends, and evaluation parameters. Three observations are consistent:

1. **Model ordering is stable** across wide parameter sweeps (Spearman $\approx 0.956$).

2. **Retrieval backend ranking is consistent** (Reranked > Dense > BM25), though absolute values differ.

3. **Human-validation subsample is representative**, with semantic similarity between subsample and full set exceeding 0.9.

Full robustness studies and sweeps are reported in Appendix O.

Table 2: Comparison of existing datasets and systems relevant to JIR.

| Work/Dataset | Primary Task | Input Data Modality | Context Type | Proactivity Model | Evaluation | Key Limitations for JIR Evaluation |
|---|---|---|---|---|---|---|
| JITIR Systems (Rhodes, 2000) | Associative Information Retrieval | Structured & unstructured textual input | Local, textual content | Continuous monitoring of text buffer | User satisfaction, qualitative feedback | Conceptual framework; not a reusable benchmark. |
| TREC CS Track (Dean-Hall et al., 2014) | Point-of-Interest (POI) Recommendation | User profiles, POI descriptions | Offline, static scenarios, e.g., trip type | Triggered by a profile/context pair for planning | Ranking (nDCG), no timeliness measurement | Context is static and given, not dynamic or implicitly sensed. Task is ad hoc planning, not real-time assistance. |
| ProCIS (Samarinas and Zamani, 2024b) | Proactive Document Retrieval | Text-based multi-party conversations | Social, conversational, linguistic | Inferred from textual dialogue flow | Proactive Ranking (npDCG) | Context is purely linguistic and social. Does not model an individual's multimodal experience during a task. |
| **JIR-ARENA (Our Work)** | **Just-in-Time Information Recommendation** | **Multimodal video/audio streams, transcripts** | **Dynamic, implicit, real-time event-driven, multimodal** | **Triggered by inferred needs within a user's ongoing activity** | **Need Inference (Precision/Recall), Recommendation Relevance, Timeliness, and Non-Distraction** | **Designed specifically for the JIR task.** |

## 7 RELATED WORK

**Foundations in Proactive and Context-Aware Systems:** The vision for JIR builds upon several foundational research areas but addresses a unique, unbenchmarked problem. Just-in-Time Information Retrieval (JITIR) introduced proactive, context-sensitive software agents designed to reduce the cognitive effort of information access (Rhodes, 2000). Parallelly, Context-Aware Recommender Systems (CARS) developed models for incorporating situational context (Adomavicius et al., 2011; Abbas, 2015). However, the primary task in CARS is preference prediction (e.g., predicting a movie rating), which is fundamentally different from the JIR task of fulfilling a transient information gap (e.g., defining an unknown term during a lecture). Consequently, established recommender system benchmarks are not appropriate for evaluating JIR systems.

**Modern Proactive Systems and Multimodal Sensing:** Recent proactive information access research has concentrated on conversational settings. The ProCIS benchmark (Samarinas and Zamani, 2024a) evaluates proactive document retrieval in multi-party, text-based conversations, focusing solely on linguistic context. In contrast, JIR emphasizes an individual's dynamic, multimodal context during non-conversational tasks. While multimodal sensing is well-studied in HCI, these efforts generally stop at user state inference, not connecting sensing to information retrieval.

**The Benchmark Gap and JIR-ARENA's Contribution:** No current benchmark fills the evaluation gap of the JIR task. For example, the TREC Contextual Suggestion Track (Dean-Hall et al., 2014) evaluated recommendations for static, pre-defined scenarios (e.g., trip planning) without real-time assistance. JIR-ARENA is the first benchmark to evaluate the full JIR challenge: inferring implicit information needs from dynamic, multimodal context to deliver timely, relevant, and unobtrusive recommendations. Table 2 summarizes this, highlighting our unique contribution.

## 8 CONCLUSIONS

This work introduces Just-in-time Information Recommendation (JIR) and highlights its potential as a transformative information service with ample opportunities for new AI research. Different from search engines and current recommender systems, a JIR system focuses on recommending relevant information to users at the most opportune moments, in a minimally intrusive manner, to address their information gaps and support decision-making. With technological advancements increasingly enabling the feasibility of JIR systems, this helpful service is poised to reshape how individuals interact with information, unlocking commercial opportunities and driving new research directions.

To facilitate and accelerate research in JIR, we formalize the JIR task as a POMDP, propose and define the first comprehensive set of evaluation metrics tailored to the expectations of well-functioning JIR systems, and create JIR-ARENA, the first comprehensive multimodal JIR benchmark dataset, which encompasses diverse, representative, and challenging scenes necessitating JIR services. Additionally, we develop a prototype JIR system and conduct the first baseline study of JIR systems with error analysis on JIR-ARENA, and propose avenues for future system improvements. To facilitate further research in this new area, we will make all associated code and data fully accessible. Moving forward, we anticipate increased exploration of JIR application scenes and advancements in developing state-of-the-art JIR systems regarding the basic performance and JIR personalization.

## 9 ETHICS STATEMENT

The authors of this paper had read the Code of Ethics, and we acknowledge that this submission adheres to the code.

## 10 REPRODUCIBILITY STATEMENT

Towards the efforts of making this work reproducible, we include our code and constructed benchmark dataset in the supplementary materials. Upon acceptance, we will release the code and dataset to the public. For our baseline experiments, we used both open and closed source language models. Reproducing the results from the open-source models should be straightforward, as we include the full description of the experiments as well as the system prompts (in the appendix). We acknowledge that the use of closed-source language models may make it difficult to reproduce the experimental results, as these models may change or non-deterministically vary their output. But given the system prompts along with our experimental descriptions, it should be possible to reproduce our closed-source results.

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

## A    THE USE OF LARGE LANGUAGE MODELS (LLMs)

We used an LLM to assist with language polishing. Specifically, the LLM was employed to improve clarity, grammar, and readability of the manuscript, while all technical content and contributions (especially the entire abstract, introduction, and conclusion section) were completely written and verified by the authors.

## B    METADATA OF PRE-PROCESSED VIDEOS, TEXTBOOK AND PAPERS

### B.1    VIDEO METADATA

- **Title**: Elements and atoms — Atoms, compounds, and ions — Chemistry — Khan Academy
  **Duration**: 13m 9s
  **Channel**: Khan Academy
  **Views**: 4,192,084
  **YouTube ID**: IFKnq9QM6_A

- **Title**: Basic trigonometry — Basic trigonometry — Trigonometry — Khan Academy
  **Duration**: 9m 17s
  **Channel**: Khan Academy
  **Views**: 4,131,663
  **YouTube ID**: Jsiy4TxgIME

- **Title**: Introduction to cellular respiration — Cellular respiration — Biology — Khan Academy
  **Duration**: 14m 19s
  **Channel**: Khan Academy
  **Views**: 3,249,406
  **YouTube ID**: 2f7YwCtHcgk

- **Title**: The World's Best Mathematician (*) - Numberphile
  **Duration**: 10m 57s
  **Channel**: Numberphile
  **Views**: 7,575,385
  **YouTube ID**: MXJ-zpJeY3E

- **Title**: 158,962,555,217,826,360,000 (Enigma Machine) - Numberphile
  **Duration**: 11m 51s
  **Channel**: Numberphile
  **Views**: 6,318,079
  **YouTube ID**: G2_Q9FoD-oQ

- **Title**: Problems with Zero - Numberphile
  **Duration**: 13m 0s
  **Channel**: Numberphile
  **Views**: 5,758,032
  **YouTube ID**: BRRolKTlF6Q

- **Title**: Perfect Shapes in Higher Dimensions - Numberphile
  **Duration**: 26m 19s
  **Channel**: Numberphile
  **Views**: 5,326,862
  **YouTube ID**: 2s4TqVAbfz4

- **Title**: e (Euler's Number) - Numberphile
  **Duration**: 10m 42s
  **Channel**: Numberphile
  **Views**: 4,811,422
  **YouTube ID**: AuA2EAgAegE

- **Title**: The hardest problem on the hardest test
  **Duration**: 11m 15s
  **Channel**: 3Blue1Brown
  **Views**: 15,931,563
  **YouTube ID**: OkmNXy7er84

- **Title**: Solving Wordle using information theory
  **Duration**: 30m 38s
  **Channel**: 3Blue1Brown
  **Views**: 10,981,963
  **YouTube ID**: v68zYyaEmEA

- **Title**: But why is a sphere's surface area four times its shadow?
  **Duration**: 15m 50s
  **Channel**: 3Blue1Brown
  **Views**: 8,278,649
  **YouTube ID**: GNcFjFmqEc8

- **Title**: Gradient descent, how neural networks learn — DL2
  **Duration**: 20m 33s
  **Channel**: 3Blue1Brown
  **Views**: 7,732,696
  **YouTube ID**: IHZwWFHWa-w

- **Title**: 1. Introduction, Financial Terms and Concepts
  **Duration**: 60m 30s
  **Channel**: MIT OpenCourseWare
  **Views**: 7,584,607
  **YouTube ID**: wvXDB9dMdEo

- **Title**: Lecture 1: Algorithmic Thinking, Peak Finding
  **Duration**: 53m 21s
  **Channel**: MIT OpenCourseWare
  **Views**: 5,733,981
  **YouTube ID**: HtSuA80QTyo

- **Title**: Lecture 25 — Probabilistic Topic Models Expectation Maximization Algorithm - Part 3 — UIUC
  **Duration**: 6m 25s
  **Channel**: Artificial Intelligence - All in One
  **Views**: 1,574
  **YouTube ID**: ghZRzOb␣bZo

- **Title**: Lecture 42 — Text Categorization Evaluation - Part 2 — UIUC
  **Duration**: 10m 52s
  **Channel**: Artificial Intelligence - All in One
  **Views**: 520
  **YouTube ID**: SoZStBaLbws

- **Title**: NeurIPS 2018: Keynote on What Bodies Think About
  **Duration**: 56m 51s
  **Channel**: ICML IJCAI ECAI 2018 Conference Videos
  **Views**: 1,600
  **YouTube ID**: Hpw31YTq␣yI

- **Title**: Are Time Series Foundation Models Ready to Revolutionize Predictive Building Analytics?
  **Duration**: 15m 44s
  **Channel**: Association for Computing Machinery (ACM)
  **Views**: 276
  **YouTube ID**: AjNkHz3fRjs

- **Title**: WebSci'24: Keynote by Dirk Hovy
  **Duration**: 60m 16s
  **Channel**: Association for Computing Machinery (ACM)
  **Views**: 162
  **YouTube ID**: 0UGhxVwRmcU

- **Title**: IOHanalyzer: Detailed Performance Analyses for Iterative Optimization Heuristics
  **Duration**: 7m 3s
  **Channel**: Association for Computing Machinery (ACM)
  **Views**: 268
  **YouTube ID**: TZhUWPzAvJc

- **Title**: EAAMO'22: Liquid Democracy in Practice: An Empirical Analysis of its Epistemic Performance
  **Duration**: 19m 28s
  **Channel**: Association for Computing Machinery (ACM)
  **Views**: 263
  **YouTube ID**: ThXTZQN␣m3E

- **Title**: First Talk IDC 2022 Europe Best Paper Video
  **Duration**: 14m 24s
  **Channel**: Association for Computing Machinery (ACM)
  **Views**: 233
  **YouTube ID**: k␣gGx9ObXeM

- **Title**: Document Clustering vs Topic Models: A Case Study
  **Duration**: 16m 31s

**Channel**: Association for Computing Machinery (ACM)
**Views**: 1,300
**YouTube ID**: 1CJvt4OjWgs

- **Title**: KDD Keynote Talk–On the Nature of Data–Jeffrey D Ullman
  **Duration**: 60m 13s
  **Channel**: Association for Computing Machinery (ACM)
  **Views**: 2,200
  **YouTube ID**: Kkx–T5NUy4

- **Title**: My Mouse, My Rules: Privacy Issues of Behavioral User Profiling via Mouse Tracking
  **Duration**: 14m 52s
  **Channel**: Association for Computing Machinery (ACM)
  **Views**: 690
  **YouTube ID**: Dxit4BOO0y8

- **Title**: Simulation of cognitive agents to explore occupants' wayfinding experience in future buildings
  **Duration**: 27m 48s
  **Channel**: Association for Computing Machinery (ACM)
  **Views**: 486
  **YouTube ID**: 7XH53R675Qw

- **Title**: SIGCOMM 2020 Keynote: Amin Vadhat: Coming of Age in the Fifth Epoch of Distributed Computing
  **Duration**: 49m 46s
  **Channel**: Association for Computing Machinery (ACM)
  **Views**: 3,200
  **YouTube ID**: 27zuReojDVw

- **Title**: ACM ICN 2020 - Discovering in-network Caching Policies in NDN Networks from a Measurement Perspect.
  **Duration**: 21m 2s
  **Channel**: Association for Computing Machinery (ACM)
  **Views**: 852
  **YouTube ID**: LvQWRp1lwzY

- **Title**: L@S 2020: "Human Languages in Source Code: Auto-Translation for Localized Instruction"
  **Duration**: 11m 43s
  **Channel**: Association for Computing Machinery (ACM)
  **Views**: 354
  **YouTube ID**: 9Rxb2px3QcI

- **Title**: Session 1A - Bipartite TSP in O(1.9999) Time, Assuming Quadratic Time Matrix Multiplication
  **Duration**: 25m 0s
  **Channel**: Association for Computing Machinery (ACM)
  **Views**: 4,500
  **YouTube ID**: -dVb6j_Ooe4

- **Title**: "Reinforcement Learning for Recommender Systems: A Case Study on Youtube," by Minmin Chen
  **Duration**: 33m 17s
  **Channel**: Association for Computing Machinery (ACM)
  **Views**: 24,000
  **YouTube ID**: HEqQ2_1XRTs

- **Title**: Phishing Attacks on Modern Android
  **Duration**: 19m 54s
  **Channel**: Association for Computing Machinery (ACM)
  **Views**: 9,100
  **YouTube ID**: J1tx4OZ_wMc

- **Title**: TINY LM Agents on Edge Devices: Can We Scale?
  **Duration**: 28m 8s

**Channel**: Discover AI
**Views**: 2,964
**YouTube ID**: TU19Orwu4jE

- **Title**: Large-Scale Student Data Reveal Sociodemographic Gaps in Procrastination Behavior
  **Duration**: 29m 51s
  **Channel**: Association for Computing Machinery (ACM)
  **Views**: 131
  **YouTube ID**: rqXZDJiKyA8

### B.2 TEXTBOOK

We used the Latex source of the textbook *Text Data Management and Analysis: A Practical Introduction to Information Retrieval and Text Mining* (Zhai and Massung, 2016) as the collection of source information related to some lecture videos in JIR-ARENA.

### B.3 PAPERS

The papers JIR-ARENA covers include: Lee et al. (2018), Lindenbaum et al. (2018), Arik et al. (2018), Chan et al. (2018), Tatbul et al. (2019), Zhang and IV (2018), Chen et al. (2019), Whittington et al. (2018), Tobar (2019), Ke et al. (2018), Yi et al. (2019), Li et al. (2020), Shivkumar et al. (2018), Sharma et al. (2018), Cen and Alur (2024), Mulayim et al. (2024), Nagarathinam and Vasan (2024), Wang et al. (2022), Revel and mrevel (2022), Tisza et al. (2022), Xia et al. (2022), Mao et al. (2022), Shah and Bender (2022), Yuan et al. (2022), Leiva et al. (2021), Perera et al. (2020), Fan et al. (2020), Piech and Abu-El-Haija (2019), Nederlof (2020), Aonzo et al. (2018), Linton (2018), Haar Horowitz et al. (2018), Yang et al. (2024), and Sabnis et al. (2022).

### B.4 JIR-ARENA DATASET STATISTICS

Table 3: Statistics of different video categories.

| Category | Scene Num | Avg View Count / Scene | Avg Duration / Scene (min) | Avg JIR Num / Min |
|---|---|---|---|---|
| Lectures | 16 | 6,100,530 | 19.9 | 13.1 |
| Conferences | 18 | 2,921 | 28.4 | 10.4 |

JIR-ARENA includes 2 categories of scenes, totaling 34 of them and spanning 831 mins. We select the most viewed videos (by **Avg View Count / Scene**) with moderate duration (by **Avg Duration / Scene (min)**) and of intensive information needs (by **Avg JIR Num / Min**).

## C RATIONALE AND VALIDATION FOR USING VLM NARRATIONS

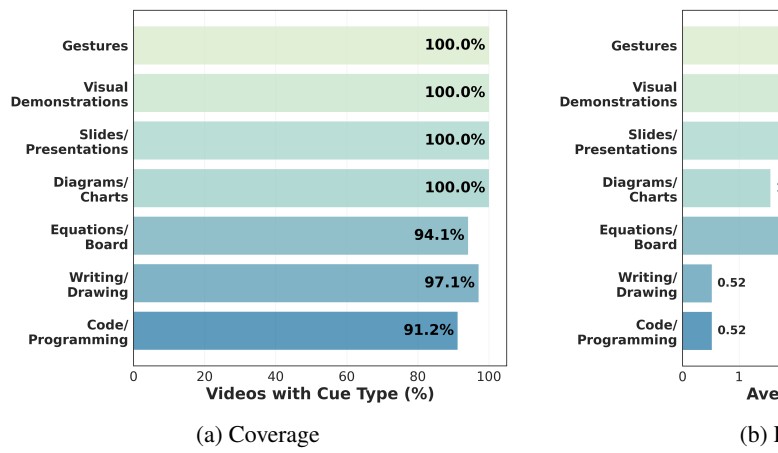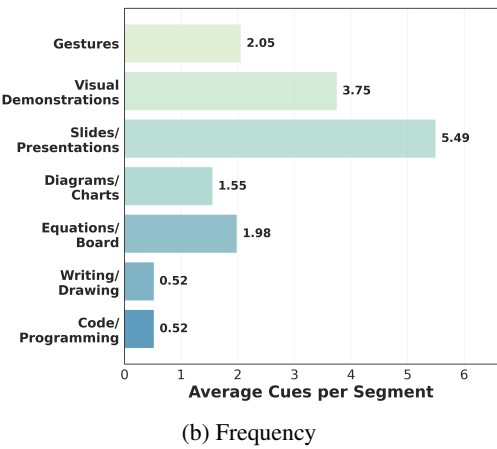

(a) Coverage                (b) Frequency

Figure 5: Analysis of informative visual cues in video narratives. (a) Coverage: Percentage of videos and segments containing each visual cue type. All videos (100%) contain gestures, diagrams/charts, visual demonstrations, and slides, demonstrating the universal presence of visual information. (b) Frequency: Average occurrences per video and per segment for each visual cue type. These indicate the VLM NVILA captures useful visual elements in narrative form.

To ensure consistency across a wide range of open-source and API-based generative models, we standardize both audio signals (via Whisper transcription) and video signals (via NVILA-generated narrations) into text form. This unified textual representation is essential for enabling modality-agnostic comparison across LLMs and VLMs, many of which cannot directly process raw video streams. More importantly, using multimodal-model-generated narrations is currently the most practical and computationally feasible method for incorporating video information while continuously processing streaming inputs. Existing video-capable models are predominantly designed and optimized for video-based question answering over short, bounded clips, rather than for simulating users' cognitive flow or predicting evolving information needs in long-horizon interactive settings (Tang et al., 2025; Zou et al., 2024; Wu et al., 2025); thus, they neither provide the required output structure nor operate efficiently at scale for our simulation tasks. Our design therefore reflects the processing constraints of underlying multimodal base models: converting visual signals into VLM-generated textual narrations preserves semantically rich video information while avoiding the prohibitive cost and limited flexibility of direct video inference.

More importantly, our analysis demonstrates that visual cues are both pervasive and highly informative, and thus converting NVILA's video narrations into text does not discard visual information. Across 34 analyzed videos (1,676 segments), 100% contained meaningful visual cues (see Figure 5). Gestures appeared in 76.07% of segments (3,443 occurrences), equations or board content in 40.04% of segments (3,323 occurrences), and presentation slides in 89.02% of segments with an average of 5.49 slide-based cues per segment (9,209 total occurrences). Videos also frequently contained diagrams and charts (2,601 occurrences), visual demonstrations (6,283 occurrences), writing/drawing (866 occurrences), and code snippets (869 occurrences). We provide representative examples for each major category of visual cues in the following section.

These cues are not only abundant but also semantically rich. For example, gestures often accompany conceptual explanations, such as an instructor repeatedly pointing to different regions of a blackboard while describing algorithm steps (Video `HtSuA80QTyo`, 1320s–1350s). Diagrams and charts supply key domain information—for instance, a detailed periodic-table slide highlighting Carbon's atomic structure (Video `IFKnq9QM6_A`, 600s–630s) or scatter-plot visualizations of language patterns on GitHub (Video `9Rxb2px3QcI`, 300s–330s). Technical content such as mathematical derivations (e.g., $T(n) = T(n/2) + O(1)$ on a blackboard) and writing of limits and functions on whiteboards also appears frequently and is often not verbalized in the audio stream (Videos `HtSuA80QTyo`, `BRRolKTlF6Q`). Likewise, code presentations and internationalization flowcharts (Video `9Rxb2px3QcI`) provide essential semantic cues that are inherently visual.

Because NVILA captures these visual elements in narrative form, converting the narrations to text enables LLMs and VLMs to leverage visual information while remaining within the computational constraints of large-scale simulation. The breadth, frequency, and semantic depth of these cues confirm that visual information is substantively preserved through our text-standardized pipeline.

## D    VISUAL CUE EXAMPLES IN VLM NARRATIONS

To complement the above quantitative results, this section provides representative qualitative examples for each major category of visual cues. These examples illustrate the diversity, ubiquity, and instructional relevance of the visual phenomena captured in NVILA's video corpus, and further motivate why converting NVILA's narrated videos into text preserves—rather than loses—visually grounded information. All examples are taken directly from the video narrations.

**Gestures.**   **Example 1** (Video: `HtSuA80QTyo`, 1320s–1350s):

The man is pointing to various parts of the blackboard with his left hand, likely explaining the concepts being displayed. He is holding a piece of paper in his right hand. The professor seems to be in the middle of explaining a particular concept, possibly demonstrating how an algorithm works or how image compression techniques are applied.

**Example 2** (Video: `HtSuA80QTyo`, 1440s–1470s):

The professor is gesturing with his hands as he speaks, using a piece of paper in his left hand. He seems to be emphasizing certain points in his explanation.

**Diagrams and Charts.**   **Example 1** (Video: `IFKnq9QM6_A`, 600s–630s):

The video displays a detailed educational presentation about the periodic table of elements. A colorful periodic table with Carbon highlighted in yellow, indicating its atomic number (6) and atomic mass

(12). A diagram showing the atomic structure of Carbon, with 6 protons, 6 neutrons, and 6 electrons. A close-up image of a diamond, another form of carbon, showcasing its crystalline structure.

**Example 2** (Video: `9Rxb2px3QcI`, 300s–330s):

The third slide is titled "Language Patterns on GitHub" and contains a scatter plot graph. There are two data points labeled "French" and "Korean" on the graph. It has the same title and graph, but the data points are labeled "French" and "Dutch."

5. The fifth slide is identical to the fourth, with the same title, graph, and data points.

**Visual Demonstrations.   Example 1** (Video: `BRRolKTlF6Q`, 420s–450s):

The video appears to be a math lesson or tutorial focused on explaining mathematical concepts. It's divided into several segments, each showing different parts of the lesson. This suggests the instructor is explaining the concept of zero raised to the power of zero. The second segment shows a close-up of a man's face.

**Example 2** (Video: `2s4TqVAbfz4`, 1140s–1170s):

It shows an older man, likely in his 70s or 80s, seated at a cluttered desk. The desk is filled with various items, including: - A computer monitor and keyboard - A stack of papers and folders - A printer - A green plastic model of a geometric shape, which the man is holding in his hands

The background shows shelves with books and other office supplies. There's also a poster or screen displaying a green geometric shape, possibly related to the model the man is holding. The man seems to be explaining or demonstrating something about the geometric model, though the exact content of his explanation isn't clear from the still images.

**Slides and Presentations.   Example 1** (Video: `v68zYyaEmEA`, 360s–390s):

The video appears to be a series of slides from a presentation, likely related to statistics or probability. The slides are set against a black background with white text, creating a stark contrast that makes the information easy to read. The first slide shows a grid with the word "WEARY" highlighted in yellow. It mentions "58 Possible matches" and "12,972 Total words," with a probability calculation of "p(WEARY) = 58 / 12,972 = 0.0045."

The second slide is similar but shows "1,387 Possible matches" and "12,972 Total words," with a probability calculation of "p(WEARY) = 1,387 / 12,972 = 0.107."

The third slide shows "1,419 Possible matches" and "12,972 Total words," with a probability calculation of "p(WEARY) = 1,419 / 12,972 = 0.1094."

The fourth slide is different, with a black background and white text.

**Example 2** (Video: `9Rxb2px3QcI`, 300s–330s):

The video appears to be a presentation slide deck discussing language patterns on GitHub. It's divided into several slides, each with different content:

1. The first slide has a gold key icon and text that reads: "You can better estimate a user's L1 language by including their git commit messages." The second slide is black with the word "[Suspense]" written in white.

**Writing and Drawing.   Example 1** (Video: `BRRolKTlF6Q`, 450s–480s):

The video shows a person writing mathematical equations on a whiteboard. The individual is using a blue pen to write various mathematical expressions, including limits and functions. The person appears to be smiling as they write. The whiteboard displays several equations, with the most prominent one being "lim x→0 $x^2$".

**Example 2** (Video: `HtSuA80QTyo`, 1980s–2010s):

The video shows a man standing in front of a blackboard. He appears to be a professor or lecturer, as he is writing on the board with a yellow marker. The blackboard is filled with mathematical equations and symbols, including "T(n)" and "work" written in white chalk. The man is actively engaged in teaching, pointing to different parts of the board and explaining the concepts.

**Equations and Board Content.   Example 1** (Video: `HtSuA80QTyo`, 2040s–2070s):

The blackboard displays a mathematical equation related to algorithm analysis:

T(n) = T(n/2) + O(1)

The man is pointing to this equation with a yellow marker. Below the main equation, there's additional text that reads: "work", "algo does on", "input of size n". The man appears to be explaining the concept of divide-and-conquer algorithms and their time complexity.

**Example 2** (Video: `BRRolKTlF6Q`, 450s–480s):

The video shows a person writing mathematical equations on a whiteboard. The individual is using a blue pen to write various mathematical expressions, including limits and functions. The whiteboard displays several equations, with the most prominent one being "lim x→0 $x^2$". This equation is written multiple times across the board, suggesting it's a key concept being explained or demonstrated.

**Code and Programming.**    **Example 1** (Video: `9Rxb2px3QcI`, 270s–300s):

The video appears to be a presentation slide about analyzing human language in code. The slide is in Chinese and English, with the main title "How did we analyze human language in code?" at the top. The slide explains that they looked at three aspects of code to analyze human language:

1. Identifiers

The English section shows code snippets with comments in English.

**Example 2** (Video: `9Rxb2px3QcI`, 510s–540s):

The video appears to be a presentation slide about "Code Internationalization" in programming. The top half shows a flowchart illustrating the process of internationalizing code. It demonstrates how code written in English is transformed into an internationalized version, which can then be translated into Chinese. The bottom half displays a comparison between English and Chinese code.

# E    LLM PROMPTS FOR USER INFORMATION NEED SIMULATION

**User Information Need Simulation from Transcript**

```
Analyze this transcript segment for information needs.  I will
provide you with a summary of the presentation so far and a
transcript segment from the presentation.

Categorize each information need into one of the following
categories:

1.  Visual References (graphs, images, diagrams, etc.)
2.  Technical Terms (jargon, acronyms, formulas, definitions)
3.  Data & Sources (uncited stats, vague claims like "studies
show...")
4.  Processes/Methods (unexplained workflows/algorithms)
5.  External Content (papers, tools, historical references without
context)
6.  Ambiguous Language (vague terms like "many" or "significant")
7.  Missing Context (assumed prior knowledge, undefined goals)
8.  Instructions/Actions (unclear steps, implied tasks)
9.  Code/Formulas (unexplained pseudocode/equations)
10.  Future Work (vague next steps, unresolved questions)
11.  Conceptual Understanding (concepts, ideas)

I need an exhaustinve list of information needs.  Each sentence
can need multiple information needs.
Return a JSON list with 'type', 'subtype', and 'reason' for each
need.  Ensure the output is strictly JSON-compliant

Summary of Presentation So Far:
{{Background_Context}}

Presentation Transcript Segment:
{{Transcript_Chunk}}
```

```
Output Format (JSON):
{ "information_needs": [ { "sentence_id": "sentence_id", "type":
"need type", "subtype": "need subtype", "reason": "reason for
need" }, { "sentence_id": "sentence_id", "type": "need type",
"subtype": "need subtype", "reason": "reason for need" }, {
"sentence_id": "sentence_id", "type": "need type", "subtype":
"need subtype", "reason": "reason for need" } ] }
```

**User Information Need Simulation from Transcript and Narrative**

Analyze this description of a presentation for information needs.
I will provide you with a summary of the presentation so far and a
segment from the presentation.

Categorize each information need into one of the following
categories:

1.  Visual References (graphs, images, diagrams, etc.)
2.  Technical Terms (jargon, acronyms, formulas, definitions)
3.  Data & Sources (uncited stats, vague claims like "studies
show...")
4.  Processes/Methods (unexplained workflows/algorithms)
5.  External Content (papers, tools, historical references without
context)
6.  Ambiguous Language (vague terms like "many" or "significant")
7.  Missing Context (assumed prior knowledge, undefined goals)
8.  Instructions/Actions (unclear steps, implied tasks)
9.  Code/Formulas (unexplained pseudocode/equations)
10.  Future Work (vague next steps, unresolved questions)
11.  Conceptual Understanding (concepts, ideas)

I need an exhaustinve list of information needs. Each sentence
can need multiple information needs.
Return a JSON list with `type`, `subtype`, and `reason` for each
need. Ensure the output is strictly JSON-compliant

Summary of Presentation So Far:
{{Background_Context}}

Presentation Description Segment:
{{Presentation_Chunk}}

Output Format (JSON):
{ "information_needs": [ { "sentence_id": sentence_id "type":
"need type", "subtype": "need subtype", "reason": "reason
for need" }, { "sentence_id": sentence_id "type": "need type",
"subtype": "need subtype", "reason": "reason for need" }, {
"sentence_id": sentence_id "type": "need type", "subtype": "need
subtype", "reason": "reason for need" }, ] }

**Summary**

You are an AI assistant repsonsible for generating a detailed
summary of all the sentences from a transcript provided to You
Ensure the output is strictly JSON-compliant

```
Input:

Transcript Sentences:
{{Sentences}}

Output Format (JSON):
{ "summary":  "This is the summary of all the sentences", }
Explanation of Each Output Field:

summary:  Detailed Summary of all the sentences as a single
paragraph
```

**Likelihood Judgement**

```
You are an AI Judge, evaluating how relevant an informational need
is to a presentation | as if you were a thoughtful human attending
the talk.
Your job is to score the need's relevance on a scale from 0 to
10 based on how likely it is that a curious, context-aware human
would naturally have this question or need at this exact point in
the presentation.
The relvance score must be in the range 0 to 10 where:

0 { Completely irrelevant:  No connection whatsoever to the
content.
1 { Barely related:  Random term overlap; totally misplaced in
context.
2 { Weakly related:  Vague thematic connection, but wouldn't arise
from this presentation.
3 { Marginally related:  A human could get here with effort, but
it feels out of place.
4 { Somewhat related:  On-topic, but not something a typical
attendee would care about now.
5 { Mildly relevant:  Plausible side question, but still feels
like a stretch or a detour.
6 { Reasonably relevant:  A thoughtful listener might ask this,
though it's not the most pressing or natural next step.
7 { Clearly relevant:  A typical, attentive participant could
raise this with no prompting.  Fits the flow.
8 { Strongly relevant:  Feels like a helpful and likely next
question from a human audience member.  Supports or extends what's
being discussed.
9 { Very relevant:  Almost anticipates what the speaker might say
next.  Shows deep understanding and interest.
10 { Perfectly aligned:  A human would almost certainly ask this
next.  Feels like the natural continuation of the discussion.

Evaluation Guidelines
Imagine yourself as a human audience member who has been following
the presentation closely.
Consider flow, timing, speaker's tone, and logical build-up.
Use 7 or higher ONLY for needs that a genuinely attentive human
would likely raise unprompted.
Err on the side of strictness:  if the connection feels forced,
don't go above 6.
This is not about keyword overlap | it's about human intent,
```

curiosity, and conversational flow.

Return a JSON object with a top-level key score, which is a list of objects. Each object must include a sentence_id, a numerical relevance_score, and a relevance_score_reason explaining why that score was given.

Summary of Presentation So Far:
{{Background_Context}}

Pervious Sentences:
{{Prev_Sentences}}

Presentation Transcript Segment:
{{Transcript_Chunk}}

Identified Information Needs:
{{Information_Need}}

Strictly follow the below JSON output format
Output Format (JSON):
{ "score": [ { "sentence_id": "sentence_id", "relevance_score_reason": "reason for the relevance score", "relevance_score": "relevance score" }, { "sentence_id": "sentence_id", "relevance_score_reason": "reason for the relevance score", "relevance_score": "relevance score" } ] }

**End Time Judgement**

You are an expert transcript analyzer. Your task is to determine how long a specific information need remains relevant in a conversation.

You will be given the following:
- A summary of the transcript up to the current point.
- A few sentences before the current transcript segment (Prev_Sentences).
- A few sentences after the current transcript segment (Next_Sentences).
- The current transcript segment being analyzed.
- An information need that this segment potentially addresses.

Based on the context, identify the last sentence (within the current and next few sentences) where this information need is still relevant. If it's no longer relevant immediately after the segment, return the current segment's last sentence.

If there are multiple information needs, you need to find the end_sentence_id for each of the needs
Only use the given context | do not assume anything beyond what is provided.
---

Summary:
{{Summary}}

```
Previous Sentences:
{{Prev_Sentences}}

Transcript Segment:
{{Transcript_Chunk}}

Next Sentences:
{{Next_Sentences}}

Information Need:
{{Information_Need}}

---

Return a JSON object with a top-level key end_time, which is a
list of objects.  Each object corresponds to one information need
and includes:
- 'end_sentence_id':  the ID of the last sentence where the need is
relevant
- 'reason':  a brief explanation for why that sentence was chosen
Ensure the output is strictly JSON-compliant and follows the below
format.

Output Format (JSON):
{ "end_time":  [ { "end_sentence_id":  "42", "reason":  "The
discussion about time management ends at this point." }, {
"end_sentence_id":  "45", "reason":  "The speaker stops referencing
motivation strategies here." } ] }
```

**Question Formatter**

```
Convert the following sentence into a question.  Questions need to
be Wh- questions

Input:

Sentence:
{{text}}

Output Format (JSON):
{"question" :  "Generated Question"}
```

## F  DETAILED ANALYSIS OF HUMAN ANNOTATIONS

In this section, we present the results of a detailed analysis of the human-annotated scenes. Our goal is two-fold: First, as JIR is a new task, we would like to better understand this task by examining the variations of human annotations. Due to the variable backgrounds of users, we expected the users to have different needs, but the interesting question here is to what extent their needs vary and whether they may also share some common information needs. Second, we are interested in making a comparison between LLM annotations and human annotations. To this end, we have four annotators watch the scenes and record the timestamp and content of questions, if any.

In Figure 6, we show the distribution of needs for a single scene. The x-axis is the timestamp, and the y-axis (and color) indicates the labeler. The graph includes the four annotators along with the LLMs.

We can make several observations from Figure 6: First, as expected, the needs identified by the human annotators vary significantly in both the number of needs and the timing of the needs. We found

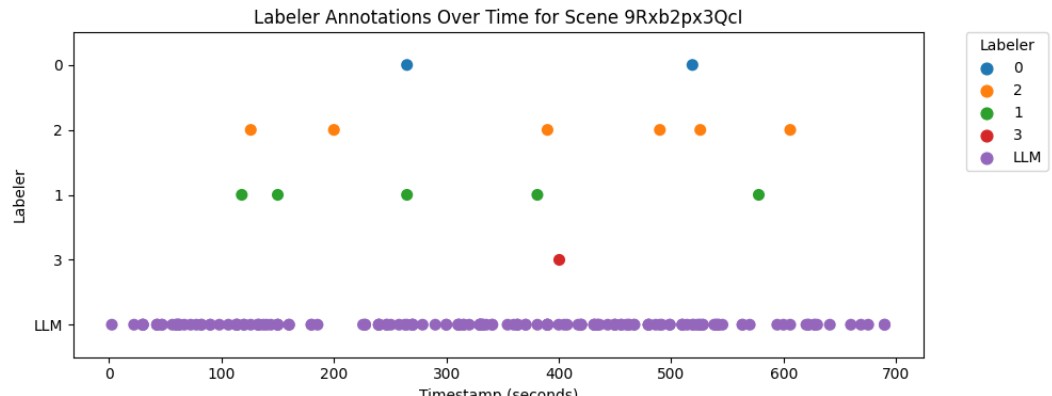

Figure 6: Example of labeled needs across scene 9Rxb2px3Qcl in addition to the generated needs by the language models.

this to be the case in all the analyzed human-annotated scenes. Using a window size of ten seconds before and after a reference time point, we can further compute the time-wise "unique" human needs - namely, those that occur more than ten seconds away from any other human need. The percentages of such unique needs for all human videos are shown in Table 4. From Table 4, we see that, in many cases, the majority of human-generated needs were unique time-wise, indicating significant variances across people in the time when a need occurs. However, we should note that in a real application scenario, a deviation of 10 seconds in recommendation of information may not affect the utility of the recommended information that much, so the natural variation of the time stamp among human annotators also means that when we evaluate a JIR system, we may also tolerate a small amount of deviation in the time of recommendation.

| Scene ID | 9Rxb2px3QcI | Dxit4BOO0y8 | TU19Orwu4jE | k_gGx9ObXeM | IFKnq9QM6_A |
|---|---|---|---|---|---|
| Percentage | 42.9% | 52.7% | 68.6% | 54.5% | 58.8% |

| Scene ID | MXJ-zpJeY3E | ghZRzOb_bZo | v68zYyaEmEA | wvXDB9dMdEo | |
|---|---|---|---|---|---|
| Percentage | 33.3% | 54.5% | 87.5% | 66% | |

Table 4: The percentage of time-wise unique human needs for each scene.

Second, we see that the LLMs identified many more needs than the human annotators. Based on human verification, most of the identified needs by LLMs are reasonable. There may be two reasons why human annotators did not identify so many needs as the LLMs: 1) Humans had limited cognitive processing capacity, and they were only able to identify some of the needs while watching a video in one pass; if human annotators were allowed to re-watch the video multiple times, they might also be able to identify more needs. 2) Humans tend to be satisfied with a "basic" understanding of the content, whereas the LLMs sometimes "demanded" more detailed justification or explanation; the LLMs appear to have a tendency to ask for an elaboration whenever a general statement is made (e.g., if A is said to be better than B, the LLM may ask for evidence). From an application perspective, the much larger number of needs identified by the LLMs makes the dataset potentially better represent a wider range of needs of real users than using humans to identify the needs. We have also found that the generated questions by LLMs are of multiple types, including both elaboration questions and why questions.

Next, we also compared the semantic similarity of the human-generated needs, both to each other and to the LLM-generated needs. As an example, Figure 7 depicts the semantic similarity heatmap comparing all human-generated needs for scene 9Rxb2px3Qcl.

From the example depicted in Figure 7, and the others, we found that many of the human-generated needs are semantically dissimilar, showing that not only did the time of need vary, but the specific information needed by humans also varies significantly. In Table 5, we list the mean similarity and standard deviation for all human-generated needs per scene. Similarity was measured using MiniLM-L6-V2 from sentence-transformers Reimers and Gurevych (2021). The low mean similarity shows that the needs identified from the same video are semantically different. This may be expected to some extent because at a different time point, the content discussed in the scene is likely different

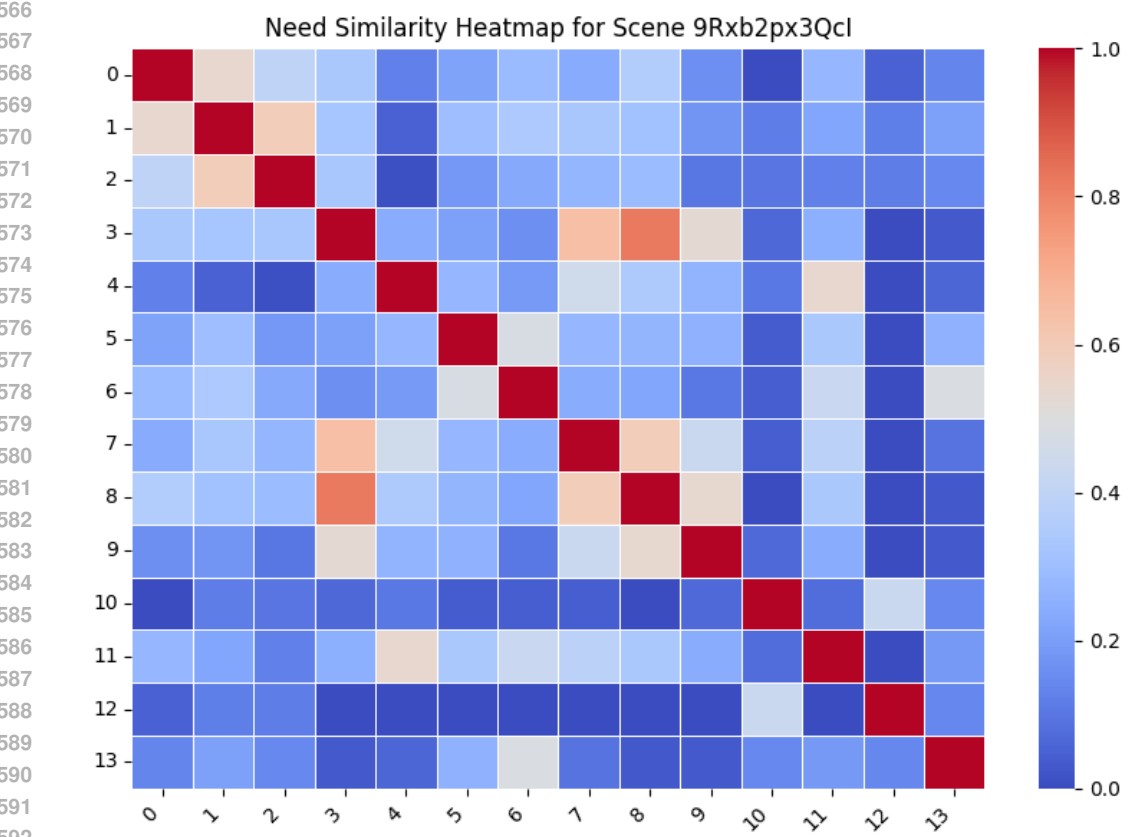

Figure 7: Example of human-generated need similarity for scene 9Rxb2px3Qcl.

and thus the need is also likely different semantically. However, what is somewhat surprising is that even the needs identified in adjacent time windows do not always have a high similarity, again suggesting variable information needs among humans. This also suggests that in a real application, it would be beneficial for a JIR system to allow a user to enter a query as well, which would facilitate human-AI collaboration in precisely specifying the information need. However, we do see that in some time periods (e.g,. at the very beginning and middle of this video), the adjacent distinct needs have higher semantic similarity, forming a small cluster. This suggests that there are also common needs shared by human annotators (those are likely similar needs identified at slightly different time points). Such common needs can also be visually seen from Figure 6.

| Scene ID | 9Rxb2px3QcI | Dxit4BOO0y8 | TU19Orwu4jE | k_gGx9ObXeM | IFKnq9QM6_A |
|---|---|---|---|---|---|
| Similarity | 0.2195 | 0.1630% | 0.1896 | 0.1734 | 0.2904 |
| Std. Dev. | 0.1839 | 0.1668% | 0.1549 | 0.1493 | 0.2069 |
| Scene ID | MXJ-zpJeY3E | ghZRzOb_bZo | v68zYyaEmEA | wvXDB9dMdEo | |
| Similarity | 0.1858 | 0.3491% | 0.1673 | 0.1951 | |
| Std. Dev. | 0.1605 | 0.2406% | 0.1597 | 0.1329 | |

Table 5: The mean similarity and standard deviation for human-generated needs of each scene.

Finally, we compared the human annotations to the LLM-generated annotations. To do so, we again applied the window, and only included LLM-generated needs if they fell within ten seconds of a human-generated need. We depict an example of this in Figure 8.

Here, we can see a slightly similar diagonal, indicating that the language models were able to generate needs that overlap with some of the human-generated needs, especially the shared common needs by human annotators at the beginning and in the middle. We further measured the number of human-generated needs that are covered by the language models (namely, above a similarity threshold of 0.55) and we report these results in Table 6. The results show that the coverage varies across scenes. A lecture-style scene on a focused topic (i.e., ghZRzOb_bZo ) may have a coverage as high as over 90%, while an informal narrative scene about a person (i.e., MXJ-zpJeY3E) may have a coverage

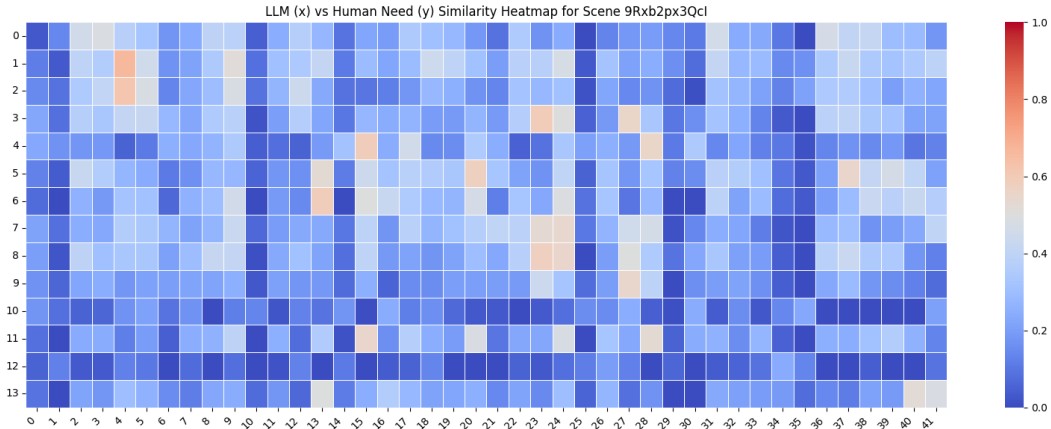

Figure 8: Example of similarity between LLM-generated needs and human-generated needs for scene 9Rxb2px3Qcl.

as low as 25%. Intuitively, this is expected, and in general, the coverage is related to the entropy of the information covered in the scene with a higher entropy likely correlated with a low coverage (as it would be intuitively harder to predict the information need). Further investigating this kind of hypothesis by doing more systematic analysis would be an interesting future direction. It also shows that our dataset can also be potentially useful for analysis of topics covered in those scenes, as a "by product." From the application perspective, a high coverage means that a JIR application system can be expected to serve many users very well via pure recommendation without requiring any user effort in the case of "low entropy scenes" (predicable information needs), while in the case of "high entropy scenes" (unpredictable information needs), the system would benefit from providing a query box to a user to accommodate ad hoc variations of their unique needs. Since the entropy can be measured using an algorithm, such an adaptive interface may be even automatically adjusted by an algorithm. This is another useful insight that we were able to derive from human annotations.

| Scene ID | 9Rxb2px3QcI | Dxit4BOO0y8 | TU19Orwu4jE | k_gGx9ObXeM | IFKnq9QM6_A |
|---|---|---|---|---|---|
| Percentage | 50% | 58.3% | 77.1% | 59.1% | 70.6% |

| Scene ID | MXJ-zpJeY3E | ghZRzOb_bZo | v68zYyaEmEA | wvXDB9dMdEo | |
|---|---|---|---|---|---|
| Percentage | 25% | 90.1% | 43.8% | 61.7% | |

Table 6: The percentage of human needs covered by language models.

# G  LLM Prompts for Information Retrieval Quality Check

**Information Retrieval Quality Check**

```
You are an assistant that evaluates whether a given query can
be answered using the content of a provided reference or source
material.
Your task is to:

1.  Determine whether the reference material contains sufficient
information to fully or partially answer the query.
2.  If yes, extract the relevant parts of the reference that help
answer the query.
3.  Return your answer in the following JSON format:

```json
{ "answerable":  true | false, "supporting_content":  "If
answerable is true, copy and paste only the relevant original
text from the reference material that helps answer the query.  Do
not rephrase, summarize, or explain|only extract directly.  If
```

```
answerable is false, leave this as an empty string." }
```
```
{{customed_instruction}}
```

```
Here is the query:
{{query}}
```

```
Here is the reference material:
{{reference_material}}
```

```
Only return the JSON object, with no extra explanation or
formatting.
```

**Customed Instructions**

```
Wikipedia:
The reference material comes from a Wikipedia article.  This
article is not the source where the query comes from, so if the
query is asking about the specific content, methods, or claims of
the paper/study being presented, this reference is insufficient.
In such cases, the correct response is änswerable:  false.
```

```
Arxiv:
The reference material includes the title and abstract of an
arXiv paper.  This is not the source where the query comes from,
so if the query is asking about the specific content, methods,
or claims of the paper/study being presented, this reference is
insufficient.  In such cases, the correct response is änswerable:
false.
```

```
Paper:
The reference material is extracted from the body of the research
paper being presented.  It may have been converted from a PDF
and could contain distortions such as missing figures with only
captions, incomplete tables, or malformed equations.  Ignore these
potential issues and assume the text reflects the intended content
of the original paper.  Your task is to evaluate whether the
given query can be answered based on the provided excerpt, even
partially, as if it were a faithful representation of the original
document.
```

```
Textbook:
The reference material comes from a relevant textbook specialized
on the topic being discussed.
```

## H  MANY-TO-MANY MATCHING ALGORITHM

We provide the mathematical formulation and key implementation details of our many-to-many matching algorithm.

### H.1  MATCHING CRITERIA

For a ground truth query $q_i \in Q$ and a candidate query $c_j \in C$, a match occurs if and only if:

1. **Time Overlap:**

$$\max(\hat{t}_{q_i}^{start}, t_{c_j}^{start}) \leq \min(\hat{t}_{q_i}^{end}, \hat{t}_{c_j}^{end}),$$

where $\hat{t}_{q_i}^{start} = t_{q_i}^{start} - \delta \cdot d$, $\hat{t}_{q_i}^{end} = t_{q_i}^{end} + \delta \cdot d$, with fuzzy interval $\delta$.

2. **Semantic Similarity:**

$$\cos(\text{Embed}(q_i.\text{text}), \text{Embed}(c_j.\text{text})) \geq \theta$$

The matching function can be expressed as:

$$M(q_i, c_j) = \mathbf{1}[\text{TimeOverlap}(q_i, c_j) \wedge \text{Sim}(q_i, c_j) \geq \theta].$$

### H.2 Handling Multiple Predictions per Query

The algorithm supports multiple candidates matching a single ground truth query:

- **All matches preserved:** $results[i] = [(c_j, s_{ij}), (c_k, s_{ik}), \dots]$.
- **Recall:** A query counts as matched if it has *at least one* match:

$$\text{Recall} = \frac{|\{q_i : \exists c_j \text{ s.t. } M(q_i, c_j) = 1\}|}{|Q|}$$

- **Relevance metrics:** All matched candidates contribute to nDCG/MRR/MAP.
- **Timeliness:** The best matching candidate (by time score) is selected.

### H.3 Avoiding Duplicate Counting

Each candidate is uniquely identified by $(\text{question\_text}, \text{start\_time}, \text{end\_time})$. Precision counts unique matched candidates:

$$\text{Precision} = \frac{|\{c_j : \exists q_i \text{ s.t. } M(q_i, c_j) = 1\}|}{|C|}.$$

Recall naturally avoids duplicates by counting queries, not matches:

$$\text{Recall} = \frac{|\{q_i : \exists c_j \text{ s.t. } M(q_i, c_j) = 1\}|}{|Q|}.$$

**Example:** If $q_1$ matches both $c_1$ and $c_2$, and $q_2$ also matches $c_2$:

- Precision counts unique candidates $\{c_1, c_2\} \to$ numerator = 2
- Recall counts matched queries $\{q_1, q_2\} \to$ numerator = 2

## I Details of the Baseline JIR Systems

### I.1 Illustration of the Pipeline of a Prototypical JIR System

This is the naive version of a JIR system, which: *i)* considers a sliding window of the multimodal context, *ii)* generates user information need with a generative model, and *iii)* for each need, retrieves relevant documents to address the need. In our implementation, we consider the most informative text modal input, using generative models GPT-4O, DEEPSEEK-V3 (671B), CLAUDE-3.7, and GEMINI-2.0-FLASH, and adopt OPENSEARCH as our retrieval model.

### I.2 Information Need Detection Prompts

```
Given the transcript formatted by lines of timestamps/words,
predict the questions that a listener might have.  Respond in the
following JSON format:

"needs":  [
"start_time":  <str, the earliest timestamp that the specific
question may appear>, "end_time":  <str, the latest timestamp
that the specific question may appear>, "question":  <str, the
description of the question>  ]
Here is the transcript:
<begin_transcript>
```

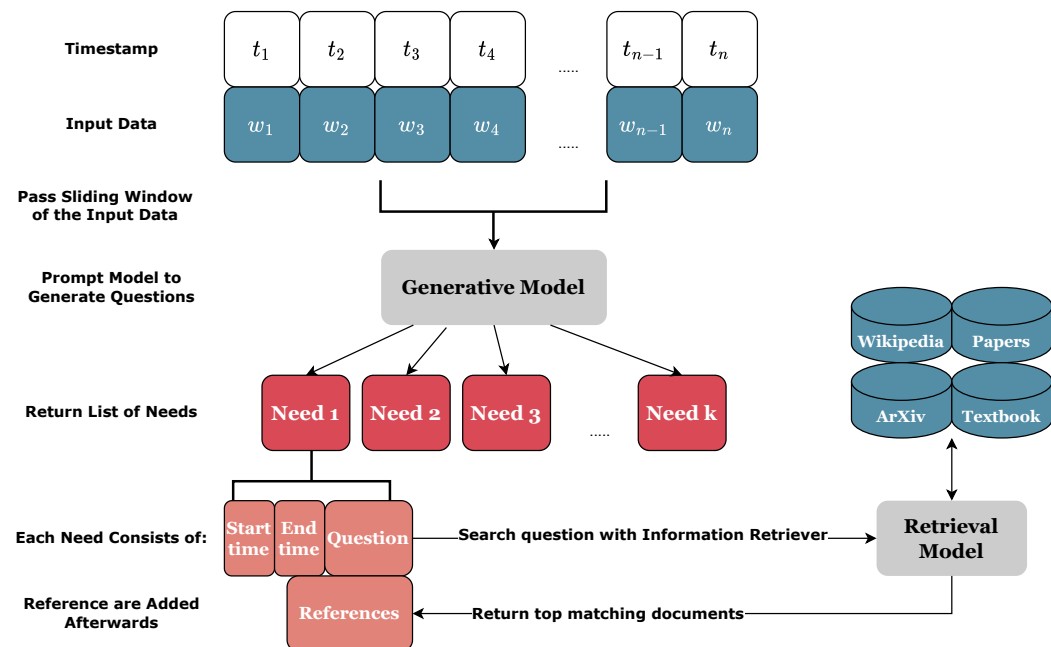

Figure 9: The prototypical JIR system pipeline.

```
<REPLACE_WITH_TRANSCRIPT>

<end_transcript>

As a reminder, given the above transcript with timestamps,
predict the questions that a listener might have.  Respond in the
following JSON format:

"needs":  [
"start_time":  <str, the earliest timestamp that the specific
question may appear>, "end_time":  <str, the latest timestamp
that the specific question may appear>, "question":  <str, the
description of the question>  ]
Make sure to follow these rules
- Include all context in a question so that it can be answered
without the transcript
- Only include a single question/need per needs block.
- Simulate a curious listener interested in the presented topic -
questions should be reasonably likely to be asked by an attendee
of this presentation.

Respond only in the above JSON format, with nothing else.
```

## J    ORACLE ANALYSIS FOR BASELINE BOTTLENECK LOCALIZATION

A JIR system consists of two main components: (1) a **generative model** that predicts user information needs and generates queries, and (2) a **retrieval model** that retrieves relevant documents using these queries. To determine whether the bottleneck lies in query generation or retrieval systems, we conducted an oracle study where we replaced the generative model's predicted queries with **ground truth queries** from the dataset (the oracle queries). This "Oracle Generator" uses perfect queries (Recall: 1.000, Precision: 1.000) to test the retrieval system's upper bound performance. Table 1

Table 7: Oracle vs Baseline Performance Comparison

| Metric | Oracle | Baseline | Improvement | Improvement % |
|---|---|---|---|---|
| Recall | **1.000** | 0.347 | +0.653 | +188.4% |
| Precision | **1.000** | 0.724 | +0.276 | +38.0% |
| Relevance (O) | 0.215 | 0.051 | +0.164 | +319.2% |
| Timeliness | **1.000** | 0.182 | +0.818 | +449.8% |

shows Oracle Generator performance (Relevance: 0.215/0.195/0.274 for O/D/R) vs baseline models that use generative model predictions. We average the baselines' results and present them in Table 7.

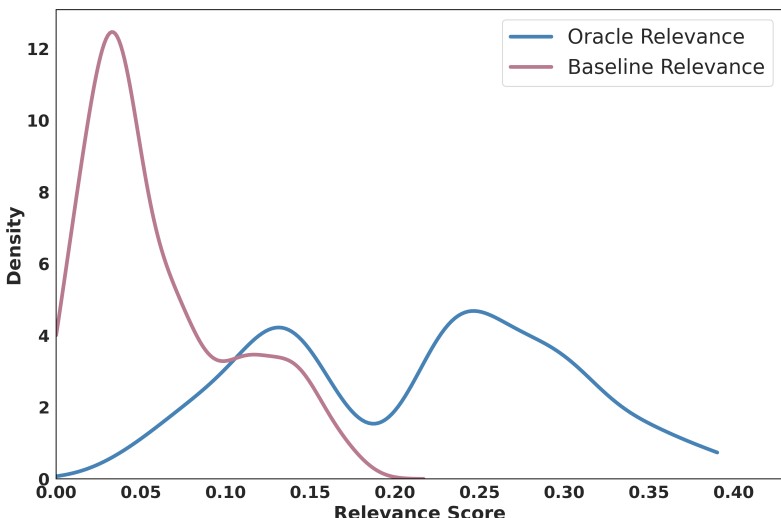

Figure 10: Density distribution of relevance scores for Oracle (using ground truth queries) and Baseline models. The Oracle distribution is shifted to higher relevance values compared to Baseline, yet even with perfect query generation, the average relevance remains low, indicating that the retrieval system itself is a bottleneck.

The oracle study reveals **both the user information need predictor and the retrieval system are bottlenecks**:

1. **User Information Need Predictor Bottleneck (Primary):** Baseline systems achieve only 0.347 average recall, meaning 65.3% of user information needs are not predicted. Error analysis shows semantic mismatch causes 62.91% of failures—predicted queries fail to semantically match oracle needs. Critically, even for high-likelihood information needs (likelihood score $\geq 8.5$ over 10, indicating high probability of being raised by users based on our multi-entity annotation process), the system fails to achieve adequate coverage; recall remains poor at only 41.38%, far below acceptable levels. This explains why baseline relevance is low (mean: 0.061): insufficient query prediction leads to too few queries to retrieve relevant documents.

2. **Retrieval System Bottleneck (Secondary but Significant):** Even with perfect ground truth queries, oracle relevance remains low (0.215 OpenSearch, 0.274 Reranked), far below ideal. The improvement from OpenSearch to Reranked (+36%) shows that retrieval models matter, but even the best retrieval system (Reranked) achieves only 0.274 relevance with perfect queries (see Figure 10 for detailed relevance density analysis). This suggests the limitation is not solely in query generation—retrieval systems themselves need improvement. One of the key reasons is that queries are highly context-dependent; without contextual information, they appear vague, leading to low retrieval quality even with oracle queries. Contextual information includes content context (topic), temporal context (timeline), visual context (diagrams, animations), domain context (technical field), and methodological context (specific methods). For example, an oracle query like "What is the 'subtle shift in perspective'?"

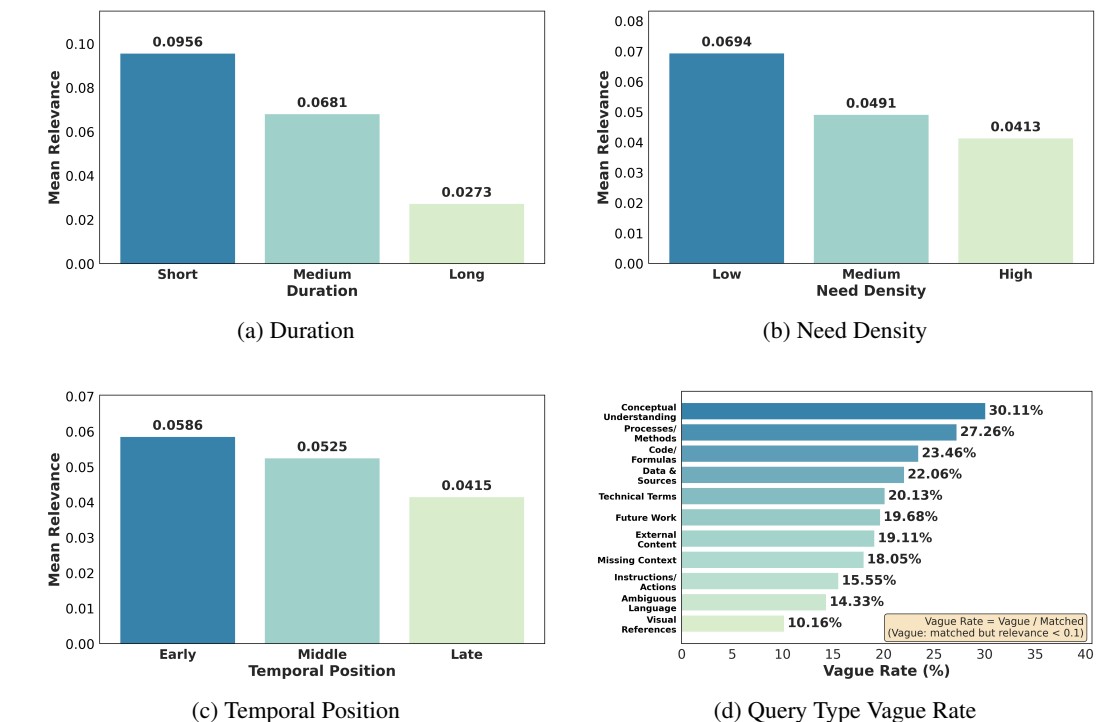

(a) Duration

(b) Need Density

(c) Temporal Position

(d) Query Type Vague Rate

Figure 11: Contextual factors affecting retrieval quality. (a) Duration: Short videos achieve the highest relevance (0.096), while long videos show the lowest (0.027). (b) Need Density: Low-density videos perform best (0.069), while high-density videos perform worst (0.041). (c) Temporal Position: Queries from early video segments achieve better relevance (0.059) compared to late segments (0.042). (d) Query Type Vague Rate: Conceptual Understanding and Processes/Methods show the highest vague rates (30.11% and 27.26%), indicating these query types require more contextual information. These systematic patterns indicate that contextual factors have consistent, predictable effects on retrieval quality, suggesting that incorporating contextual information could improve JIR system performance.

becomes vague without knowing it refers to a geometric transformation in a sphere surface area calculation lecture.

In summary, baseline relevance (mean: 0.061, max: 0.181) confirms user information need prediction bottleneck. Oracle relevance (mean: 0.212, max: 0.390) shows that retrieval systems need enhancement even with perfect queries. This dual bottleneck indicates future JIR systems should focus on both improving query generation coverage (addressing semantic mismatch) and developing contextualized retrieval systems that incorporate query context.

## K  NEED TYPE ANALYSIS

We analyzed performance across user information need types. Table 8 and Figure 12 shows the breakdown. A *challenge score* combines recall, relevance, and timeliness (higher scores indicate more challenging types):

*Challenge Score* is defined as:

$$\text{Challenge Score} = (1 - \text{Recall}) \times 0.4 + (1 - R_{\text{relevance}}) \times 0.3 + (1 - R_{\text{timeliness}}) \times 0.3$$

Higher scores indicate more challenging need types.

The challenge score distribution shows a clear pattern: needs that require contextual reconstruction, implicit inference, or multimodal understanding are consistently the hardest, whereas lexically explicit or procedure-oriented needs are the easiest. This reinforces the dual bottleneck finding that both semantic grounding in generation and multimodal/contextual recovery in retrieval remain major limitations of current JIR systems.

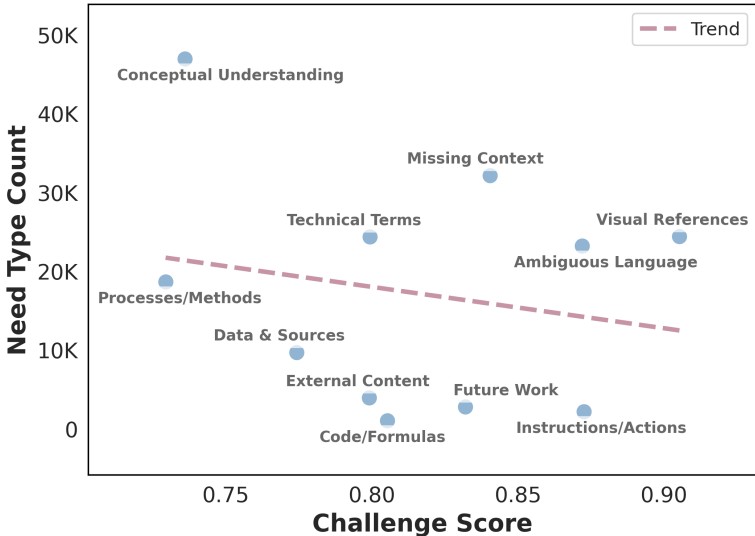

Figure 12: Relationship between challenge score and query count across information need types. Challenge score is computed as a weighted combination of recall, relevance, and timeliness failures. Visual References, Instructions/Actions, and Ambiguous Language exhibit the highest challenge scores, indicating these types pose the greatest difficulty for JIR systems. The trend line reveals the overall relationship between challenge level and frequency of occurrence.

Table 8: Need Type Distribution and Performance

| Need Type | % | Recall | $R_{\text{relevance}}$ | $R_{\text{timeliness}}$ | Challenge Score |
|---|---|---|---|---|---|
| Conceptual Understanding | 24.8% | 0.474 | 0.063 | 0.247 | 0.736 |
| Missing Context | 17.0% | 0.283 | 0.038 | 0.154 | 0.840 |
| Technical Terms | 12.9% | 0.367 | 0.062 | 0.179 | 0.799 |
| Visual References | 12.9% | 0.171 | 0.027 | 0.087 | 0.905 |
| Ambiguous Language | 12.3% | 0.227 | 0.031 | 0.124 | 0.872 |
| Processes/Methods | 9.9% | 0.483 | 0.077 | 0.258 | 0.729 |
| Data & Sources | 5.1% | 0.399 | 0.074 | 0.220 | 0.774 |
| External Content | 2.1% | 0.356 | 0.063 | 0.195 | 0.799 |
| Future Work | 1.5% | 0.306 | 0.043 | 0.152 | 0.832 |
| Instructions/Actions | 1.2% | 0.212 | 0.023 | 0.142 | 0.873 |
| Code/Formulas | 0.6% | 0.345 | 0.041 | 0.189 | 0.805 |

## L  MODEL SCALE EFFECTS

We analyze scaling effects across generative models for predicting user information needs. By investigating 5 open-source models with known sizes (Qwen3-4B, Phi-4-14B, Llama-3-70B, Cohere-111B, DeepSeek-V3-671B), we found clear scaling laws (as shown in Figure 13): larger models achieve better recall, relevance, and timeliness. Cross-metric correlations are high (recall-relevance: 0.969, recall-timeliness: 0.994), indicating consistent scaling patterns across evaluation dimensions. The results indicate that model size is a key factor determining performance.

**Small Model Precision Anomaly Explained:** Small models (Qwen3-4B, Phi-4) show higher precision than some larger models. This stems from fewer total predictions. Examining matched candidates (total_predictions × precision) shows absolute number of correctly predicted needs scales with model size. Small models generate fewer queries but a higher proportion match ground truth, leading to higher precision. Precision evaluates what proportion of predictions are useful, balancing coverage (recall) with quality (precision).

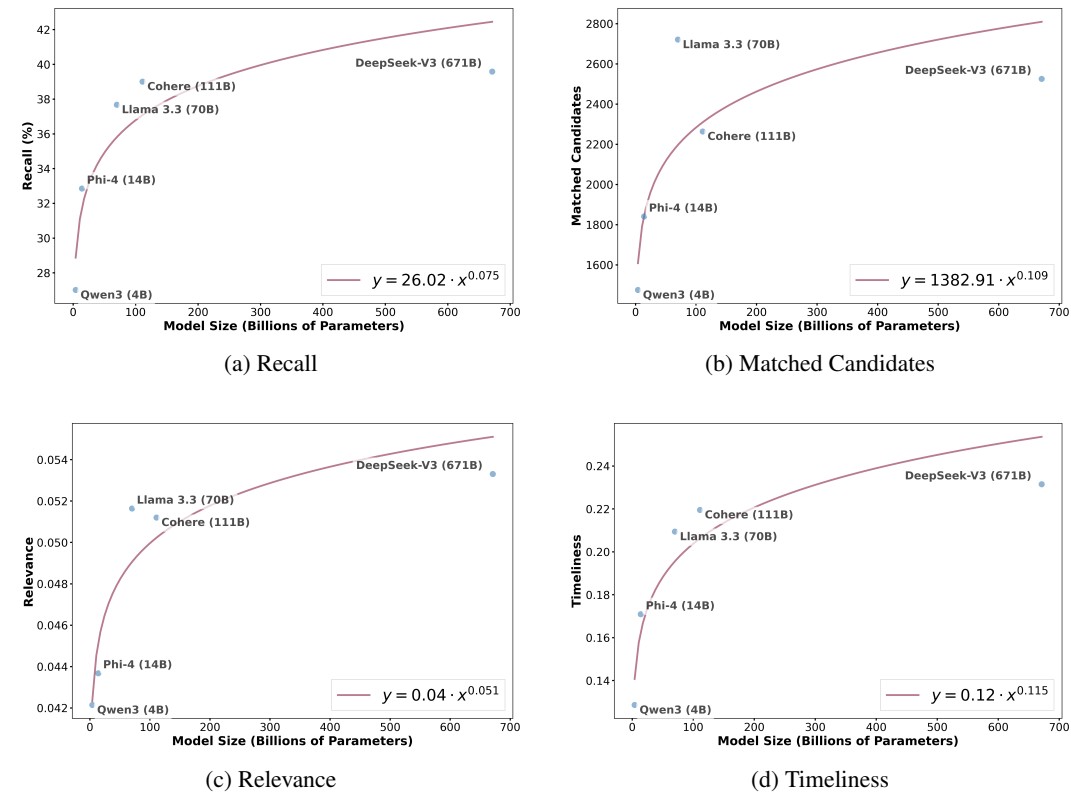

(a) Recall

(b) Matched Candidates

(c) Relevance

(d) Timeliness

Figure 13: Scaling laws for model size vs. performance metrics across open-source models. All four metrics (Recall, Matched Candidates, Relevance, and Timeliness) exhibit positive correlations with model size, demonstrating that larger models achieve better performance. The matched candidates plot shows that while small models may have higher precision, larger models predict more correct information needs in absolute terms.

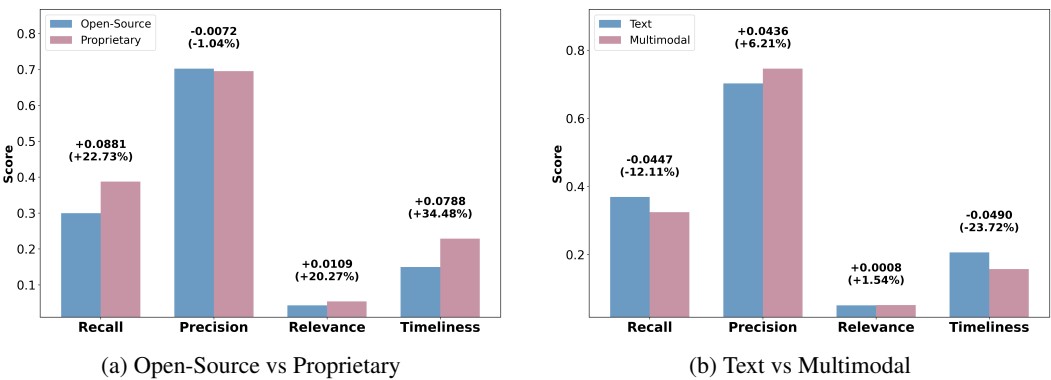

(a) Open-Source vs Proprietary

(b) Text vs Multimodal

Figure 14: Performance comparison across model categories. (a) Proprietary models consistently outperform open-source models across all metrics, with the performance gap explained by scaling laws (proprietary models are typically larger). (b) Text models outperform multimodal models, particularly in recall and timeliness, indicating that visual information may introduce noise in time-sensitive information retrieval tasks.

## M  PROPRIETARY VS OPEN-SOURCE MODEL DIFFERENCES

Proprietary models on average outperform open-source models across most metrics: recall (+22.73%), relevance (+20.27%), and timeliness (+34.48%) (see Figure 14a). This difference is primarily explained by model size: proprietary models are typically larger, and our scaling law analysis confirms larger models perform better. Precision shows a smaller gap due to small models' conservative pre-

dictions yielding higher precision, but absolute performance (matched candidates) favors proprietary models.

## N  TEXT-ONLY VS MULTIMODAL MODEL COMPARISONS

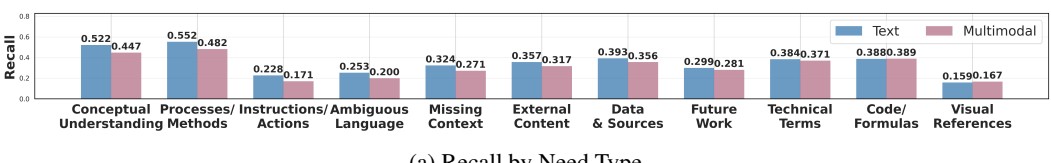

(a) Recall by Need Type

(b) Timeliness by Need Type

Figure 15: Performance comparison between text and multimodal models across different information need types. (a) Recall performance shows multimodal models consistently underperform text models across most need types, with the largest gaps in Process/Method and Instructional types. (b) Timeliness performance reveals similar patterns, with multimodal models showing reduced time matching accuracy, particularly for process-oriented and instructional content types.

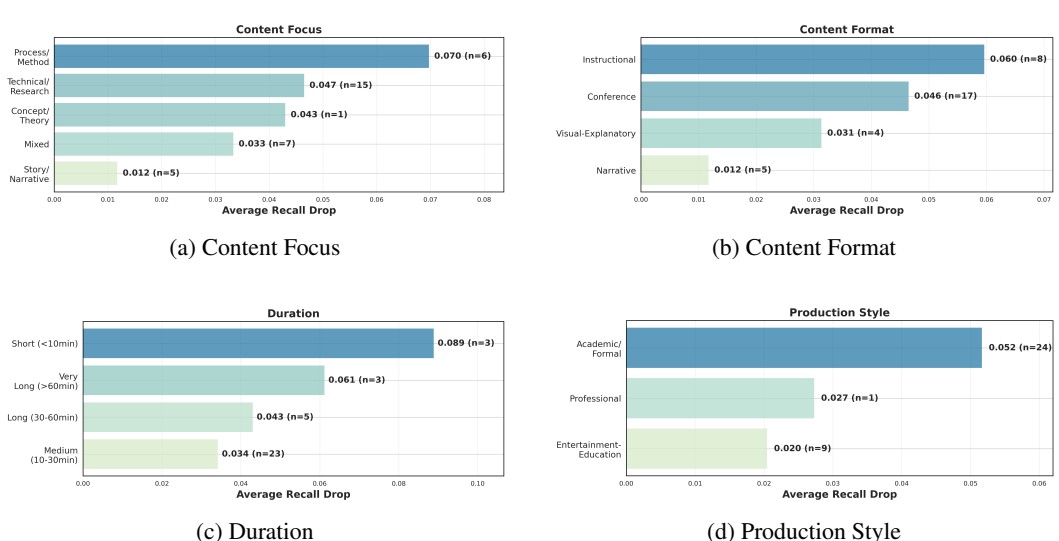

(a) Content Focus

(b) Content Format

(c) Duration

(d) Production Style

Figure 16: Performance drop analysis for multimodal models across video dimensions. (a) Content Focus: Process/Method videos show the largest drop (0.0697), while Story/Narrative videos show minimal drop (0.0117). (b) Content Format: Instructional videos exhibit higher drop (0.0642) compared to Narrative videos (0.0117). (c) Duration: Short videos ($<$10min) show the largest drop (0.0889), while medium-length videos (10-30min) show the smallest drop (0.0342). (d) Production Style: Academic/Formal videos show higher drop (0.0517) than Entertainment-Education videos (0.0204). These patterns indicate that multimodal models struggle more with process-oriented, instructional, short, and formal content.

The comparison overview between text-only models and multimodal models is shown in Figure 14b. We analyze why multimodal models show lower recall (0.412 vs 0.429 text-only average). Recall drop decomposed into: (1) temporal matching—whether predicted queries align with ground truth in time; and (2) semantic similarity—whether predicted queries semantically match ground truth. Temporal matching degradation contributes 86.79% to recall drop, semantic similarity only 13.21%.

**When Visual Information Helps:** Narrative content (Story/Narrative) and entertainment-education videos show smaller recall drops (0.0117 and 0.0204), benefiting from consistent visual context.

**When Visual Information Introduces Noise:** Process/method-oriented instructional content and academic/formal styles show larger performance degradation (0.0697 and 0.0517). Text-only models can reason through textual content; multimodal models struggle due to rapidly changing visual elements, causing temporal misalignment. Future JIR systems need better visual temporal localization, especially for real-time video streams.

Finer-grained investigations are shown in Figure 15 and 16.

## O   EVALUATION METHODOLOGY AND VALIDATION

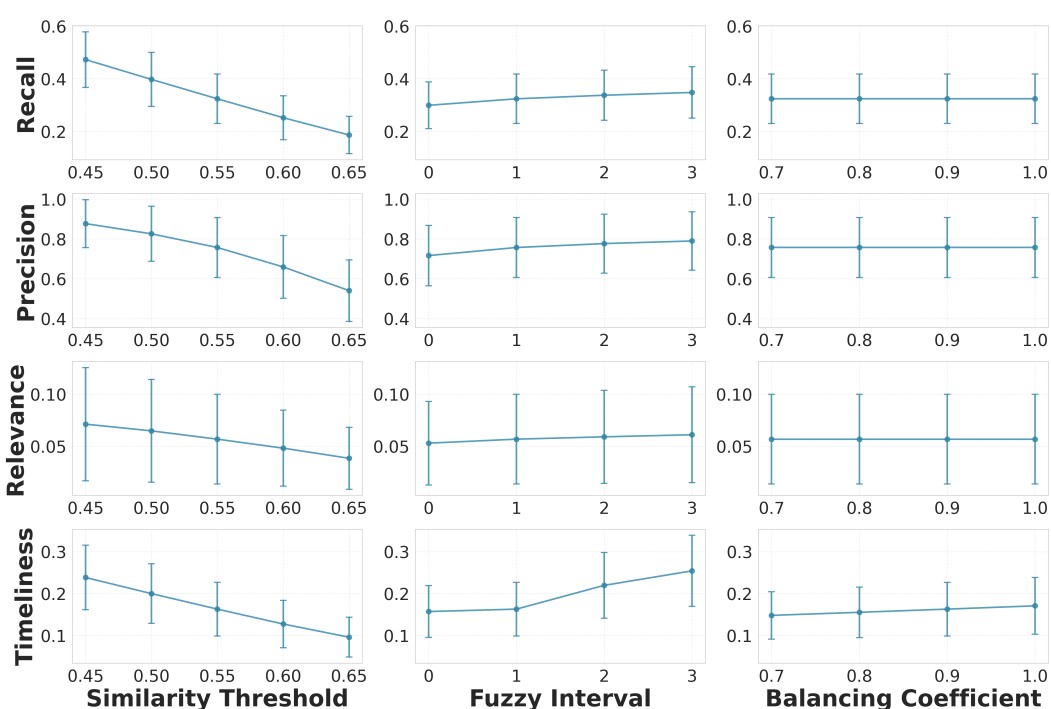

Figure 17:  Parameter sensitivity analysis showing how evaluation metrics (Recall, Precision, Relevance, Timeliness) vary across different parameter values. Each row represents a metric, and each column represents a parameter (Similarity Threshold, Fuzzy Interval, Balancing Coefficient). Error bars show mean ± std across models. Similarity threshold has the strongest impact on all metrics, with higher thresholds leading to lower scores. Fuzzy interval primarily affects timeliness, while balancing coefficient only influences timeliness. The current parameter choices (0.55, 1, 0.9) represent balanced selections that prioritize evaluation reliability over maximizing individual metrics.

**Evaluation Framework and Parameter Selection:** Our evaluation follows a two-stage process. First, we perform matching between ground truth JIR instances and candidate JIR instances generated by systems using semantic similarity and temporal windows. Second, based on matched pairs, we compute four metrics: Recall (coverage of ground truth needs), Precision (effectiveness of generated needs), $R_{relevance}$ (retrieval quality using nDCG), and $R_{timeliness}$ (temporal accuracy). Since matching directly affects all four metrics' calculations, we conducted sensitivity analysis across hyperparameter selection in the matching algorithm: semantic similarity threshold (0.45–0.65), time fuzzy interval (0–3), and temporal balancing coefficient (0.7–1.0). Table 9 and Figure 17 shows the results.

Current parameters (semantic similarity threshold 0.55, time fuzzy interval 1, balancing coefficient 0.9) balance multiple objectives: semantic similarity threshold mainly balances recall and precision; time fuzzy interval provides temporal flexibility without excessive noise; balancing coefficient 0.9 slightly favors start time over end time (reflecting that accurately predicting when an information need arises is practically more critical than predicting when it ends). **We want to emphasize that model rankings remain highly stable (Spearman correlation 0.956) across all parameter variations, confirming that parameter selection does not bias model comparisons (see Figure 18).**

**Benchmark Metrics' Alignment with User Satisfaction:** We conducted a user study to validate the simulated need distributions with real audiences. Participants watched videos and wrote down their

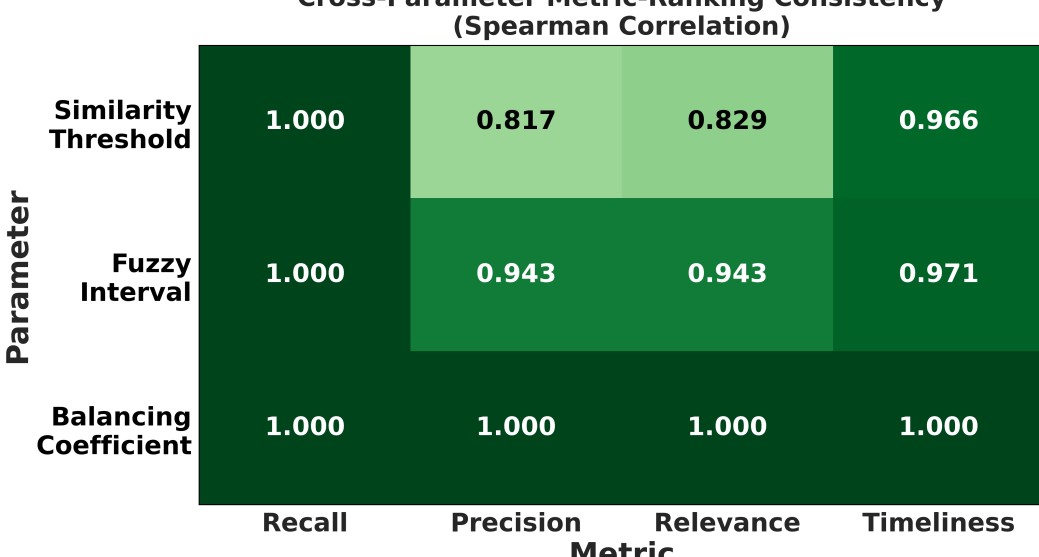

Figure 18: Ranking stability analysis showing Spearman correlation of model rankings across different parameter values. The heatmap demonstrates high ranking consistency across all parameters and metrics, with Recall and Timeliness achieving perfect stability (1.000). This indicates that parameter choices do not significantly bias model comparisons, validating the robustness of the evaluation framework.

Table 9: Parameter Sensitivity Analysis

| Value | Recall | Precision | Relevance | Timeliness |
|---|---|---|---|---|
| **Semantic Similarity Threshold** | | | | |
| 0.45 | 0.473 | 0.877 | 0.071 | 0.238 |
| 0.55 | 0.324 | 0.757 | 0.057 | 0.163 |
| 0.65 | 0.186 | 0.540 | 0.039 | 0.096 |
| **Fuzzy Interval** | | | | |
| 0 | 0.299 | 0.717 | 0.053 | 0.157 |
| 1 | 0.324 | 0.757 | 0.057 | 0.163 |
| 3 | 0.348 | 0.790 | 0.061 | 0.254 |
| **Balancing Coefficient** | | | | |
| 0.7 | 0.324 | 0.757 | 0.057 | 0.148 |
| 0.9 | 0.324 | 0.757 | 0.057 | 0.163 |
| 1.0 | 0.324 | 0.757 | 0.057 | 0.171 |

actual information needs, resulting in 83 recorded needs across nine of our videos. We then compared these needs to both the LLM-generated needs and the full benchmark dataset (LLM-generated + human-simulated needs). A match was counted if a need from our dataset occurred within a time window of ±3 sentences and had a cosine similarity above 0.55. Of the 83 user study needs, 62.7% matched needs from the full benchmark dataset. 48.2% matched needs from the LLM-generated subset alone. The user study shows that our dataset covers a significant portion of actual user information needs, demonstrating that the simulated distributions are broadly representative of real audiences.

**Benchmark Data Reliability:** Our dataset construction uses a multi-entity collaborative approach with voting mechanisms to generate information needs, mitigating subjectivity through consensus. To validate quality, human annotators independently rated the quality of each information need's

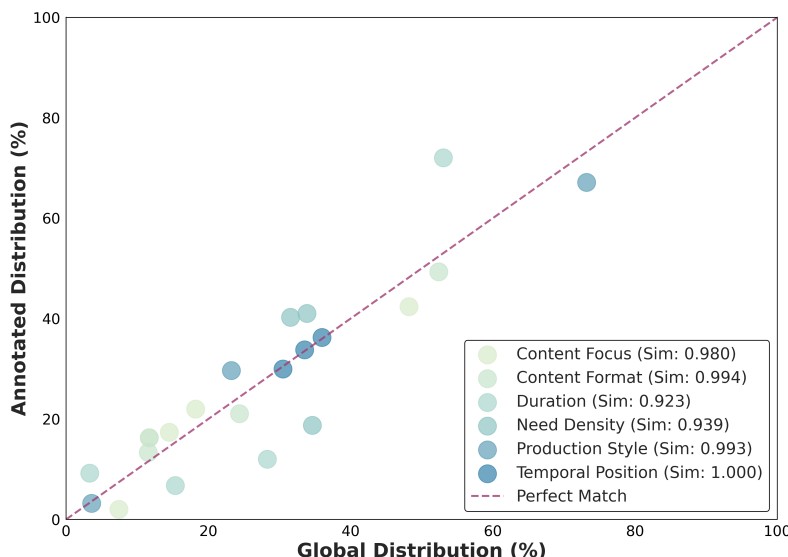

Figure 19: Distribution similarity between annotated samples and global dataset across 8 dimensions. Each point represents a category, colored by dimension. Points near the diagonal line indicate high similarity. All dimensions show similarity scores >0.9 (with Temporal Position achieving perfect match at 1.000), demonstrating that the annotated samples are randomly sampled and representative of the entire dataset, validating the sufficiency of the annotation coverage.

components: Reason (why the need arises), Need (the information gap), and Question (the query formulation). Human validation scores (Reason: 3.26, Need: 3.77, Question: 3.80, all on 0–4 scale) indicate high quality of the dataset. Figure 19 shows that the samples with quality validation can represent the global well in all dimensions. The user study correlation (61%)—measuring alignment between user needs and our benchmark oracle needs—provides evidence that benchmark distributions approximate real-world user behavior.

## P  DATASET SCALE AND COVERAGE

The JIR-Arena benchmark dataset (34 scenes, 831 minutes) focuses on academic lectures and conference talks. We chose to focus on such data for constructing the very first JIR benchmark datasets because they represent high-value real-world JIR scenarios where information gaps are intensive and addressing the information need is necessary and challenging. Scene selection prioritized high-view videos to ensure content quality and relevance. High view counts indicate high demand—these are scenarios where the most users are likely to encounter information gaps requiring just-in-time information recommendation. As with any system deployment, we first cover the most typical cases before expanding to edge cases.

The static knowledge bases (Wikipedia, textbooks, arXiv) were chosen to ensure reproducible and comparable evaluation across different time periods and models by controlling the retrieval source as a variable—this is an essential requirement for benchmark development to ensure result reproducibility. In lecture and conference scenarios, Wikipedia, textbooks, and arXiv provide high credibility and broad coverage. Real-world deployed systems can of course leverage more advanced search APIs.

We want to clarify that "comprehensive" in the title refers to the **evaluation framework's comprehensiveness** (multi-dimensional metrics: Recall, Precision, Relevance, Timeliness; diverse model comparisons: 21 models including proprietary, open-source, text-only, and multimodal).

As the **first benchmark** for JIR, we prioritized **quality control**—including multi-entity collaborative annotation, multi-stage verification, and final human auditing—and **reproducibility** over scale. Building a reliable foundation for the community required substantial resources, and given the tradeoff between breadth and depth, we deliberately invested in depth to ensure that the benchmark remained rigorous, interpretable, and trustworthy. We acknowledge that JIR applications extend beyond academic settings (e.g., meetings, coding sessions, medical consultations, daily tasks). The benchmark dataset's extensible design enables community contributions and future expansion. Indeed, the proposed new evaluation methodology, quality control processes, and metrics are all general and

thus can be directly used by the research community to leverage our initial foundational work to further expand JIR-Arena by including many more datasets in all those other application domains of JIR.

## Q    BROADER IMPACTS

While it is unlikely that there are direct negative impacts or harms that arise from our proposed benchmark, it is worth considering the potential second-order negative effects of JIR systems. One potential negative effect is the lack of friction and its effect on learning. Namely, individuals may learn better when it is harder to answer a question, or when they have to come up with the question themselves. If a system always provides them the "right question to ask," then they may become reliant on the system for their exploration. Because the overall space of just-in-time recommendation agents is fairly nascent, the effects of these systems on learning or exploration remains an open research question.

