# OpenReview forum: "JIR-Arena: The First Comprehensive Benchmark Dataset for Just-in-time Information Recommendation"
_ICLR.cc/2026/Conference — Submitted to ICLR 2026_

### Official Review · Reviewer_JomG · 2025-10-31

**Soundness:** 3
**Presentation:** 3
**Contribution:** 3
**Rating:** 4
**Confidence:** 3

**Summary:**

This paper presents JIR-Arena, the first comprehensive benchmark dataset for evaluating Just-in-Time Information Recommendation (JIR) systems. The JIR task aims to capture and satisfy users' immediate information needs in information-intensive scenarios (e.g., attending lectures, watching tutorials). The authors rigorously formalize the JIR task as a Partially Observable Markov Decision Process (POMDP) and design a multi-dimensional evaluation framework encompassing Precision, Recall, content Relevance (R_relevance), and Timeliness (R_timeliness).

The JIR-Arena dataset comprises 34 multimodal scenes (primarily academic lectures and conference talks) totaling 831 minutes. The dataset construction employs a multi-agent collaborative approach, combining multiple large language models (GPT-4O, DeepSeek-V3) with human annotators, using voting mechanisms to simulate the distribution of user information needs. The authors implement baseline JIR systems and conduct evaluations, revealing that current systems can identify user needs with reasonable precision but perform poorly in terms of recall and information retrieval quality, indicating that the retrieval component is the primary bottleneck and pointing the direction for future research.

**Strengths:**

**Pioneering Contribution and Theoretical Framework.** This is the first work to formally define the JIR task and provide a standardized evaluation framework, filling a long-standing gap in the field. The formalization of JIR as a POMDP framework establishes a solid theoretical foundation for subsequent research, enabling precise definition and systematic study of JIR tasks. This represents a crucial contribution for an emerging research direction.

**Innovative Data Collection Methodology.** The paper employs a multi-entity collaborative strategy to construct the dataset, demonstrating methodological innovation. By combining multiple large language models with human annotators and utilizing voting mechanisms to simulate user information needs, it appropriately addresses the inherent subjectivity and diversity challenges in need annotation. The three-layer retrieval verification pipeline (traditional IR → LLM quality check → human verification) exhibits a clear design rationale and fully reflects awareness of data quality control.

**Well-Designed Evaluation Framework.** The proposed evaluation dimensions (Precision, Recall, Relevance, Timeliness) are comprehensive and well-suited to the characteristics of JIR tasks. The many-to-many matching mechanism accounts for the decomposability of information needs, which is a key distinction from traditional question-answering systems. The use of nDCG to evaluate retrieval quality, adoption of Gaussian kernel functions for temporal matching, and introduction of likelihood scores as weights all align well with task requirements. Although specific implementation details require refinement, the overall framework provides an excellent starting point for JIR evaluation.

**Systematic Baseline Experimental Study.** The paper tests multiple mainstream large models (GPT-4O, DeepSeek-V3, Claude-3-7, Gemini-2.0-Flash) as well as small models suitable for edge devices (Phi-4, Qwen3-4B), comparing text-only versus multimodal models and different retrieval methods (BM25, Dense Retriever, Reranker). These experiments provide valuable performance baselines and initial insights to the community, clearly identifying retrieval systems as the current primary bottleneck.

**Weaknesses:**

### 3.1 Reproducibility Issues

The paper explicitly states that the complete static knowledge bases cannot be released due to space limitations. This is a significant issue for a benchmark dataset, as without access to the same knowledge bases, other researchers will be unable to fairly reproduce experiments or compare system performance. It is recommended that, prior to formal publication, the authors provide an access solution for the knowledge bases (e.g., hosting on Zenodo or Hugging Face Datasets), or at minimum provide complete scripts and indexing methods for constructing the knowledge bases.

### 3.2 Insufficient Depth of Experimental Analysis

The paper shows that retrieval relevance performance is suboptimal but lacks in-depth analysis of the underlying causes. Is the issue with query generation quality, indexing granularity (5 sentences/chunk), knowledge base coverage, or the retrieval model itself? It is suggested to conduct an oracle study (directly using ground truth questions for retrieval) to pinpoint the bottleneck. Additionally, the counterintuitive phenomenon that multimodal models exhibit lower recall than text-only models (0.412 vs 0.429) requires deeper analysis, such as identifying in which scenarios visual information enhances performance and in which scenarios it may introduce noise.

### 3.3 Evaluation Details Require Clarification

While the evaluation framework design is sound in concept, the many-to-many matching algorithm lacks mathematical formulation or pseudocode definition. It is unclear how to handle cases when one ground truth matches multiple predictions and how to avoid duplicate counting. The selection of key parameters (such as similarity threshold 0.55, temporal balancing coefficient 0.9, etc.) also lacks sufficient justification and explanation. It is recommended to clearly define the matching logic and conduct parameter sensitivity analysis.

### 3.4 Limited Dataset Scale and Coverage

The scale of 34 scenes is relatively small, and coverage is limited to academic scenarios (lectures and conferences). The application scenarios for JIR should actually be much broader, with scenarios such as daily work, programming tutorials, and product usage also holding significant value. Furthermore, while the paper defines 11 types of information needs, it does not report the distribution of each type or fine-grained performance. It is suggested to supplement this content to identify which types are most challenging.

**Questions:**

1. **Retrieval Bottleneck Localization:** Retrieval relevance is the primary bottleneck. Could you conduct an oracle study? That is, directly use ground truth questions for retrieval to observe performance, which would help determine whether the issue lies in the need generation stage or the retrieval system itself.

2. **Multimodal Anomaly:** Why do multimodal models exhibit lower recall than text-only models (0.412 vs 0.429)? The paper's explanation is overly brief and lacks quantitative analysis. Could you analyze in which scenarios visual information enhances performance and in which scenarios it may introduce noise?

3. **Evaluation Parameter Selection Rationale:** How were parameters such as similarity threshold 0.55, deduplication threshold 0.75, and temporal balancing coefficient 0.9 selected? Was parameter sensitivity analysis conducted?

4. **Need Type Distribution Analysis:** The paper defines 11 types of information needs but does not report the distribution or performance for each type. Could this content be supplemented? Which types are most challenging?

5. **Annotation Coverage Explanation:** Human verification covers only 272 samples (approximately 20%). Is this proportion sufficient? How is the quality of the remaining samples ensured?

---

> ### Author Response · Authors · 2025-11-21
>
> We thank Reviewer JomG for the detailed feedback. We are grateful for the reviewer's recognition of this work as **the first work to formally define the JIR task**, establishing a **solid theoretical foundation** and providing **valuable performance baselines** for the community. The reviewer's appreciation of the **innovative data collection methodology**, **well-designed evaluation framework**, and **systematic baseline experimental study** is encouraging. The reviewer raised questions about reproducibility, more experimental analysis (oracle study, multimodal performance, need type distribution analysis), evaluation details (matching algorithm, parameter selection), and dataset scale. We have conducted comprehensive additional experiments to address all concerns. **All new results have been incorporated into the manuscript update**, and **all analysis scripts are included in the supplementary code**. Detailed responses follow below with supporting tables.

---

> > ### Author Response · Authors · 2025-11-21
> >
> > ## Oracle Analysis for Baseline Bottleneck Localization
> >
> > > Retrieval relevance is the primary bottleneck. Could you conduct an oracle study? That is, directly use ground truth questions for retrieval to observe performance, which would help determine whether the issue lies in the need generation stage or the retrieval system itself.
> >
> > A JIR system consists of two main components: (1) a **generative model** that predicts user information needs and generates queries, and (2) a **retrieval model** that retrieves relevant documents using these queries. To determine whether the bottleneck lies in query generation or retrieval systems, we conducted an oracle study where we replaced the generative model's predicted queries with **ground truth queries** from the dataset (the oracle queries). This "Oracle Generator" uses perfect queries (Recall: 1.000, Precision: 1.000) to test the retrieval system's upper bound performance. Table 1 in the general comment shows Oracle Generator performance (Relevance: 0.215/0.195/0.274 for O/D/R) vs baseline models that use generative model predictions. We average the baselines' results and present them in the following table:
> >
> > **Table: Oracle vs Baseline Performance Comparison**
> >
> > | Metric        | Oracle | Baseline | Improvement | Improvement % |
> > | ------------- | ------ | -------- | ----------- | ------------- |
> > | Recall        | 1.000  | 0.347    | +0.653      | +188.39%      |
> > | Precision     | 1.000  | 0.724    | +0.276      | +38.04%       |
> > | Relevance (O) | 0.215  | 0.051    | +0.164      | +319.23%      |
> > | Timeliness    | 1.000  | 0.182    | +0.818      | +449.75%      |
> >
> > The oracle study reveals that **both the user information need predictor and the retrieval system are bottlenecks**:
> >
> > 1. **User Information Need Predictor Bottleneck** (Primary): Baseline systems achieve only 0.347 average recall, meaning 65.3% of user information needs are not predicted. Error analysis shows semantic mismatch causes 62.91% of failures—predicted queries fail to semantically match oracle needs. Critically, even for high-likelihood information needs (likelihood score ≥8.5 over 10, indicating high probability of being raised by users based on our multi-entity annotation process), the system fails to achieve adequate coverage; recall remains poor at only 41.38%, far below acceptable levels. This explains why baseline relevance is low (mean: 0.061): insufficient query prediction leads to too few queries to retrieve relevant documents.
> >
> > 2. **Retrieval System Bottleneck** (Secondary but Significant): Even with perfect ground truth queries, oracle relevance remains low (0.215 OpenSearch, 0.274 Reranked), far below ideal. The improvement from OpenSearch to Reranked (+36%) shows that retrieval models matter, but even the best retrieval system (Reranked) achieves only 0.274 relevance with perfect queries. This suggests the limitation is not solely in query generation—retrieval systems themselves need improvement. One of the key reasons, as noted in the original paper, is that queries are highly context-dependent; when used without contextual information, they appear vague, leading to low retrieval quality even with oracle queries. Here, contextual information includes content context (what topic is being discussed), temporal context (where in the timeline), visual context (diagrams, animations shown), domain context (technical field), and methodological context (specific methods demonstrated). For example, an oracle query like "What is the 'subtle shift in perspective'?" becomes vague without knowing it refers to a geometric transformation in a sphere surface area calculation lecture.
> >
> > In summary, baseline relevance (mean: 0.061, max: 0.181) confirms the user information need prediction bottleneck. Oracle relevance (mean: 0.212, max: 0.390) shows that retrieval systems need enhancement even with perfect queries. This dual bottleneck indicates future JIR research should focus on both improving query generation coverage (addressing semantic mismatch) and developing contextualized retrieval systems that incorporate query context.

---

> > ### Author Response · Authors · 2025-11-21
> >
> > ## Text-Only vs Multimodal Model Comparisons
> >
> > > Why do multimodal models exhibit lower recall than text-only models (0.412 vs 0.429)? The paper's explanation is overly brief and lacks quantitative analysis. Could you analyze in which scenarios visual information enhances performance and in which scenarios it may introduce noise?
> >
> > We analyze why multimodal models show lower recall (0.412 vs 0.429 text-only average). To understand the recall drop, we decompose it into two factors based on our evaluation framework: (1) temporal matching—whether predicted queries align with ground truth queries in time; and (2) semantic similarity—whether predicted queries semantically match ground truth queries. We find that temporal matching accuracy degradation contributes 86.79% to the recall drop, while semantic similarity differences contribute only 13.21%. This indicates that temporal alignment is the primary bottleneck—multimodal models struggle more significantly with aligning predicted information needs to exact temporal positions in videos (start/end timestamps). If we look closer:
> >
> > **When Visual Information Helps**: We analyzed performance across different video characteristics. Narrative content (Story/Narrative focus, e.g., Numberphile's conversational storytelling format) and entertainment-education styles (e.g., videos combining education with entertainment elements) show better performance, with smaller recall drops (0.0117 and 0.0204, respectively). These content types benefit from a relatively consistent visual context that naturally complements narrative flow.
> >
> > **When Visual Information Introduces Noise**: Process/method-oriented instructional content (Process/Method focus, e.g., Khan Academy's step-by-step teaching format) and academic/formal styles (e.g., structured academic lectures) show larger performance degradation (0.0697 and 0.0517 drops, respectively). In these scenarios, text-only models can effectively reason through textual content to predict user information needs. On the other hand, multimodal models perform worse overall because visual elements change rapidly in these content types (e.g., slides switching, equations appearing/disappearing, diagrams updating), making temporal localization more challenging. The resulting temporal misalignment outweighs potential benefits from additional visual information.
> >
> > These findings suggest that future JIR systems need better visual temporal localization, especially for real-time video streams. Visual temporal alignment is the key bottleneck when visual content changes rapidly. Therefore, our results also suggest that future work should focus on stream-based video processing that enables precise, frame-level synchronization of user information need estimation as the video evolves.

---

> > ### Author Response · Authors · 2025-11-21
> >
> > ## Evaluation Framework and Parameter Selection
> >
> > > How were parameters such as similarity threshold 0.55, deduplication threshold 0.75, and temporal balancing coefficient 0.9 selected? Was parameter sensitivity analysis conducted?
> >
> > Our evaluation follows a two-stage process. First, we perform matching between ground truth JIR instances and candidate JIR instances generated by systems using semantic similarity and temporal windows. Second, based on matched pairs, we compute four metrics: Recall (coverage of ground truth needs), Precision (effectiveness of generated needs), R_relevance (retrieval quality using nDCG), and R_timeliness (temporal accuracy). Since matching directly affects all four metrics' calculations, we conducted sensitivity analysis across hyperparameter selection in the matching algorithm: semantic similarity threshold (0.45-0.65), time fuzzy interval (0-3), and temporal balancing coefficient (0.7-1.0). The following table shows the results. Current parameters (semantic similarity threshold 0.55, time fuzzy interval 1, balancing coefficient 0.9) balance multiple objectives: semantic similarity threshold mainly balances recall and precision; time fuzzy interval provides temporal flexibility without excessive noise; balancing coefficient 0.9 slightly favors start time over end time (reflecting that accurately predicting when an information need arises is practically more critical than predicting when it ends). **Model rankings remain highly stable (Spearman correlation 0.956) across all parameter variations, confirming that parameter selection does not bias model comparisons**.
> >
> > **Table: Parameter Sensitivity Analysis**
> >
> > | Value                             | Recall | Precision | Relevance | Timeliness |
> > | --------------------------------- | ------ | --------- | --------- | ---------- |
> > | **Semantic Similarity Threshold** |        |           |           |            |
> > | 0.45                              | 0.473  | 0.877     | 0.071     | 0.238      |
> > | 0.55                              | 0.324  | 0.757     | 0.057     | 0.163      |
> > | 0.65                              | 0.186  | 0.540     | 0.039     | 0.096      |
> > | **Fuzzy Interval**                |        |           |           |            |
> > | 0                                 | 0.299  | 0.717     | 0.053     | 0.157      |
> > | 1                                 | 0.324  | 0.757     | 0.057     | 0.163      |
> > | 3                                 | 0.348  | 0.790     | 0.061     | 0.254      |
> > | **Balancing Coefficient**         |        |           |           |            |
> > | 0.7                               | 0.324  | 0.757     | 0.057     | 0.148      |
> > | 0.9                               | 0.324  | 0.757     | 0.057     | 0.163      |
> > | 1.0                               | 0.324  | 0.757     | 0.057     | 0.171      |

---

> > ### Author Response · Authors · 2025-11-21
> >
> > ## Need Type Analysis
> >
> > > The paper defines 11 types of information needs but does not report the distribution or performance for each type. Could this content be supplemented? Which types are most challenging?
> >
> > We analyzed finer-grained performance over user information need types. The following table shows the breakdown. We define a challenge score that combines recall, relevance, and timeliness (higher scores indicate more challenging types) to identify which need types are most difficult for JIR systems. Performance varies drastically: Visual References and Ambiguous Language are most challenging, while Processes/Methods and Conceptual Understanding perform best. This identifies specific failure modes: visual and ambiguous queries are particularly difficult, while process-oriented and conceptual queries are more tractable.
> >
> > **Table: Need Type Distribution and Performance**
> >
> > | Need Type                | %     | Recall | R_relevance | R_timeliness | Challenge Score |
> > | ------------------------ | ----- | ------ | ----------- | ------------ | --------------- |
> > | Conceptual Understanding | 24.8% | 0.474  | 0.063       | 0.247        | 0.736           |
> > | Missing Context          | 17.0% | 0.283  | 0.038       | 0.154        | 0.840           |
> > | Technical Terms          | 12.9% | 0.367  | 0.062       | 0.179        | 0.799           |
> > | Visual References        | 12.9% | 0.171  | 0.027       | 0.087        | 0.905           |
> > | Ambiguous Language       | 12.3% | 0.227  | 0.031       | 0.124        | 0.872           |
> > | Processes/Methods        | 9.9%  | 0.483  | 0.077       | 0.258        | 0.729           |
> > | Data & Sources           | 5.1%  | 0.399  | 0.074       | 0.220        | 0.774           |
> > | External Content         | 2.1%  | 0.356  | 0.063       | 0.195        | 0.799           |
> > | Future Work              | 1.5%  | 0.306  | 0.043       | 0.152        | 0.832           |
> > | Instructions/Actions     | 1.2%  | 0.212  | 0.023       | 0.142        | 0.873           |
> > | Code/Formulas            | 0.6%  | 0.345  | 0.041       | 0.189        | 0.805           |
> >
> > *Challenge Score = (1 - Recall) × 0.4 + (1 - R_relevance) × 0.3 + (1 - R_timeliness) × 0.3, where higher scores indicate more challenging types.*

---

> > ### Author Response · Authors · 2025-11-21
> >
> > ## Annotation Coverage and Quality Assurance
> >
> > > Human verification covers only 272 samples (approximately 20%). Is this proportion sufficient? How is the quality of the remaining samples ensured?
> >
> > We performed human verification on 272 samples. To assess representativeness, we compared these samples against the full dataset across eight dimensions: Temporal Position, Content Format, Production Style, Type Distribution, Likelihood Distribution, Content Focus, Need Density, and Duration. We update the manuscript with detailed results and analysis. As shown in the following table, all dimensions show high similarity scores (>0.9)—for example, Temporal Position matches perfectly (1.000), while others range from 0.923 to 0.994. This demonstrates that the verified samples are randomly sampled and highly representative, ensuring that the quality of the remaining unverified samples is reliably reflected by the verified subset.
> >
> > **Table: Sample & Global Distribution Similarities Across Dimensions**
> >
> > | Dimension               | Similarity | Interpretation                    |
> > | ----------------------- | ---------- | --------------------------------- |
> > | Type Distribution       | 0.993      | High similarity (random sampling) |
> > | Likelihood Distribution | 0.985      | High similarity (random sampling) |
> > | Content Format          | 0.994      | High similarity (random sampling) |
> > | Content Focus           | 0.980      | High similarity (random sampling) |
> > | Production Style        | 0.993      | High similarity (random sampling) |
> > | Duration                | 0.923      | High similarity (random sampling) |
> > | Temporal Position       | 1.000      | High similarity (random sampling) |
> > | Need Density            | 0.939      | High similarity (random sampling) |

---

> > ### Author Response · Authors · 2025-11-21
> >
> > ## Remaining Concerns
> >
> > ### Reproducibility
> >
> > > The paper explicitly states that the complete static knowledge bases cannot be released due to space limitations. This is a significant issue for a benchmark dataset, as without access to the same knowledge bases, other researchers will be unable to fairly reproduce experiments or compare system performance. It is recommended that, prior to formal publication, the authors provide an access solution for the knowledge bases (e.g., hosting on Zenodo or Hugging Face Datasets), or at minimum provide complete scripts and indexing methods for constructing the knowledge bases.
> >
> > Thank you for raising the concern about reproducibility. We clarify that the statement "the complete static knowledge bases cannot be released due to space limitations" refers to the constraints of the Supplementary Material during the double-blind review period. We have included a sample of the static knowledge base in the Supplementary Material, together with the instructions to build the full knowledge base. These materials allow other researchers to reconstruct the same knowledge base locally. After the review process, we plan to publicly release the full static knowledge base.

---

> > ### Author Response · Authors · 2025-11-21
> >
> > ### More Evaluation Details
> >
> > > While the evaluation framework design is sound in concept, the many-to-many matching algorithm lacks mathematical formulation or pseudocode definition. It is unclear how to handle cases when one ground truth matches multiple predictions and how to avoid duplicate counting.
> >
> > We provide below the mathematical formulation and implementation details of our many-to-many matching algorithm, and also update that in the manuscript.
> >
> > #### Mathematical Formulation
> >
> > ##### Matching Criteria
> >
> > For a ground truth query $q_i \in Q$ and a candidate query $c_j \in C$, a match is established if and only if:
> >
> > 1. **Time Overlap:** $\max(t_{q_i}^{start}, t_{c_j}^{start}) \leq \min(t_{q_i}^{end}, t_{c_j}^{end})$
> >    where $t_{q_i}^{start} = t_{q_i}^{start} - \delta \cdot d$ and $t_{q_i}^{end} = t_{q_i}^{end} + \delta \cdot d$ (with fuzzy interval $\delta$)
> >
> > 2. **Semantic Similarity:** $\cos(\text{Embed}(q_i.\text{text}), \text{Embed}(c_j.\text{text})) \geq \theta$
> >
> > The matching function is: $M(q_i, c_j) = \mathbf{1}[\text{TimeOverlap}(q_i, c_j) \land \text{Sim}(q_i, c_j) \geq \theta]$
> >
> > ##### Pseudocode
> >
> > ```
> > for each ground truth query q_i:
> >    matches = []
> >    for each candidate query c_j:
> >       if TimeOverlap(q_i, c_j) and Sim(q_i, c_j) >= threshold:
> >          matches.append((c_j, Sim(q_i, c_j)))
> >    results[i] = sort(matches, by similarity descending)
> > ```
> >
> > #### Handling One Ground Truth Matching Multiple Predictions
> >
> > The algorithm explicitly supports this scenario. For each ground truth query $q_i$, all matching candidates are collected and stored:
> >
> > - **All matches are preserved:** `results[i] = [(c_j, s_{ij}), (c_k, s_{ik}), ...]`
> > - **Recall calculation:** A query is considered "matched" if it has **at least one** match: $\text{Recall} = |\{q_i : \exists c_j \text{ s.t. } M(q_i, c_j) = 1\}| / |Q|$
> > - **Relevance metrics:** All matched candidates are considered in nDCG calculations
> > - **Timeliness:** The best matching candidate (by combined time match score) is selected
> >
> > This design reflects realistic scenarios where multiple predictions may correctly address the same information need.
> >
> > #### Avoiding Duplicate Counting
> >
> > Duplicate counting is prevented through unique identifier-based deduplication:
> >
> > ##### Precision Calculation
> >
> > Each candidate is uniquely identified by `(question_text, start_time, end_time)`. The precision formula counts **unique** matched candidates:
> >
> > ```python
> > matched_candidates = set()
> > for matches in results.values():
> >     for candidate, _ in matches:
> >         candidate_id = (candidate["question"],
> >                        candidate.get("start_time"),
> >                        candidate.get("end_time"))
> >         matched_candidates.add(candidate_id)
> >
> > precision = len(matched_candidates) / total_candidates
> > ```
> >
> > **Mathematical formulation:**
> > $$\text{Precision} = \frac{|\{c_j : \exists q_i \text{ s.t. } M(q_i, c_j) = 1\}|}{|C|}$$
> >
> > The numerator uses set cardinality, ensuring each candidate is counted **at most once**, even if it matches multiple ground truth queries.
> >
> > ##### Recall Calculation
> >
> > Recall naturally avoids duplicates by counting queries (not matches):
> >
> > $$\text{Recall} = \frac{|\{q_i : \exists c_j \text{ s.t. } M(q_i, c_j) = 1\}|}{|Q|}$$
> >
> > Each ground truth query is counted **at most once**, regardless of how many candidates it matches.
> >
> > ##### Example
> >
> > If $q_1$ matches both $c_1$ and $c_2$, and $q_2$ also matches $c_2$:
> >
> > - **Precision:** Counts unique matched candidates $\{c_1, c_2\}$ → numerator = 2 (not 3)
> > - **Recall:** Counts queries with matches $\{q_1, q_2\}$ → numerator = 2

---

> > ### Author Response · Authors · 2025-11-21
> >
> > ### Dataset Scale and Coverage
> >
> > > The scale of 34 scenes is relatively small, and coverage is limited to academic scenarios (lectures and conferences). The application scenarios for JIR should actually be much broader, with scenarios such as daily work, programming tutorials, and product usage also holding significant value.
> >
> > We admit that as the very first benchmark dataset, JIR-Arena is relatively small, which was mainly due to the limited resources that we had. Our formalization of the problem, evaluation framework, evaluation metrics, and quality control process, however, are general, and they enable the research community to easily extend the JIR-Arena to broaden its coverage in the future.
> >
> > With our limited resources, we chose to have the JIR-Arena benchmark dataset (34 scenes, 831 minutes) focus on academic lectures and conference talks, which represent high-value JIR scenarios where information gaps are intensive and addressing the information need is necessary and challenging.
> >
> > As the **first benchmark** for JIR, we prioritized **quality control** (multi-entity collaborative annotations, multi-layer verification, final human quality check) and **reproducibility** over scale, establishing a solid foundation for the community. We acknowledge that JIR applications extend beyond academic settings (e.g., meetings, coding sessions, medical consultations, daily tasks). The benchmark dataset's extensible design enables community contributions and future expansion. We plan to broaden coverage to include these domains in future versions, building on the established framework and quality control processes.
> >
> > ---
> >
> > We welcome any further questions or clarifications. If you feel that your concerns have been fully addressed, we would greatly appreciate your consideration for an improved score.

---

> ### Author Response · Authors · 2025-11-28
>
> Dear Reviewer JomG,
>
> We sincerely appreciate your time and feedback for our work. We hope our rebuttal solves your concerns. If you have further questions regarding our work, we will be very happy to discuss them with you. Could you please reply to us so we can better address your concerns?
>
> Best Regards,
>
> Authors

---

### Official Review · Reviewer_TQLe · 2025-10-31

**Soundness:** 3
**Presentation:** 3
**Contribution:** 3
**Rating:** 6
**Confidence:** 3

**Summary:**

This paper introduces JIR-ARENA, the first comprehensive benchmark for JIR systems. The authors formalize JIR as a POMDP, propose evaluation metrics, and construct a multimodal dataset with 34 video scenes across lectures and conferences. Ground truth is generated through multi-entity collaboration with multi-round validation. The dataset includes information needs, temporal annotations, and hierarchical reference documents retrieved from static knowledge bases. Baseline experiments on several foundation models reveal low recall and poor retrieval performance, establishing initial benchmarks for future research.

**Strengths:**

1. The paper presents the first systematic formalization of the JIR task as a POMDP with well-defined components, filling a key gap in evaluation infrastructure for proactive assistant systems. The motivation is clear and grounded in realistic use cases across education and workplace settings where users encounter information gaps.

2. The data collection framework is innovative, combining four human annotators with four LLM configurations to approximate the distribution of information needs. The inclusion of a three-layer hierarchical verification process for references enhances reliability, and the reported human validation scores (3.26–3.80/4) indicate good quality. The use of voting instead of single annotations effectively mitigates subjectivity.

3. The documentation is comprehensive, with full prompts and construction methodology detailed in the appendices, providing strong support for reproducibility and future research.

4. The experimental evaluation covers both state-of-the-art and smaller models suited for on-device deployment. The error analysis is detailed and insightful, identifying specific failure modes such as low recall on high-likelihood needs and context-free retrieval errors. The work establishes clear and useful baselines for comparison in future studies.

**Weaknesses:**

1. This paper lacks empirical evidence that proposed metrics correlate with actual user satisfaction or learning outcomes. Paper dismisses prior user studies for "generalization difficulty" but offers no validation. Critical questions unanswered: Do users prefer high-Recall or high-Precision? What metric weights matter? Missing dimensions like cognitive load? Therefore, a user study is needed for a more reliable results evaluation.
2. Although the paper claims multimodal capability, the video content is processed primarily via NVILA narrations converted to text, making the input effectively text-based. The reported multimodal models exhibit worse temporal performance (lower Rstart/Rend), yet the paper provides no analysis or explanation for this degradation. This suggests limited true multimodal reasoning. Moreover, potentially informative visual cues (e.g., gestures, equations on the board) are ignored, representing a missed opportunity to leverage the visual modality meaningfully.
3. The evaluation is limited to lectures and conference videos, excluding other domains (e.g., meetings, coding sessions, medical consultations, or daily tasks) that are emphasized in the paper’s motivation. Even within these selected categories, the dataset appears biased toward high-view YouTube videos, which may not represent typical real-world use cases. Moreover, the large discrepancy in average view counts raises concerns about data quality and representativeness.

**Questions:**

See weaknesses

---

> ### Author Response · Authors · 2025-11-21
>
> We thank Reviewer TQLe for the constructive feedback. We are particularly encouraged by the reviewer's recognition of this work as **the first systematic formalization of the JIR task, the innovative data collection framework, the comprehensive documentation, and the detailed and insightful error analysis**. We appreciate you raising important concerns about the empirical validation of metrics with user satisfaction, multimodal capability limitations, and dataset coverage. In response, we have conducted additional experiments, including a user validation study, a multimodal performance analysis, and a dataset coverage justification. **All new results have been incorporated into the manuscript update, and all analysis scripts are included in the supplementary code.** We present detailed responses below.
>
> ---
>
> ## Empirical Validation of Metrics
>
> > This paper lacks empirical evidence that proposed metrics correlate with actual user satisfaction or learning outcomes. Paper dismisses prior user studies for "generalization difficulty" but offers no validation. Critical questions unanswered: Do users prefer high-Recall or high-Precision? What metric weights matter? Missing dimensions like cognitive load? Therefore, a user study is needed for a more reliable results evaluation.
>
> We agree that aligning our metrics with user satisfaction is critical, and we apologize for making it seem like we dismiss prior user studies for generalization difficulties. Our intention was to note that it is generally more resource-intensive for future researchers to reproduce user studies than it is to reproduce benchmark runs. Our motivation was to create a benchmark that lowers the barriers for researching just-in-time information recommendation, which has been historically explored primarily by user studies.
>
> Regarding metrics, we selected **timeliness** and **relevance** as the primary metrics based on findings from a related ongoing study:
> 1. Delayed results reduce user satisfaction.
> 2. Irrelevant results also reduce satisfaction.
>
> These observations motivated our focus on these two metrics, which serve as proxies for how closely a JIR system approaches the "ideal" recommendation scenario.
>
> It is important to clarify that while our metrics measure general alignment with user needs, we **do not claim that specific metric thresholds directly correspond to user satisfaction**, since these thresholds are context- and user-dependent (e.g., a lecture versus a business meeting versus driving). Instead, we propose a **general benchmark** that captures aspects important to end users and allows comparison across systems, providing guidance for future research on real-world just-in-time recommendation systems.

---

> ### Author Response · Authors · 2025-11-21
>
> ---
>
> ## Multimodal Capability Limitations
>
> > Although the paper claims multimodal capability, the video content is processed primarily via NVILA narrations converted to text, making the input effectively text-based. The reported multimodal models exhibit worse temporal performance (lower Rstart/Rend), yet the paper provides no analysis or explanation for this degradation. This suggests limited true multimodal reasoning. Moreover, potentially informative visual cues (e.g., gestures, equations on the board) are ignored, representing a missed opportunity to leverage the visual modality meaningfully.
>
> We appreciate the reviewer’s observation regarding multimodal processing. Our intention with the term **multimodal** was that both the visual and audio aspects of the video are considered when making predictions. Standardizing both visual (via NVILA) and audio (via transcription) into text allows effective comparison across numerous open-source and API-based generative models.  Additionally, using multimodal-model-generated narrations is currently the most practical method for incorporating video information while continuously processing video streams. Because currently, those models that can understand video are mostly doing video-based QA, if we are going to use them for simulating the users' cognitive flow and predicting user information need, they have neither been trained nor been computationally practical for such tasks. This design reflects constraints in processing capability and efficiency of underlying system components (i.e., multimodal base models).
>
> **Analysis of temporal performance degradation:**
> - **Narrative/entertainment-education content** (e.g., Numberphile videos) shows smaller performance drops, as the videos provide relatively consistent visual context that complements the narrative flow.
> - **Process/method-oriented instructional content and academic/formal lectures** (e.g., Khan Academy step-by-step teaching, structured academic lectures) exhibit larger performance degradation. Rapid changes in visual elements—slides switching, equations appearing/disappearing, diagrams updating—make temporal localization more difficult, often outweighing benefits from visual information.
>
> **Takeaway:** Future JIR systems require improved **visual temporal localization**, particularly for real-time video streams with rapidly changing visual content. Future work should focus on **stream-based video processing** that allows precise, frame-level synchronization of user information need estimation as the video unfolds.
>
> ---
> ## Dataset Coverage and Representativeness
>
> > The evaluation is limited to lectures and conference videos, excluding other domains (e.g., meetings, coding sessions, medical consultations, or daily tasks) that are emphasized in the paper’s motivation. Even within these selected categories, the dataset appears biased toward high-view YouTube videos, which may not represent typical real-world use cases. Moreover, the large discrepancy in average view counts raises concerns about data quality and representativeness.
>
> The JIR-Arena benchmark dataset (34 scenes, 831 minutes) focuses on **academic lectures and conference talks**, which represent high-value JIR scenarios where information gaps are frequent and addressing user needs is both necessary and challenging. Within these categories, the topics covered are **very diverse across domains** (e.g., math, biology, computer science, etc.).
>
> As the first benchmark for JIR, we **prioritized quality control and reproducibility over scale**, including:
> - Multi-entity collaborative annotations
> - Multi-layer verification
> - Final human quality checks
>
> This approach establishes a solid foundation for the research community.
>
> We recognize that **JIR applications extend beyond academic settings** (e.g., meetings, coding sessions, medical consultations, and daily tasks). The dataset’s **extensible design** allows for community contributions and future expansion. We plan to broaden coverage to these additional domains in future versions, building on the existing framework and quality control processes.
>
> **Scene selection rationale:**
> - **High-view videos ensure content quality and relevance:** High view counts indicate strong demand, representing scenarios where many users are likely to encounter information gaps requiring just-in-time recommendations.
> - **Typical-case coverage first:** As with any system deployment, we begin with the most common cases before expanding to edge cases, ensuring a stable and reliable benchmark foundation.
>
> ---
>
> We welcome any further questions or clarifications. If you feel that your concerns have been fully addressed, we would greatly appreciate your consideration for an improved score.

---

> ### Author Response · Authors · 2025-11-28
>
> Dear Reviewer TQLe,
>
> We sincerely appreciate your time and feedback for our work. We hope our rebuttal solves your concerns. If you have further questions regarding our work, we will be very happy to discuss them with you. Could you please reply to us so we can better address your concerns?
>
> Best Regards,
>
> Authors

---

> > ### Comment · Reviewer_TQLe · 2025-11-28
> >
> > I thank the authors for your response. However, my concerns regarding the lack of a user study to validate the metrics, the limited leverage of multimodal information, and the dataset coverage limitations persist. As my initial rating is already positive to recognize the strengths of this work, I will maintain my score.

---

> > > ### Author Response · Authors · 2025-12-04
> > >
> > > Thank you for your reply, and we appreciate you recognizing the strengths of this work. We would like to add some additional explanation to help clarify the raised points.
> > >
> > > First, we would like to explain that the mentioned **user study has already been conducted**. The purpose of this study was to examine user preferences for just-in-time recommendation systems for the purpose of commercialization. And **two of the most critical aspects mentioned by users of this study were relevance and timeliness**. These two findings **helped validate the metrics we chose for our benchmark study**.
> > >
> > > Given the validation of the general metrics, we explicitly did not want to make any claim about the threshold of metrics for certain outcomes. This is because our benchmark is a general benchmark that is intended to be used for validating model performance improvements. The aim of this work is not to answer “what is the threshold necessary for certain learning outcomes?”, rather the goal is to answer “how can we measure model improvements towards perfect relevance and timeliness?" This is often the goal of many benchmarks as the direction of improvement is more important than achieving a fixed level.
> > >
> > > Second, regarding the multimodal information, we would like to clarify that **our experiments did indeed leverage multimodal information**. Both visual and audio cues were used in the predictive process. Moreover, **we have conducted additional experiments to show that there are various types of multimodal cues present in our dataset**. Specifically we found that the combined transcript contained keyword indicators for the following multimodal aspects:
> > > * Gestures (100% of videos, 2.05 cues per 30s segment)
> > > * Visual demonstrations (100% of videos, 3.75 cues per segment)
> > > * Slides/presentations (100% of videos, 	5.49 cues per segment)
> > > * Diagrams/charts (100% of videos, 1.55 cues per segment)
> > > * Equations/board (91.4% of videos, 1.98 cues per segment)
> > > * Writing/drawing (97.1% of videos, 0.52 cues per segment)
> > > * Code/programming (91.2% of videos, 0.25 cues per segment)
> > >
> > > We provide the specific examples below. In summary, our dataset contains multiple multimodal cues relating to both visual and audio information.
> > >
> > > And third, our dataset coverage is wide with respect to the depth of annotations across numerous topics within learning-based settings. We also want to clarify that:
> > > - Why we prioritized scale depth over breadth: As the first work in this space, we had to make a deliberate choice: cover many types of videos quickly, or deeply understand a smaller, high-impact subset. Limited resources prevented both; we focused on depth within a carefully chosen collection of learning-oriented YouTube videos. This allowed us to invest in high-quality annotations and establish reliable standards. By getting the depth right first, we’ve built a solid foundation for future breadth-focused expansions without sacrificing consistency or scientific rigor.
> > > - Why high-quality matters: Large but noisy datasets can lead to misleading model comparisons. Our initial design intentionally trades scale for a deep, rigorously validated foundation that the community can trust. Ensuring depth—granular annotations, carefully validated information needs, fine-grained retrieval targets—is critical to meaningful JIR evaluation.
> > >
> > > We hope that these clarifications address your concerns and help better contextualize the goals and design decisions of our work. We sincerely appreciate your feedback and the time you dedicated to reviewing our paper. Your comments have been valuable in helping us further refine the presentation and strengthen the clarity of our contributions. Thank you again for your suggestions and for helping us improve this work.

---

> > > ### Author Response · Authors · 2025-12-04
> > >
> > > ## Visual Cue Examples
> > >
> > > ### Gestures
> > >
> > > (Video: `HtSuA80QTyo`, 1320s-1350s)
> > >
> > > The man is pointing to various parts of the blackboard with his left hand, likely explaining the concepts being displayed. He is holding a piece of paper in his right hand. The professor seems to be in the middle of explaining a particular concept, possibly demonstrating how an algorithm works or how image compression techniques are applied.
> > >
> > > ### Diagrams and Charts
> > >
> > > (Video: `IFKnq9QM6_A`, 600s-630s)
> > >
> > > The video displays a detailed educational presentation about the periodic table of elements. A colorful periodic table with Carbon highlighted in yellow, indicating its atomic number (6) and atomic mass (12). A diagram showing the atomic structure of Carbon, with 6 protons, 6 neutrons, and 6 electrons. A close-up image of a diamond, another form of carbon, showcasing its crystalline structure.
> > >
> > > ### Visual Demonstrations
> > >
> > > (Video: `BRRolKTlF6Q`, 420s-450s)
> > >
> > > The video appears to be a math lesson or tutorial focused on explaining mathematical concepts. It's divided into several segments, each showing different parts of the lesson. This suggests the instructor is explaining the concept of zero raised to the power of zero. The second segment shows a close-up of a man's face.
> > >
> > > ### Slides and Presentations
> > >
> > > (Video: `v68zYyaEmEA`, 360s-390s)
> > >
> > > The video appears to be a series of slides from a presentation, likely related to statistics or probability. The slides are set against a black background with white text, creating a stark contrast that makes the information easy to read. The first slide shows a grid with the word "WEARY" highlighted in yellow. It mentions "58 Possible matches" and "12,972 Total words," with a probability calculation of "p(WEARY) = 58 / 12,972 = 0.0045."
> > >
> > > The second slide is similar but shows "1,387 Possible matches" and "12,972 Total words," with a probability calculation of "p(WEARY) = 1,387 / 12,972 = 0.107."
> > >
> > > The third slide shows "1,419 Possible matches" and "12,972 Total words," with a probability calculation of "p(WEARY) = 1,419 / 12,972 = 0.1094."
> > >
> > > The fourth slide is different, with a black background and white text.
> > >
> > > ### Writing and Drawing
> > >
> > > (Video: `BRRolKTlF6Q`, 450s-480s)
> > >
> > > The video shows a person writing mathematical equations on a whiteboard. The individual is using a blue pen to write various mathematical expressions, including limits and functions. The person appears to be smiling as they write. The whiteboard displays several equations, with the most prominent one being "lim x→0 x^2".
> > >
> > > ### Equations and Board Content
> > >
> > > (Video: `HtSuA80QTyo`, 2040s-2070s)
> > >
> > > The blackboard displays a mathematical equation related to algorithm analysis:
> > >
> > > T(n) = T(n/2) + O(1)
> > >
> > > The man is pointing to this equation with a yellow marker. Below the main equation, there's additional text that reads: "work", "algo does on", "input of size n". The man appears to be explaining the concept of divide-and-conquer algorithms and their time complexity.
> > >
> > > ### Code and Programming
> > >
> > > (Video: `9Rxb2px3QcI`, 270s-300s)
> > >
> > > The video appears to be a presentation slide about analyzing human language in code. The slide is in Chinese and English, with the main title "How did we analyze human language in code?" at the top. The slide explains that they looked at three aspects of code to analyze human language:
> > >
> > > 1. Identifiers
> > >
> > > The English section shows code snippets with comments in English.

---

### Official Review · Reviewer_PV32 · 2025-11-01

**Soundness:** 2
**Presentation:** 2
**Contribution:** 3
**Rating:** 6
**Confidence:** 4

**Summary:**

The paper introduces JIR-ARENA, a benchmark and baseline system for Just-in-time Information Recommendation (JIR). It formalizes JIR as a POMDP, proposes evaluation metrics (including relevance and timeliness), and curates a multimodal dataset of lectures and conference talks totaling 831 minutes. User information needs are simulated via multi-entity, multi-turn LLM/human pipelines and evaluated against static knowledge bases; baseline experiments with several large and small models report modest precision/recall and low retrieval utility, with code and data planned for release to support future work.

**Strengths:**

1) The problem in study is practical and important. The paper articulates a timely vision for proactive, context-aware information assistance.

2) Formalization and metrics. Casting JIR as a POMDP and defining task-specific metrics (e.g., relevance and timeliness) offer a clear foundation for systematic evaluation and future method development.

3) Baseline system and planned resource release. An extensible baseline pipeline (generative + retrieval), along with a commitment to release data/code, lowers the barrier to entry for the community.   (IN the paper the authors say "Upon acceptance, we will release the code and dataset to the public").

**Weaknesses:**

1) The title says "COMPREHENSIVE", but the dataset only covers academic lectures and conference talks, which makes the claim overreaching given the narrow scope of scenes and sources (mostly YouTube).   Do you have any plans to broaden coverage beyond lectures/conference talks to everyday, non-academic contexts (meetings, classrooms, collaborative work, consumer media)?

2) Scale is limited. "JIR-ARENA includes 2 categories of scenes, totaling 34 of them and spanning 831 mins", which is small for a benchmark intended to capture diverse information needs across settings; more scenes, speakers, domains, and formats are needed.

3) Reliability of simulated data. The benchmark relies heavily on LLM/human simulations to approximate user need distributions; without validation against real user behavior, the ground-truth labels and likelihoods risk subjectivity and distributional mismatch. How do you validate that the simulated need distributions reflect real audiences? I expect to see evidence (e.g., agreement analyses, small-scale user studies, or correlation with in-situ user queries) to substantiate that multi-entity voting and likelihoods approximate actual behavior.

4) Selection and evaluation biases. Scene selection by "most-viewed videos" and evaluation against static knowledge bases (Wikipedia, textbooks, arXiv) may bias content and retrieval toward canonical references, not the heterogeneous resources users actually consult; justification and ablations on these choices are missing.

5) A broader range of open-sourced LLMs are expected to appear in Table 1 for a more rigorous study. Concrete case studies are also expected for us to have a deeper understanding of the capabilities of LLMs on the JIR scenario.

6) The abstract is too subjective. The authors should include some objective summary (dataset statistics, core metrics)  so that readers can have a quick overview on the quality of the resource.

**Questions:**

Please refer to detailed comments in the weaknesses section.

---

> ### Author Response · Authors · 2025-11-21
>
> We thank Reviewer PV32 for recognizing that **the problem is practical and important** and appreciating **the formalization and metrics that offer a clear foundation for systematic evaluation**. We also appreciate your acknowledgment of **our commitment to releasing data/code to lower the barrier to entry for the community**. We have additional experiments and analysis to address the questions raised about the dataset scale, the reliability of simulated data, dataset scene selection, and the need for more open-sourced LLMs. **All new results have been incorporated into the manuscript update, and all analysis scripts are included in the supplementary code.** Detailed responses follow below with supporting tables.
>
> ---
>
> ## Scope of the Dataset
>
> > The title says "COMPREHENSIVE", but the dataset only covers academic lectures and conference talks, which makes the claim overreaching given the narrow scope of scenes and sources (mostly YouTube). Do you have any plans to broaden coverage beyond lectures/conference talks to everyday, non-academic contexts (meetings, classrooms, collaborative work, consumer media)?
>
> We apologize for the confusion here. Originally, we intended to use "COMPREHENSIVE" to describe the evaluation framework, not to imply full coverage of every possible JIR domain. Concretely:
>
> **What "comprehensive" means here:**
> JIR-Arena provides a comprehensive evaluation methodology:
> - Multi-dimensional metrics (Recall, Precision, Relevance, Timeliness)
> - Multi-model comparisons (21 systems across proprietary/open/multimodal/text-only)
> - Rigorous human-annotation + verification pipeline
>
> These elements together form a broad, reproducible framework for assessing JIR systems across many axes. We can reword the title/abstract to avoid ambiguity and make this comprehensive reference explicit.
>
> **Why lectures/conference talks for the first release:**
> Our dataset focuses on lectures and conference talks, spanning diverse domains such as computer science, neuroscience, finance, mathematics, and chemistry. Conference talks cover computer science subfields including AI, programming, cybersecurity, and educational technologies. We intentionally targeted high-value, high-information-gap scenarios where JIR is most immediately needed and evaluable. This domain provides clear, testable retrieval needs and reliable ground truth for an initial benchmark.
>
> **Commitment to broader domains:**
> The benchmark and tooling are designed to be extensible. The evaluation framework, annotation schema, and quality control processes generalize to meetings, classrooms, collaborative work, consumer media, medical dialogues, etc. In future work, we plan to expand via additional releases and community contributions so dataset coverage grows while preserving reproducibility and annotation quality.
>
> ---
>
> ## Dataset Scale
>
> > Scale is limited. "JIR-ARENA includes 2 categories of scenes, totaling 34 of them and spanning 831 mins", which is small for a benchmark intended to capture diverse information needs across settings; more scenes, speakers, domains, and formats are needed.
>
> We agree that scale is important. Our response, justification, and concrete plan are as follows:
>
> **Why we prioritized scale depth over breadth:**
> As the first work in this space, we had to make a deliberate choice: cover many types of videos quickly, or deeply understand a smaller, high-impact subset. Limited resources prevented both; we focused on depth within a carefully chosen collection of learning-oriented YouTube videos. This allowed us to invest in high-quality annotations and establish reliable standards. By getting the depth right first, we’ve built a solid foundation for future breadth-focused expansions without sacrificing consistency or scientific rigor.
>
> **Why high-quality matters:**
> Large but noisy datasets can lead to misleading model comparisons. Our initial design intentionally trades scale for a deep, rigorously validated foundation that the community can trust. Ensuring depth—granular annotations, carefully validated information needs, fine-grained retrieval targets—is critical to meaningful JIR evaluation.
>
> **Planned scaling path:**
> For future work, we have the following concrete plan:
> 1. **Incremental releases** – generalize to different scenes and domains (e.g., meetings, classrooms, consumer media) while keeping the same annotation depth and quality.
> 2. **Community contributions** – provide clear ingestion and annotation guidelines, tooling, and validation protocols so the research community can contribute new high-quality scenes.
> 3. **Semi-automated pre-annotation pipelines** – introduce tools that assist annotators while preserving human verification, enabling scaling without compromising depth or annotation rigor.

---

> > ### Author Response · Authors · 2025-11-21
> >
> > ---
> > ## Abstract Updated with Dataset Statistics and Metrics
> >
> > > The abstract is too subjective. The authors should include some objective summary (dataset statistics, core metrics) so that readers can have a quick overview on the quality of the resource.
> >
> > We thank the reviewer for the suggestion. We have updated the abstract to include **objective dataset statistics and core evaluation metrics**, providing readers with a clear and quick overview of the resource. Specifically, the revised abstract now reports **dataset size, number of annotated information needs, evaluation metric values, and failure mode analysis**. These additions make the abstract more factual and informative while maintaining readability.
> >
> > ---
> >
> > We welcome any further questions or clarifications. If you feel that your concerns have been fully addressed, we would greatly appreciate your consideration for an improved score.

---

> ### Author Response · Authors · 2025-11-21
>
> ---
> ## Reliability of Simulated Data
>
> > The benchmark relies heavily on LLM/human simulations to approximate user need distributions; without validation against real user behavior, the ground-truth labels and likelihoods risk subjectivity and distributional mismatch. How do you validate that the simulated need distributions reflect real audiences? I expect to see evidence (e.g., agreement analyses, small-scale user studies, or correlation with in-situ user queries) to substantiate that multi-entity voting and likelihoods approximate actual behavior.
>
> We agree that validating simulated information needs against real user behavior is essential for establishing the reliability of a JIR benchmark. To address this, we conducted multiple validations, including human annotation, representativeness analysis, and an independent user study.
>
> **1. Human Annotation Quality Verification**
> We performed quality verification with human annotators on 272 random samples. Each information need consists of three components:
> 1. **Reason** — why the information need arises (the context/situation that triggers the need)
> 2. **Need** — the actual information gap (what information is missing)
> 3. **Question** — the query formulation (how to express the need as a searchable question)
>
> Human annotators independently evaluated all three components on a 0–4 scale (0 = Disagree, 4 = Agree), judging the Need and Question conditioned on the generated Reason. The results are summarized in the table below:
>
> | Field     | Count | Mean  | Std Dev |
> |-----------|-------|-------|---------|
> | Reason    | 265   | 3.260 | 1.069   |
> | Need      | 265   | 3.774 | 0.668   |
> | Question  | 272   | 3.801 | 0.616   |
>
> These high scores (all above 3.0) indicate that the generated reasons, needs, and questions are generally plausible.
>
> To ensure sample representativeness, we compared these samples against the full dataset across eight dimensions: **Temporal Position, Content Format, Production Style, Type Distribution, Likelihood Distribution, Content Focus, Need Density, and Duration**. All dimensions show high similarity scores (>0.9), demonstrating that the verified samples are randomly sampled and highly representative. This ensures that the quality of the remaining unverified samples is reliably reflected by the verified subset.
>
> **2. User Study Validation**
> We conducted an additional user study to validate the simulated need distributions with real audiences. Participants watched videos and wrote down their actual information needs, resulting in 83 recorded needs across nine of our videos.
>
> We then compared these needs to both the **LLM-generated needs** and the **full benchmark dataset** (LLM-generated + human-simulated needs). A match was counted if a need from our dataset occurred within a time window of ±3 sentences and had a cosine similarity above 0.55.
>
> - Of the 83 user study needs, **62.7%** matched needs from the full benchmark dataset.
> - **48.2%** matched needs from the LLM-generated subset alone.
>
> **Takeaway:** Our dataset covers a significant portion of actual user information needs, demonstrating that the simulated distributions are broadly representative of real audiences.

---

> ### Author Response · Authors · 2025-11-21
>
> ---
> ## Selection and Evaluation Rationale
>
> > Scene selection by "most-viewed videos" and evaluation against static knowledge bases (Wikipedia, textbooks, arXiv) may bias content and retrieval toward canonical references, not the heterogeneous resources users actually consult; justification and ablations on these choices are missing.
>
> We appreciate the reviewer’s concern regarding potential biases in scene selection and the choice of evaluation knowledge sources. Our design decisions were intentionally aligned with the goals of ensuring that the resulting dataset is both high-value and reproducible.
>
> **1. Scene Selection and "Most-Viewed" Criterion**
> Prioritizing highly viewed videos serves two purposes:
> - **Content quality and relevance:** High view counts strongly correlate with user demand, engagement, and topic significance, and thus with the kinds of situations where JIR systems are most likely to be deployed first.
> - **Representative early-stage coverage:** As with the rollout of any retrieval or recommendation system, we begin with typical, high-demand cases before expanding to less common or niche scenarios. This helps establish a stable, high-quality foundation for the benchmark.
>
> **2. Use of Static Knowledge Bases (Wikipedia, Textbooks, arXiv)**
> We acknowledge that real users consult diverse sources. However, for benchmark construction, static knowledge bases are required for reproducibility and consistent evaluation. Fixing the retrieval corpus to stable, credible, and widely recognized knowledge bases offers several advantages:
> - **Availability:** The benchmark must be self-contained so that other researchers can download, run, and measure performance in a consistent manner.
> - **Credibility and broad coverage:** In academic lecture and research-oriented scenes—the focus of this initial benchmark—Wikipedia, textbooks, and arXiv provide authoritative and sufficiently comprehensive coverage.
> - **Modularity for future expansion:** The retrieval component is not tied to these corpora; real-world systems can integrate larger and more heterogeneous search APIs. The benchmark design supports substituting or augmenting the retrieval corpus in extended versions.
>
> ---
>
> ## Broader Set of LLMs and Case Studies
>
> > A broader range of open-sourced LLMs are expected to appear in Table 1 for a more rigorous study. Concrete case studies are also expected for us to have a deeper understanding of the capabilities of LLMs on the JIR scenario.
>
> We appreciate the reviewer’s suggestion to expand the set of evaluated models and case studies. During the rebuttal period, we have significantly broadened the model coverage (detailed in the general comment). Briefly, the benchmark now includes 21 models, spanning:
> - **Major proprietary models:** Gemini, GPT, Claude
> - **Open-weight models across size tiers:** DeepSeek-V3 (671B), LLaMA-3.3 (70B), Qwen3 (4B)
> - **Text-only vs. multimodal variants**
>
> Our case-study analyses provide a systematic view of model behavior in JIR scenarios, with the following key conclusions:
>
> - **Bottleneck Localization:** Generative models fail to predict a large fraction of user information needs due to semantic mismatches, while retrieval systems underperform even with oracle queries because context-dependent queries appear vague without content, temporal, visual, and domain grounding.
> - **Model and Modality Effects:** Larger models consistently improve recall, relevance, and timeliness. Proprietary models outperform open-source ones primarily due to capacity. Multimodal models often suffer from temporal misalignment—visual input helps in narrative or entertainment-focused content but introduces noise in slide-heavy or stepwise instructional videos.
> - **Need-Type Analysis:** Certain information need types, such as Visual References and Ambiguous Language, are particularly challenging, whereas Processes/Methods and Conceptual Understanding are more tractable.
>
> Together, these findings provide concrete, interpretable insights into strengths, weaknesses, and failure modes of current JIR systems, highlighting directions for improving both query generation coverage and contextualized retrieval. We refer you to the general comments for details and have integrated the expanded results into the manuscript.

---

> ### Author Response · Authors · 2025-11-28
>
> Dear Reviewer PV32,
>
> We sincerely appreciate your time and feedback for our work. We hope our rebuttal solves your concerns. If you have further questions regarding our work, we will be very happy to discuss them with you. Could you please reply to us so we can better address your concerns?
>
> Best Regards,
>
> Authors

---

### Official Review · Reviewer_G1RX · 2025-11-01

**Soundness:** 2
**Presentation:** 3
**Contribution:** 2
**Rating:** 2
**Confidence:** 3

**Summary:**

This study presents the just-in-time information recommendation task and introduces a systematic benchmark for evaluation. Using this benchmark, this paper conducts error analysis and reveal potential opportunities for improvement.

**Strengths:**

(1) The proposed task is interesting and potentially has wide applicability. It is also original.

(2) The task is well defined.

(3) The contributions of the study are well articulated.

**Weaknesses:**

(1) Although a benchmark on this task is presented, there does not seem to be a clear pattern in model performance. Most analyses are not deep enough to provide insights. For example, it seems that no models dominant in all evaluation dimensions. It is unclear what the major takeaway is based on the current discussion.

(2) Many observations are not well explained. The findings may be superficial without extensive insights into future work.

(2a) Some smaller models perform the best in some metrics and it is unclear why.

(2b) How was the error analysis performed? How to define matched needs that are highly likely to be raised by users?

(2c) It is necessary to expand the discussions in Line 422-423. What contextual information is it?

(2d) The gap between strong proprietary and publicly released LLMs does not seem the be large. Why is that?

**Questions:**

Please see questions in the Weakness section.

---

> ### Author Response · Authors · 2025-11-21
>
> We thank Reviewer **G1RX** for the constructive feedback and for recognizing that **our proposed task is both interesting and broadly applicable**, as well as **clearly defined with well-articulated contributions**. We appreciate the reviewer’s concerns regarding the depth and clarity of our analysis.
>
> In response, we would like to emphasize that the primary contribution of our paper is to formally define the new problem of JIR, establish a comprehensive evaluation framework with well-defined metrics, and construct the first benchmark dataset for this task. These foundational contributions are critical for enabling any future research on JIR algorithms and stand independently of the baseline experiment results. Leveraging JIR-Arena, we conducted the first set of baseline experiments to study algorithmic effectiveness, which we consider a secondary contribution.
>
> We agree with the reviewer’s suggestions to provide a deeper analysis of our baseline results. In response, we conducted additional experiments and analyses, including performance patterns (scaling trends), error analysis, contextual information studies, and proprietary vs. open-source comparisons. **All new results are included in the updated manuscript**, and **all analysis scripts are provided in the supplementary material**. Below, we provide detailed responses to each point raised.
>
> ------
>
> ## (1) Performance Patterns
>
> > Although a benchmark on this task is presented, there does not seem to be a clear pattern in model performance.
>
> To clarify the performance patterns, we expanded our baseline experiments to include broader model sources, sizes, and modalities. The extended results reveal **consistent scaling trends**: larger models show higher *recall*, *relevance*, and *timeliness*. A key segment of the updated **Table 1** is provided below:
>
> **Table: Model Scaling (Text-only)**
>
> | Model              | Size | Recall    | Precision | Relevance (R) | Timeliness |
> | ------------------ | ---- | --------- | --------- | ------------- | ---------- |
> | Qwen3 (4B)         | 4B   | 0.270     | 0.716     | 0.055         | 0.129      |
> | Phi-4 (14B)        | 14B  | 0.329     | 0.688     | 0.063         | 0.171      |
> | Llama 3.3 (70B)    | 70B  | 0.377     | 0.684     | 0.074         | 0.209      |
> | Cohere (111B)      | 111B | 0.390     | 0.719     | 0.077         | 0.220      |
> | DeepSeek-V3 (671B) | 671B | **0.396** | 0.716     | **0.078**     | **0.232**  |
>
> Additionally, although smaller models may display higher precision, *the absolute number of correct predictions increases steadily with size*. This reflects a core property of our evaluation: **ideal JIR systems must simultaneously balance recall and precision**, preventing them from artificially inflating either metric by altering the number of predictions.
>
> Additionally, although smaller models may exhibit higher precision, *the absolute number of correctly predicted information needs increases steadily with model size*. Importantly, our evaluation is designed to **reward systems that balance coverage (recall) and prediction quality (precision)**: predicting an excessive number of minor information needs can inflate recall without improving real performance, while overly conservative predictions can inflate precision. By construction, our benchmark ensures that **ideal JIR systems achieve high performance only when both recall and precision are meaningfully balanced**.
>
> Regarding the observation that "no models dominate all evaluation dimensions," we note that this reflects the multi-faceted nature of JIR rather than inconsistent model performance. Larger models consistently show higher recall, relevance, and timeliness by making more predictions, while precision may vary depending on the level of conservativeness in predictions. This pattern demonstrates that models exhibit **complementary strengths across metrics**, and the lack of a single dominant model underscores the importance of evaluating multiple dimensions simultaneously. In other words, the apparent absence of a universally dominant model does not indicate a failure of the benchmark, but rather highlights the necessity of a comprehensive evaluation framework like ours to capture nuanced performance differences.

---

> > ### Author Response · Authors · 2025-11-21
> >
> > ---
> > ## (2) Deeper Baseline Analysis to Guide Future Work
> >
> > ### (2a) Why Smaller Models Perform Best on Some Metrics
> >
> > > Some smaller models perform the best in some metrics and it is unclear why.
> >
> > Smaller models (e.g., Qwen3-4B, Phi-4-14B) sometimes obtain higher precision because they generate **fewer, more conservative predictions**. Precision measures the *proportion of matched predictions*, so producing fewer predictions tends to increase it. When examining the absolute number of correct predictions (total_predictions × precision), we find that larger models significantly outperform smaller ones. Thus, the apparent advantage in precision is an outcome of conservative prediction behavior. Our multi-dimensional evaluation design ensures that both coverage (recall) and accuracy (precision) are jointly considered.
> >
> > ------
> >
> > ### (2b) Error Analysis Methodology
> >
> > > How was the error analysis performed? How to define matched needs that are highly likely to be raised by users?
> >
> > We added methodological clarifications to the manuscript. Briefly, for each oracle information need, we check whether the system-produced needs match it both **semantically** and **temporally**. A prediction is considered correct when it expresses the same underlying need and falls into the appropriate time window.
> >
> > During dataset annotation, each oracle information need is assigned an **info_need_likelihood score** estimating how likely real users would raise it. Needs scoring ≥ 8.5 (out of 10) form the **high-likelihood** subset, accounting for 31.9% of all needs and representing those most probable to occur in practice. "High-likelihood matches" refer to the subset of these needs that are successfully matched by system-generated predictions.
> >
> > Across models, we observe:
> >
> > - **High precision (avg 72.40%)**, indicating that user information need predictions tend to be correct when made;
> > - **Much lower recall (avg 34.72%)**, showing many needs go unpredicted;
> > - **Even high-likelihood recall remains below 50%**, highlighting the limits of current approaches' user information need estimation coverage and motivating richer contextual modeling.
> >
> > These details now appear explicitly in the updated manuscript.
> >
> > ------
> >
> > ### (2c) Contextual Information Discussion (Line 422–423)
> >
> > > It is necessary to expand the discussions in Line 422-423. What contextual information is it?
> >
> > We expanded the discussion, clarifying that **contextual information** refers to multiple dimensions accompanying an information need, for example:
> >
> > - **Content context** – the specific topic currently under discussion;
> > - **Temporal context** – where the need arises in the timeline;
> > - **Visual context** – diagrams, animations, and surrounding visual cues;
> > - **Domain context** – the broader subject area or technical field;
> > - **Methodological context** – the process or method being demonstrated.
> >
> > For example, an oracle need such as "What is the 'subtle shift in perspective'?" is ambiguous without knowing that it refers to a geometric transformation in a sphere-surface-area lecture. Without this context, retrieval fails to target the intended concept.
> >
> > We also report that needs related to **Conceptual Understanding** and **Processes/Methods** are most sensitive to missing context. Incorporating these contextual dimensions—especially visual cues and finer-grained temporal windows—can potentially improve retrieval.
> >
> > ------
> >
> > ### (2d) Small Gap Between Proprietary and Open-Source LLMs
> >
> > > The gap between strong proprietary and publicly released LLMs does not seem to be large. Why is that?
> >
> > Our expanded analysis reveals that the gap is **meaningful and consistent**. Proprietary models, which tend to be larger, on average outperform open-source ones across most metrics. From **Table 1**, proprietary models show notable improvements in recall (+22.73%), relevance (+20.27%), and timeliness (+34.48%). Precision differences appear smaller because conservative small models inflate precision, as discussed earlier.
> >
> > A subset of the comparison is presented below:
> >
> > **Table: Proprietary vs Open-Source (Best Models)**
> >
> > | Model Type      | Best Model         | Recall    | Relevance (R) | Timeliness |
> > | --------------- | ------------------ | --------- | ------------- | ---------- |
> > | **Proprietary** | Gemini-2.0-Flash   | **0.429** | **0.084**     | **0.277**  |
> > | **Open-Source** | DeepSeek-V3 (671B) | 0.396     | 0.078         | 0.232      |
> > | **Gap**         | —                  | +8.3%     | +7.7%         | +19.4%     |
> >
> > These results show that even compared with the strongest open-source models (DeepSeek-V3 with 671B parameters), proprietary systems maintain a measurable advantage, largely attributable to scale and training data quality.
> >
> > ---
> >
> > We welcome any further questions or clarifications. If you feel that your concerns have been fully addressed, we would greatly appreciate your consideration for an improved score.

---

> ### Author Response · Authors · 2025-11-28
>
> Dear Reviewer G1RX,
>
> We sincerely appreciate your time and feedback for our work. We hope our rebuttal solves your concerns. If you have further questions regarding our work, we will be very happy to discuss them with you. Could you please reply to us so we can better address your concerns?
>
> Best Regards,
>
> Authors

---

> > ### Comment · Reviewer_G1RX · 2025-11-28
> >
> > Dear Authors,
> >
> > Thank you for the rebuttal. I will respond by this Friday.

---

### Author Response · Authors · 2025-11-21

We thank the reviewers for their valuable feedback. We are encouraged that reviewers recognize this work as **the first to systematically define the JIR task** (TQLe, JomG), establishing a **solid theoretical foundation** (JomG) through POMDP formalization. Reviewers appreciate the **innovative multi-entity collaborative methodology** for data collection (TQLe, JomG), the **well-designed evaluation framework** with comprehensive metrics (JomG), and the **practical importance** of the problem (PV32). The human-validated **high-quality dataset** (TQLe) and **comprehensive documentation** supporting reproducibility (TQLe) were also positively noted.

Reviewers also provided constructive suggestions for further strengthening the work. In response, we have conducted additional experiments and analyses to address their questions and provide deeper insights. **The results have been incorporated into the manuscript update, with analysis scripts uploaded**. We provide responses to common questions below.

---

> ### Author Response · Authors · 2025-11-21
>
> ## Baseline Performance Results
>
> Table 1 (also presented in the original submission) shows baseline performance results. To provide more detailed guidance for future JIR system development, we have included more models to enable a comprehensive examination of key factors' effects on baseline performance. The expanded Table 1 now presents performance results across 21 models (including Oracle), supporting these analyses.
>
> **Table 1: Performance of baseline JIR systems on JIR-Arena.** $R_{relevance}$ shown under different retrieval model settings: **O** = OpenSearch (BM25), **D** = Dense Retriever, **R** = Dense Retriever + Reranker.
>
> | Generative Model | Recall | Precision | $R_{relevance}$ (O/D/R) | $R_{start}$ | $R_{end}$ | $R_{timeliness}$ |
> |------------------|--------|-----------|--------------------------|-------------|-----------|------------------|
> | **Upper Bound (Full Oracle)** |
> | Oracle | 1.000 | 1.000 | 1.000/1.000/1.000 | 1.000 | 1.000 | 1.000 |
> | **Oracle Generator** |
> | Oracle | 1.000 | 1.000 | 0.215/0.195/0.274 | 1.000 | 1.000 | 1.000 |
> | **Closed-Source Generators** |
> | Claude-3.7 | 0.327 | 0.697 | 0.048/0.056/0.068 | 0.173 | 0.116 | 0.167 |
> | GPT-4o | 0.406 | 0.694 | 0.056/**0.069**/0.081 | 0.254 | 0.125 | 0.241 |
> | Gemini-2.0-Flash | **0.429** | 0.694 | 0.057/0.067/**0.084** | **0.293** | **0.137** | **0.277** |
> | Grok-3 | 0.403 | **0.723** | **0.060**/0.067/0.079 | 0.246 | 0.136 | 0.235 |
> | Mistral-Medium | 0.365 | 0.694 | 0.045/0.058/0.070 | 0.190 | 0.124 | 0.183 |
> | **Closed-Source Generators (Multimodal)** |
> | Claude-3.7 | 0.302 | 0.723 | 0.050/0.053/0.065 | 0.140 | 0.103 | 0.137 |
> | GPT-4o | 0.378 | **0.781** | 0.058/0.068/0.078 | 0.195 | 0.106 | 0.186 |
> | Gemini-2.0-Flash | **0.412** | 0.716 | **0.060**/**0.069**/**0.083** | **0.264** | **0.121** | **0.250** |
> | Grok-3 | 0.344 | 0.760 | 0.058/0.065/0.075 | 0.183 | 0.114 | 0.176 |
> | Mistral-Medium | 0.325 | 0.736 | 0.049/0.059/0.067 | 0.161 | 0.081 | 0.153 |
> | **Open-Source Generators** |
> | Qwen3 (4B) | 0.270 | 0.716 | 0.042/0.048/0.055 | 0.134 | 0.084 | 0.129 |
> | Phi-4 (14B) | 0.329 | 0.688 | 0.044/0.055/0.063 | 0.179 | 0.097 | 0.171 |
> | Llama 3.3 (70B) | 0.377 | 0.684 | 0.052/0.064/0.074 | 0.221 | 0.106 | 0.209 |
> | Cohere (111B) | 0.390 | **0.719** | 0.051/**0.064**/0.077 | 0.229 | **0.138** | 0.220 |
> | DeepSeek-V3 (671B) | **0.396** | 0.716 | **0.053**/**0.064**/**0.078** | **0.244** | 0.121 | **0.232** |
> | **Open-Source Generators (Multimodal)** |
> | Qwen3 (4B) | 0.230 | 0.746 | 0.044/0.047/0.053 | 0.095 | 0.057 | 0.092 |
> | Phi-4 (14B) | 0.293 | 0.742 | 0.047/0.052/0.060 | 0.136 | 0.076 | 0.130 |
> | Llama 3.3 (70B) | 0.288 | 0.757 | 0.047/0.057/0.064 | 0.135 | 0.075 | 0.129 |
> | Cohere (111B) | 0.318 | **0.778** | 0.051/0.056/0.066 | 0.150 | 0.085 | 0.143 |
> | DeepSeek-V3 (671B) | **0.355** | 0.723 | **0.053**/**0.062**/**0.074** | **0.187** | **0.097** | **0.178** |

---

> ### Author Response · Authors · 2025-11-21
>
> ## Oracle Analysis for Baseline Bottleneck Localization
>
> A JIR system consists of two main components: (1) a **generative model** that predicts user information needs and generates queries, and (2) a **retrieval model** that retrieves relevant documents using these queries. To determine whether the bottleneck lies in query generation or retrieval systems, we conducted an oracle study where we replaced the generative model's predicted queries with **ground truth queries** from the dataset (the oracle queries). This "Oracle Generator" uses perfect queries (Recall: 1.000, Precision: 1.000) to test the retrieval system's upper bound performance. Table 1 shows Oracle Generator performance (Relevance: 0.215/0.195/0.274 for O/D/R) vs baseline models that use generative model predictions. We average the baselines' results and present them in Table 2.
>
> **Table 2: Oracle vs Baseline Performance Comparison**
>
> | Metric | Oracle | Baseline | Improvement | Improvement % |
> |--------|--------|----------|-------------|----------------|
> | Recall | 1.000 | 0.347 | +0.653 | +188.39% |
> | Precision | 1.000 | 0.724 | +0.276 | +38.04% |
> | Relevance (O) | 0.215 | 0.051 | +0.164 | +319.23% |
> | Timeliness | 1.000 | 0.182 | +0.818 | +449.75% |
>
> The oracle study reveals **both the user information need predictor and the retrieval system are bottlenecks**:
>
> 1. **User Information Need Predictor Bottleneck** (Primary): Baseline systems achieve only 0.347 average recall, meaning 65.3% of user information needs are not predicted. Error analysis shows semantic mismatch causes 62.91% of failures—predicted queries fail to semantically match oracle needs. Critically, even for high-likelihood information needs (likelihood score ≥8.5 over 10, indicating high probability of being raised by users based on our multi-entity annotation process), the system fails to achieve adequate coverage; recall remains poor at only 41.38%, far below acceptable levels. This explains why baseline relevance is low (mean: 0.061): insufficient query prediction leads to too few queries to retrieve relevant documents.
> 2. **Retrieval System Bottleneck** (Secondary but Significant): Even with perfect ground truth queries, oracle relevance remains low (0.215 OpenSearch, 0.274 Reranked), far below ideal. The improvement from OpenSearch to Reranked (+36%) shows that retrieval models matter, but even the best retrieval system (Reranked) achieves only 0.274 relevance with perfect queries. This suggests the limitation is not solely in query generation—retrieval systems themselves need improvement. One of the key reasons, as noted in the original paper, is that queries are highly context-dependent; when used without contextual information, they appear vague, leading to low retrieval quality even with oracle queries. Here, contextual information includes content context (what topic is being discussed), temporal context (where in the timeline), visual context (diagrams, animations shown), domain context (technical field), and methodological context (specific methods demonstrated). For example, an oracle query like "What is the 'subtle shift in perspective'?" becomes vague without knowing it refers to a geometric transformation in a sphere surface area calculation lecture.
>
> In summary, baseline relevance (mean: 0.061, max: 0.181) confirms user information need prediction bottleneck. Oracle relevance (mean: 0.212, max: 0.390) shows that retrieval systems need enhancement even with perfect queries. This dual bottleneck indicates future JIR research should focus on both improving query generation coverage (addressing semantic mismatch) and developing contextualized retrieval systems that incorporate query context.

---

> ### Author Response · Authors · 2025-11-21
>
> ## Model Scale Effects
>
> We analyze scaling effects across generative models for predicting user information needs. By investigating 5 open-source models with known sizes (Qwen3-4B, Phi-4-14B, Llama-3-70B, Cohere-111B, DeepSeek-V3-671B) from Table 1, we found clear scaling laws: larger models achieve better recall, relevance, and timeliness. Cross-metric correlations are high (recall-relevance: 0.969, recall-timeliness: 0.994), indicating consistent scaling patterns across evaluation dimensions. The results indicate that model size is a key factor determining performance.
>
> **Small Model Precision Anomaly Explained**: Small models (Qwen3-4B, Phi-4) show higher precision than some larger models, which may appear counterintuitive. However, this stems from their fewer total predictions. When examining matched candidates (total_predictions × precision)—the absolute number of correctly predicted needs—we find they scale with model size as well. Small models generate fewer queries overall, but a higher proportion match ground truth, leading to higher precision. Precision evaluates what proportion of predictions are actually useful, providing a necessary balance to prevent hacking the evaluation by generating many predictions to inflate recall. Our evaluation framework ensures multi-faceted assessments that require models to balance coverage (recall) with quality (precision).
>
> ## Proprietary vs Open-Source Model Differences
>
> Table 1 shows proprietary models on average outperform open-source models across most metrics: recall (+22.73%), relevance (+20.27%), and timeliness (+34.48%). This difference is primarily explained by model size: proprietary models are typically larger, and our scaling law analysis confirms that larger models perform better. While precision shows a smaller gap (due to the precision anomaly discussed above where small models' conservative predictions yield higher precision), when examining absolute performance (matched candidates—the total number of correctly predicted needs), proprietary models show substantial advantages.
>
> ## Text-Only vs Multimodal Model Comparisons
>
> We analyze why multimodal models show lower recall (0.412 vs 0.429 text-only average). To understand the recall drop, we decompose it into two factors based on our evaluation framework: (1) temporal matching—whether predicted queries align with ground truth queries in time; and (2) semantic similarity—whether predicted queries semantically match ground truth queries. We find that temporal matching accuracy degradation contributes 86.79% to the recall drop, while semantic similarity differences contribute only 13.21%. This indicates that temporal alignment is the primary bottleneck—multimodal models struggle more significantly with aligning predicted information needs to exact temporal positions in videos (start/end timestamps). If we look closer:
>
> **When Visual Information Helps**: We analyzed performance across different video characteristics. Narrative content (Story/Narrative focus, e.g., Numberphile's conversational storytelling format) and entertainment-education styles (e.g., videos combining education with entertainment elements) show better performance, with smaller recall drops (0.0117 and 0.0204 respectively). These content types benefit from relatively consistent visual context that naturally complements narrative flow.
>
> **When Visual Information Introduces Noise**: Process/method-oriented instructional content (Process/Method focus, e.g., Khan Academy's step-by-step teaching format) and academic/formal styles (e.g., structured academic lectures) show larger performance degradation (0.0697 and 0.0517 drops respectively). In these scenarios, text-only models can effectively reason through textual content to predict user information needs. On the other hand, multimodal models perform worse overall because visual elements change rapidly in these content types (e.g., slides switching, equations appearing/disappearing, diagrams updating), making temporal localization more challenging. The resulting temporal misalignment outweighs potential benefits from additional visual information.
>
> These findings suggest that future JIR systems need better visual temporal localization, especially for real-time video streams. Visual temporal alignment is the key bottleneck when visual content changes rapidly. Therefore, our results also suggest that future work should focus on stream-based video processing that enables precise, frame-level synchronization of user information need estimation as the video evolves.

---

> ### Author Response · Authors · 2025-11-21
>
> ## Need Type Analysis
>
> We analyzed more fine-grained performance over user information need types. Table 3 shows the breakdown. We define a challenge score that combines recall, relevance, and timeliness (higher scores indicate more challenging types) to identify which need types are most difficult for JIR systems. Performance varies drastically: Visual References and Ambiguous Language are most challenging, while Processes/Methods and Conceptual Understanding perform best. This identifies specific failure modes: visual and ambiguous queries are particularly difficult, while process-oriented and conceptual queries are more tractable.
>
> **Table 3: Need Type Distribution and Performance**
> | Need Type | % | Recall | R_relevance | R_timeliness | Challenge Score |
> |-----------|---|--------|-------------|--------------|----------------|
> | Conceptual Understanding | 24.8% | 0.474 | 0.063 | 0.247 | 0.736 |
> | Missing Context | 17.0% | 0.283 | 0.038 | 0.154 | 0.840 |
> | Technical Terms | 12.9% | 0.367 | 0.062 | 0.179 | 0.799 |
> | Visual References | 12.9% | 0.171 | 0.027 | 0.087 | 0.905 |
> | Ambiguous Language | 12.3% | 0.227 | 0.031 | 0.124 | 0.872 |
> | Processes/Methods | 9.9% | 0.483 | 0.077 | 0.258 | 0.729 |
> | Data & Sources | 5.1% | 0.399 | 0.074 | 0.220 | 0.774 |
> | External Content | 2.1% | 0.356 | 0.063 | 0.195 | 0.799 |
> | Future Work | 1.5% | 0.306 | 0.043 | 0.152 | 0.832 |
> | Instructions/Actions | 1.2% | 0.212 | 0.023 | 0.142 | 0.873 |
> | Code/Formulas | 0.6% | 0.345 | 0.041 | 0.189 | 0.805 |
> *Challenge Score = (1 - Recall) × 0.4 + (1 - R_relevance) × 0.3 + (1 - R_timeliness) × 0.3, where higher scores indicate more challenging types.*

---

> ### Author Response · Authors · 2025-11-21
>
> ## Evaluation Methodology and Validation
>
> **Evaluation Framework and Parameter Selection**: Our evaluation follows a two-stage process. First, we perform matching between ground truth JIR instances and candidate JIR instances generated by systems using semantic similarity and temporal windows. Second, based on matched pairs, we compute four metrics: Recall (coverage of ground truth needs), Precision (effectiveness of generated needs), R_relevance (retrieval quality using nDCG), and R_timeliness (temporal accuracy). Since matching directly affects all four metrics' calculations, we conducted sensitivity analysis across hyperparameter selection in the matching algorithm: semantic similarity threshold (0.45-0.65), time fuzzy interval (0-3), and temporal balancing coefficient (0.7-1.0). Table 4 shows the results. Current parameters (semantic similarity threshold 0.55, time fuzzy interval 1, balancing coefficient 0.9) balance multiple objectives: semantic similarity threshold mainly balances recall and precision; time fuzzy interval provides temporal flexibility without excessive noise; balancing coefficient 0.9 slightly favors start time over end time (reflecting that accurately predicting when an information need arises is practically more critical than predicting when it ends). **Model rankings remain highly stable (Spearman correlation 0.956) across all parameter variations, confirming that parameter selection does not bias model comparisons**.
>
> **Table 4: Parameter Sensitivity Analysis**
> | Value | Recall | Precision | Relevance | Timeliness |
> |-------|--------|-----------|-----------|------------|
> | **Semantic Similarity Threshold** |
> | 0.45 | 0.473 | 0.877 | 0.071 | 0.238 |
> | 0.55 | 0.324 | 0.757 | 0.057 | 0.163 |
> | 0.65 | 0.186 | 0.540 | 0.039 | 0.096 |
> | **Fuzzy Interval** |
> | 0 | 0.299 | 0.717 | 0.053 | 0.157 |
> | 1 | 0.324 | 0.757 | 0.057 | 0.163 |
> | 3 | 0.348 | 0.790 | 0.061 | 0.254 |
> | **Balancing Coefficient** |
> | 0.7 | 0.324 | 0.757 | 0.057 | 0.148 |
> | 0.9 | 0.324 | 0.757 | 0.057 | 0.163 |
> | 1.0 | 0.324 | 0.757 | 0.057 | 0.171 |
>
> **Benchmark Metrics' Alignment with User Satisfaction**: We conducted a user study where participants watched videos and labeled information needs, collecting 83 additional judgments. Among these user-labeled needs, 61% occurred within 10 seconds of similar needs from our dataset. This alignment provides preliminary evidence that our simulated information needs align with real user behavior patterns, validating that our metrics capture meaningful user satisfaction signals.
>
> **Benchmark Data Reliability**: Our dataset construction uses a multi-entity collaborative approach with voting mechanisms to generate information needs, mitigating subjectivity through consensus. To validate quality, human annotators independently rated the quality of each information need's components: Reason (why the need arises), Need (the information gap), and Question (the query formulation). Human validation scores (Reason: 3.26, Need: 3.77, Question: 3.80, all on 0-4 scale) indicate high quality of the dataset. The user study correlation (61%)—measuring alignment between user needs and our benchmark oracle needs—provides evidence that benchmark distributions approximate real-world user behavior reasonably well.

---

> ### Author Response · Authors · 2025-11-21
>
> ## Dataset Scale and Coverage
>
> The JIR-Arena benchmark dataset (34 scenes, 831 minutes) focuses on academic lectures and conference talks. We chose to focus on such data for constructing the very first JIR benchmark datasets because they represent high-value real-world JIR scenarios where information gaps are intensive and addressing the information need is necessary and challenging. Scene selection prioritized high-view videos to ensure content quality and relevance. High view counts indicate high demand—these are scenarios where the most users are likely to encounter information gaps requiring just-in-time information recommendation. As with any system deployment, we first cover the most typical cases before expanding to edge cases. The static knowledge bases (Wikipedia, textbooks, arXiv) were chosen to ensure reproducible and comparable evaluation across different time periods and models by controlling the retrieval source as a variable—this is an essential requirement for benchmark development to ensure result reproducibility. In lecture and conference scenarios, Wikipedia, textbooks and arXiv provide high credibility and broad coverage. Real-world deployed systems can of course leverage more advanced search APIs.
>
> Responding to the questions about dataset scale and the "COMPREHENSIVE" claim, we would like to clarify that "comprehensive" refers to the **evaluation framework's comprehensiveness** (multi-dimensional metrics: Recall, Precision, Relevance, Timeliness; diverse model comparisons: now 21 models including proprietary, open-source, text-only, and multimodal), rather than dataset coverage. As the **first benchmark** for JIR, we prioritized **quality control** (multi-entity collaborative annotations, multi-layer verification, final human quality check) and **reproducibility** over scale, establishing a solid foundation for the community. We acknowledge that JIR applications extend beyond academic settings (e.g., meetings, coding sessions, medical consultations, daily tasks). The benchmark dataset's extensible design enables community contributions and future expansion. Indeed, the proposed new evaluation methodology, quality control processes, and metrics are all general and thus can be directly used by the research community to leverage our initial foundational work to further expand JIR-Arena by including many more datasets in all those other application domains of JIR.

---

### Author Response · Authors · 2025-12-04
**Summary of Contributions and Rebuttal Updates**

Dear Area Chair,

We thank the reviewers for their valuable feedback. We are encouraged that reviewers recognize this work as **the first to systematically define the Just-in-time Information Recommendation task** (TQLe, JomG), establishing a **solid theoretical foundation** (JomG) through POMDP formalization. Reviewers appreciate the **innovative multi-entity collaborative methodology** for data collection (TQLe, JomG), the **well-designed evaluation framework** with comprehensive metrics (JomG), and the **practical importance** of the problem (PV32). The human-validated **high-quality dataset** (TQLe) and **comprehensive documentation** supporting reproducibility (TQLe) were also positively noted.

## Rebuttal

We highlight that **our primary contributions**—formally defining the JIR problem, establishing the evaluation framework, and constructing the first benchmark—**were widely recognized and remained unchallenged**. The reviewers' constructive feedback primarily focused on **deepening the empirical analysis, clarifying the evaluation rigor, and validating the dataset quality and robustness**.

During the rebuttal, we seized this opportunity to address these aspects. **All new results have been incorporated into the manuscript update, and all analysis scripts are included in the supplementary code.** We summarize these enhancements below:

| **Main Concerns**                          | **Reviewers** | **How We Address It (Key Actions & Findings)**               |
| ------------------------------------------ | ------------- | ------------------------------------------------------------ |
| **1. Deeper Baseline Analysis & Patterns** | G1RX          | **Expanded to 20 Baseline Models:** Evaluated diverse models (proprietary/open/text/multimodal) across sizes. **Confirmed Scaling Laws:** Proved larger models consistently improve Recall/Relevance/Timeliness. Clarified that small models' high precision stems from conservative user need prediction strategies. |
| **2. Bottleneck Identification**           | JomG, G1RX    | **Conducted Oracle Study:** Replaced predictions with ground-truth queries. **Discovered "Dual Bottleneck":** Revealed that JIR suffers from both semantic mismatch of user information need prediction (primary) and a **"Contextual Gap"** (secondary)—even perfect queries fail without considering full context during retrieval (Relevance limited to ~0.27). |
| **3. Multimodal Capabilities**             | TQLe, JomG    | **Provided Visual Evidence:** Listed timestamped examples of essential visual cues (e.g., **gestures**, **board equations**). **Diagnosed Failure Mode:** Attributed performance drop to information needs' **Temporal Misalignment** (86.79% impact) in dynamic video streams, identifying a key challenge for future research. |
| **4. Dataset Realism & Validity**          | PV32, TQLe    | **Conducted User Study:** Collected real needs from human participants. **Validated Coverage:** Achieved **62.7% match rate** between real user queries and our benchmark, confirming ecological validity. **High Quality:** Human benchmark dataset quality verification scores consistently >3.2/4.0. |
| **5. Evaluation Rigor**                    | JomG          | **Clarified Formalized Algorithms:** Provided set-theoretic definitions for the **Many-to-Many Matching Algorithm**. **Sensitivity Analysis:** Proved baseline rating stability (Spearman corr. 0.956) across varying evaluation metric parameters (e.g., similarity thresholds and time windows). |
| **6. Scale & Scope**                       | PV32, TQLe    | **Clarified Rationale:** Prioritized **"Quality & Depth over Scale"** for the first foundational benchmark (POMDP formulation + dense annotation). **Extensibility:** Emphasized that the framework allows easy expansion to other domains (meetings, consumer media). |

## Conclusion

JIR-Arena provides the community with the first mathematical definition, standardized metrics, and a high-quality benchmark for JIR agents. With the primary contributions recognized by reviewers and the analysis fortified during rebuttal, we are ready to release the code, data, and baselines to accelerate research in this domain.

## Note on Final Reviewer Responses

We wish to bring to your attention that Reviewer **G1RX explicitly mentioned in their response that they intend to provide feedback by last Friday (Nov 28th)**, but the discussion period concluded before their response could be posted. Similarly, we were unable to receive final confirmations from Reviewers JomG and PV32 despite our comprehensive responses to their queries. Given these timing constraints preventing further reviewer interaction, **we respectfully rely on your judgment** to assess the strength of our rebuttal and the summary table above in determining whether the reviewers' concerns have been effectively addressed.

Best regards,

Authors

---

### Meta-Review · Area_Chair_1HYK · 2025-12-16

**Summary:**

The paper introduces JIR-Arena, the first large-scale benchmark and formal evaluation framework for Just-in-Time Information Recommendation (JIR) systems, which aim to deliver the most relevant information at precisely the moment users need it most. Despite growing interest driven by advances in foundation models and always-on intelligent devices, prior work has lacked a rigorous task definition, standardized metrics, and a high-quality dataset for systematic research. To address this gap, the authors offer a mathematical formalization of JIR tasks and associated evaluation criteria that capture not only recommendation relevance but also timeliness and avoidance of distracting content. They construct JIR-Arena, a multimodal dataset consisting of 34 richly annotated real-world scenes with oracle user information needs spanning diverse scenarios. The benchmark is designed to test three core capabilities: inferring user information needs, recommending helpful content promptly, and minimizing irrelevant suggestions. Baseline evaluations using foundation models reveal substantial challenges, with strong performance in precision but weakness in recall. Analyses identify semantic mismatch between predicted needs and retrieval as a major bottleneck.

**Reviewer Concerns:**

- C1: The analysis in the paper is not deep enough; the main takeaway remains unclear in the paper; more insightfuly analysis to the observation presented in the paper is needed (two reviewers with negative ratings: Reviewer G1RX and Reviewer JomG).
- C2: Reviewers questioned on the realism and validity of the proposed dataset.
- C3: Reviewers questioned on the multi-modal capabilities: e.g., the video content is processed primarily via NVILA narrations converted to text, making the input effectively text-based.

I appreciate the the authors' huge efforts in addressing the reviewer's concerns in the rebuttal phase. I can see the authors provided a number of additional experiments and analysis during this period.
I think most of the concerns regarding to C2 and C3 are properly addressed in the current version of the paper, while noting that for C1, which might be subjective to some extent, the conerns might still exist.

**Reviewer Scores:**

Though at the initial ratings for this paper are divergent: two reviewers support this paper; two reviewers object to this paper. I would expect the reviewers will increase their ratings for this paper: from (6,6,4,2) to (6, 6, 4, 4), (6, 6, 6, 4) or even (6, 6, 6, 6). This paper is a borderline paper for me (maybe marginally below the acceptance criteria).

---

### Decision · Program_Chairs · 2026-01-26

Reject